# Cell surface patching via CXCR4-targeted nanothreads for cancer metastasis inhibition

Minglu Zhou[1,2], Chendong Liu[1,2], Bo Li[1], Junlin Li[1], Ping Zhang[1], Yuan Huang[1] & Lian Li[1] ✉

The binding of therapeutic antagonists to their receptors often fail to translate into adequate manipulation of downstream pathways. To fix this 'bug', here we report a strategy that stitches cell surface 'patches' to promote receptor clustering, thereby synchronizing subsequent mechano-transduction. The "patches" are sewn with two interactable nanothreads. In sequence, Nanothread-1 strings together adjacent receptors while presenting decoy receptors. Nanothread-2 then targets these decoys multivalently, intertwining with Nanothread-1 into a coiled-coil supramolecular network. This stepwise actuation clusters an extensive vicinity of receptors, integrating mechano-transduction to disrupt signal transmission. When applied to antagonize chemokine receptors CXCR4 expressed in metastatic breast cancer of female mice, this strategy elicits and consolidates multiple events, including interception of metastatic cascade, reversal of immunosuppression, and potentiation of photodynamic immunotherapy, reducing the metastatic burden. Collectively, our work provides a generalizable tool to spatially rearrange cell-surface receptors to improve therapeutic outcomes.

Nearly all breast cancer mortality stems from metastasis, whereby tumor cell "seeds" selectively disseminate along chemokine gradients to organs furnished with pre-metastatic niches (PMN "soils") secreting chemokines[1–3]. CXC chemokine receptor 4 (CXCR4), a G protein-coupled receptor spanning cell membranes to covert extracellular chemokine CXCL12 binding into intracellular signaling, is frequently hijacked by breast cancer, exerting multifaceted effects on metastatic seeds, pro-metastatic PMN soils, and their crosstalk[4,5]. CXCR4 exists in dynamic equilibrium as monomers, dimers, and higher-order assemblies[6]. This flexibility in aggregation and dissociation allows cells to correctly sense gradients, adapt their migration, and metastasize in a non-random fashion[7]. So far, the only two licensed CXCR4-targeted therapies are bicyclam AMD3100 and cyclic-peptide motixafortide, both relying on single molecule-receptor interactions. Strategies beyond standard monovalent antagonism are currently lacking. Theoretically, spatially rearranging cell-surface CXCR4, such as simultaneously crosslinking multiple CXCR4 to form receptor condensate, can perturb the dynamic CXCR4 monomer-dimer-multimer equilibrium, thereby thwarting metastasis. However, it remains unclear whether clustering CXCR4 through artificial self-assembly or mechanical traction leads to allosteric regulation of downstream signaling network, and how it relates to final therapeutic outcome of metastasis inhibition.

Recent growing evidence has highlighted that CXCR4 antagonists capable of dimerizing or oligomerizing the receptors can better hinder metastasis[8–12]. Individual CXCR4-antagonizing peptides or liposomes functionalized with lower peptide densities failed to induce the same anti-metastatic effect as when the peptides were arrayed on a liposome surface at a defined density that mimics the distances between neighboring CXCR4 on the cell membrane[8]. The trivalent ligand, with a rigid linker length enabling it to reach a third CXCR4 receptor at a proximal distance, showed a higher rate of competitive inhibition compared to its bivalent counterpart[9]. Polymer-CXCR4 antagonists, enabling tunable multivalent display of ligands,

[1]Key Laboratory of Drug-Targeting and Drug Delivery System of the Education Ministry and Sichuan Province, Sichuan Engineering Laboratory for Plant-Sourced Drug and Sichuan Research Center for Drug Precision Industrial Technology, West China School of Pharmacy, Sichuan University, Chengdu 610041, China. [2]These authors contributed equally: Minglu Zhou, Chendong Liu. ✉e-mail: liliantripple@163.com

significantly suppressed CXCL12-induced cell migration compared to free antagonists[10,11]. Additionally, increasing the valency of these antagonists further enhances their ability to inhibit metastasis[12]. These findings suggest that antagonists presented by nanoconstructs with diverse modes of crosslinking CXCR4 may cause specific conformational changes, differentially disrupt signal transmission, and lead to an escalation in metastasis inhibition efficiency that positively correlates with the number of engaged receptors per cluster. Therefore, complete eradication of metastasis may be warranted through CXCR4 antagonism within larger receptor cluster. Nevertheless, the size of multivalent constructs limits the further increase of concurrently crosslinked receptors, as receptors in a wider vicinity are beyond the reach of the nanoscale scaffold. Moreover, random receptor collisions in individually scattered clusters can generate separate mechanotransduction pathways that function asynchronously, thereby impairing clustering efficiency[13–15]. To amplify the overall outcome, a next-generation strategy requires synchronizing CXCR4 clustering over an expanded cell surface area.

Beyond metastasis inhibition, CXCR4 antagonism has been reported to alleviate the hypoxic microenvironment within tumors[16], potentially augmenting the effectiveness of oxygen-dependent photodynamic therapy (PDT). Moreover, CXCR4 antagonism can reverse the immunosuppressive microenvironment of tumors[17], potentially enhancing the anti-tumor immune response induced by PDT. Therefore, we envision CXCR4 antagonism might also benefit PDT for better tumor eradication.

In this study, we propose a stepwise strategy for hierarchically enlarging CXCR4 clustering using two interactable polymer nanothreads. The process involves sequential delivery of Nanothread-1 and Nanothread-2, akin to sewing patches on the cell surface. Nanothread-1, delivered first, comprises a polymeric string skeleton with multiple copies of two pendant segments − targeting segments that anchor to CXCR4 receptors, and random coil segments that function as decoy receptors. Nanothread-2, delivered subsequently, has an identical string skeleton grafted with multiple complementary coil segments that can form coiled-coil structures with Nanothread-1 decoys. We show Nanothread-1 strings numerous receptors together while multivalently presenting decoy receptors on the cell surface. Nanothread-2 then stretches for the decoys, concurrently crosslinking with different chains of neighboring Nanothread-1. This biorecognition intertwines multiple Nanothread-1 and Nanothread-2 strings, assembling a netlike patch on the cell surface. This patch connects an expanded area of CXCR4 receptors and integrates them into a supercluster that produces an amplified anti-metastatic effect. Additionally, photosensitizers are further conjugated onto Nanothread-2 for a 'hitchhike' to tumor for targeted PDT. As a result, nanothread 'patching' seals the potential for spontaneous metastasis, constraining cancer cells within the primary tumor. In parallel, tumor-localized PDT transforms the tumor into an in situ vaccine by inducing immunogenic cell death (ICD), triggering local anti-tumor immune responses while establishing abscopal protection against disseminated metastasis (Fig. 1).

## Results

### Coiled-coil driven assembly to supramolecular network

Nanothread-1 comprises a linear polymer backbone grafted with multiple copies of a CXCR4 binding sequence (BS) and a coiled motif (CM1), designated P-BS-CM1. Nanothread-2 comprises an identical backbone grafted with multiple copies of a complementary coiled motif (CM2) and chlorin e6 photosensitizer (PS), designated P-PS-CM2. The complementary CM1 and CM2 pentaheptad peptides were designed to enable coiled-coil assembly through optimized hydrophobic cores, electrostatic interfaces, and helical propensities[18–20], as shown by the wheel diagram (Fig. 2a). The synthetic schemes are shown in Supplementary Fig. 1. To synthesize the linear backbones, N-

(2-hydroxypropyl) methacrylamide (HPMA) polymer was prepared by reversible addition-fragmentation chain transfer polymerization, followed by the partial conversion of amine pendants into maleimides. P-BS-CM1 was fabricated by sequentially attaching BS and CM1 peptides to the backbone via thiol-ene click chemistry between the maleimides and cysteines embedded within BS or at CM1 N-termini. P-PS-CM2 was fabricated by sequentially attaching PS and CM2 to the backbone through amide bond formation and thiol-ene click chemistry, respectively. Comprehensive characterization of the nanothreads, derivatives, and controls is presented in Supplementary Table 1. All constructs exhibited well-defined architectures with narrow polydispersity. While their molecular weight, ζ-potential, size, fluorescence labels, and PS content varied slightly depending on different grafts, BS, CM1, and CM2 valences were precisely controlled to 6, 4, and 12, respectively. The 3-fold higher CM2 valence (12 copies on Nanothread-2) compared to the CM1 valence (4 copies on Nanothread-1) was intentionally designed to promote extensive multivalent interactions, such that each P-PS-CM2 string could simultaneously engage multiple flanking P-BS-CM1 strings via the complementary coiled-coil motifs. This aimed to drive the assembly of an expansive interchain network. Since P-BS-CM1 and P-CM2 have similar polymer lengths but differ in coiled motif valency, mixing them at various CM1-to-CM2 ratios may lead to distinct structure formations (Fig. 2b). (1) When the CM1 ratio is excessively high, P-CM2 exhibits a preference for acting as a backbone scaffold for branched polymers, allowing the grafting of multiple P-BS-CM1 while also leaving a partial proportion of P-BS-CM1 unbounded. (2) At an optimal ratio, multiple chains of P-BS-CM1 and P-CM2 can crosslink and self-assemble into a supramolecular network. (3) At a lower CM1 ratio, steric hindrance and motif accessibility increase the likelihood of P-BS-CM1 and P-CM2 forming dual-chain structures. To characterize polymer assembly with varying CM1-to-CM2 ratios while maintaining a constant total polymer concentration, three ratios (3:1, 1:1, and 1:3) were selected.

To confirm the interchain biorecognition, förster resonance energy transfer (FRET) pair, Cy5 and Cy3, were conjugated to P-BS-CM1 and P-CM2, respectively. Fluorescent spectra at Fig. 2c showed that irradiation of P-BS-CM1-Cy5+P-CM2-Cy3 with equimolar CM1-to-CM2 mixture at the Cy3 excitation wavelength of 550 nm gave rise to a peak at the Cy5 emission wavelength of 670 nm, clearly indicating an energy transfer due to the close interactions of the two polymers. As expected, circular dichroism spectra showed that P-BS-CM1 and P-CM2 individually were unfolded, whereas significant signal of coiled-coil formation (minima at 208 and 222 nm) was observed upon their CM1-to-CM2 equimolar mixture (Fig. 2d). Comparatively, P-BS-CM1 and P-CM2 mixtures at the imbalanced ratios (CM1:CM2 = 3:1 and 1:3) generated impaired coiled-coil signals. Relative to the individual nanothreads (~20 nm), dynamic light scattering revealed spontaneous expansion of P-BS-CM1 + P-CM2 (CM1:CM2 = 1:1) into supermolecules larger than 1000 nm. Meanwhile, the major peaks of P-BS-CM1 and P-CM2 mixture at the imbalanced ratio (CM1:CM2 = 3:1 and 1:3) shifted to ~500 nm and ~30 nm, respectively, forming smaller assemblies (Fig. 2e). Additionally, rheological measurements demonstrated sol-to-gel transition for P-BS-CM1 + P-CM2 (CM1:CM2 = 1:1), which was in contrast to polymers alone or their unequal CM1-to-CM2 molar mixtures (Fig. 2f). Scanning electron microscopy revealed transformation from a flat texture of the polymer alone to an organized, porous morphology upon gelation of P-BS-CM1 + P-CM2 (CM1:CM2 = 1:1) (Fig. 2g). The morphology of P-BS-CM1 + P-CM2 (CM1:CM2 = 1:3) remained close to that of polymer alone, while P-BS-CM1 + P-CM2 (CM1:CM2 = 3:1) showed an inhomogeneous morphology, resembling an intermediate state between a flaky texture and porous structure. These data demonstrate Nanothread-1 and Nanothread-2 can spontaneously assemble into supramolecular networks through coiled-coil interactions when appropriately mixed.

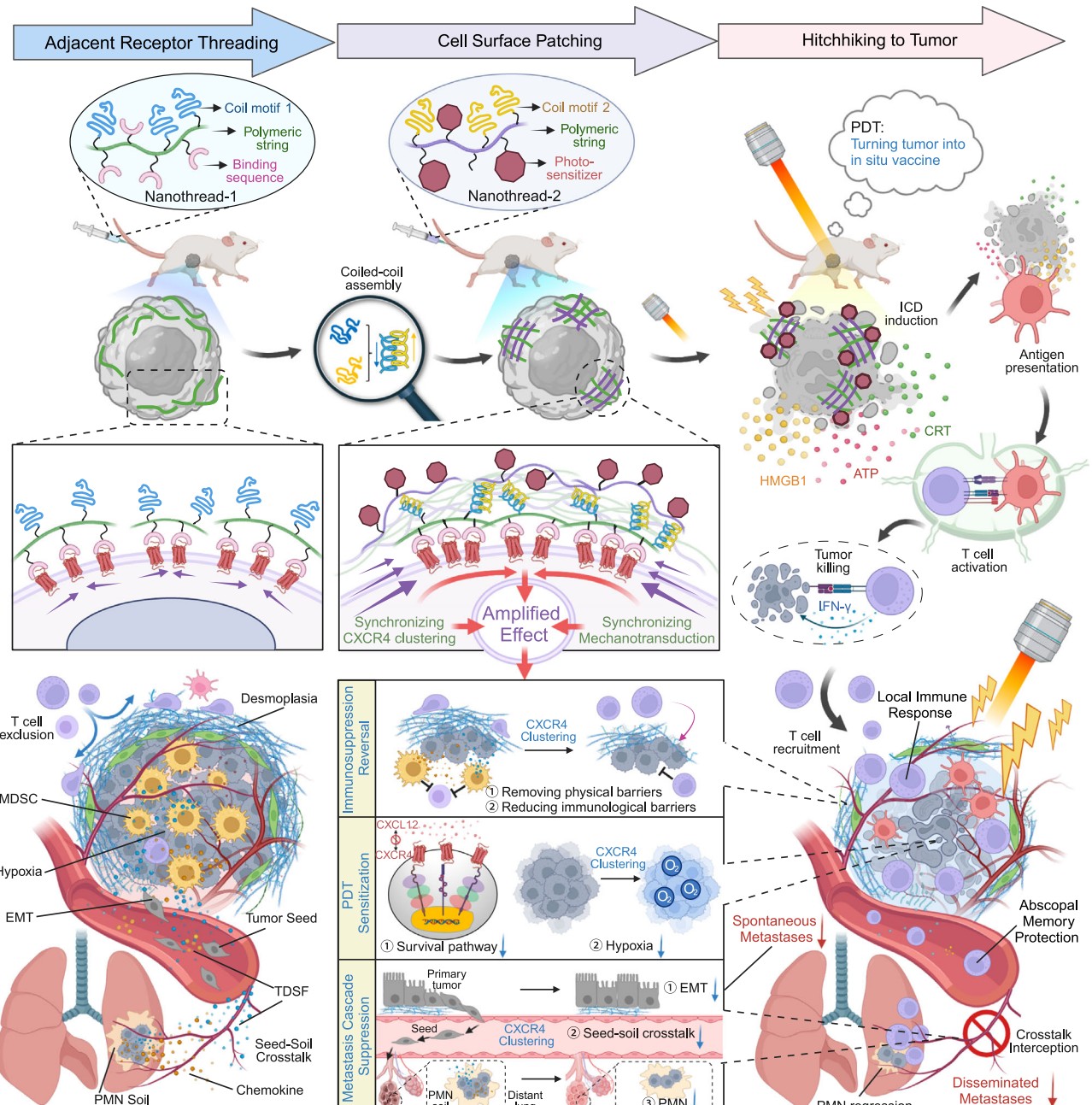

**Fig. 1 | Stepwise nanothread patching to escalate C-X-C motif chemokine receptor 4 (CXCR4) clustering and inhibit cancer metastasis.** In stage I, Nanothread-1 recognizes and multivalently binds CXCR4, aligning adjacent receptors while presenting coiled motif decoys on the cancer cell surface. Subsequently in stage II, Nanothread-2 engages these decoys, intertwining with Nanothread-1 into a coiled-coil supramolecular network. This sequential actuation is proposed to generate physical dragging forces that cluster an expanded vicinity of receptors, synchronizing CXCR4 clustering and amplifying mechanotransduction to enable adequate manipulation of downstream signaling events. Concurrently, photosensitizers 'hitchhike' on Nanothread-2 to the tumor site for targeted photodynamic therapy (PDT), inducing immunogenic cell death (ICD) to transform the primary tumor into an in-situ vaccine. Nanothread 'patching' on CXCR4 superclusters is proposed to seal spontaneous metastatic potential by disrupting the metastasis cascade – reshaping metastatic tumor cell 'seeds', intercepting 'seed-soil' crosstalk, and regressing the pre-metastatic niche 'soil'. Moreover, manipulated CXCR4 clustering downstream effects, including survival pathway interference, hypoxia alleviation and immunosuppression reversal, are expected to sensitize the tumor to PDT. Consequently, a localized anti-tumor immune response would be initiated against the primary tumor, while also establishing an abscopal memory effect against disseminated metastases. CRT, calreticulin; ATP, adenosine triphosphate; HMGB1 high mobility group protein B1; IFN-γ, interferon-γ; MDSC, myeloid-derived suppressor cells; EMT, epithelial–mesenchymal transition; TDSF, tumor-derived secreted factors; PMN, pre-metastatic niche; CXCL12, Chemokine (C-X-C Motif) Ligand 12. Created with BioRender.com.

## Manipulation of receptor clustering

Having demonstrated the interchain assembly, we next examined the cell-surface distribution of sequentially delivered Nanothread-1 and Nanothread-2. Initially, the hydrophilicity and macromolecular size of P-CM1-Cy5 resulted in minimal cellular uptake (Supplementary Fig. 2).

After multivalent modification with BS, P-BS-CM1-Cy5 showed a uniform ring-like binding pattern around 4T1 cell surfaces (Fig. 3a). We further demonstrated high specificity of BS for CXCR4 rather than other chemokine receptors frequently observed in breast cancer (Supplementary Fig. 3a). Moreover, P-BS-CM1-Cy5 exclusively bound

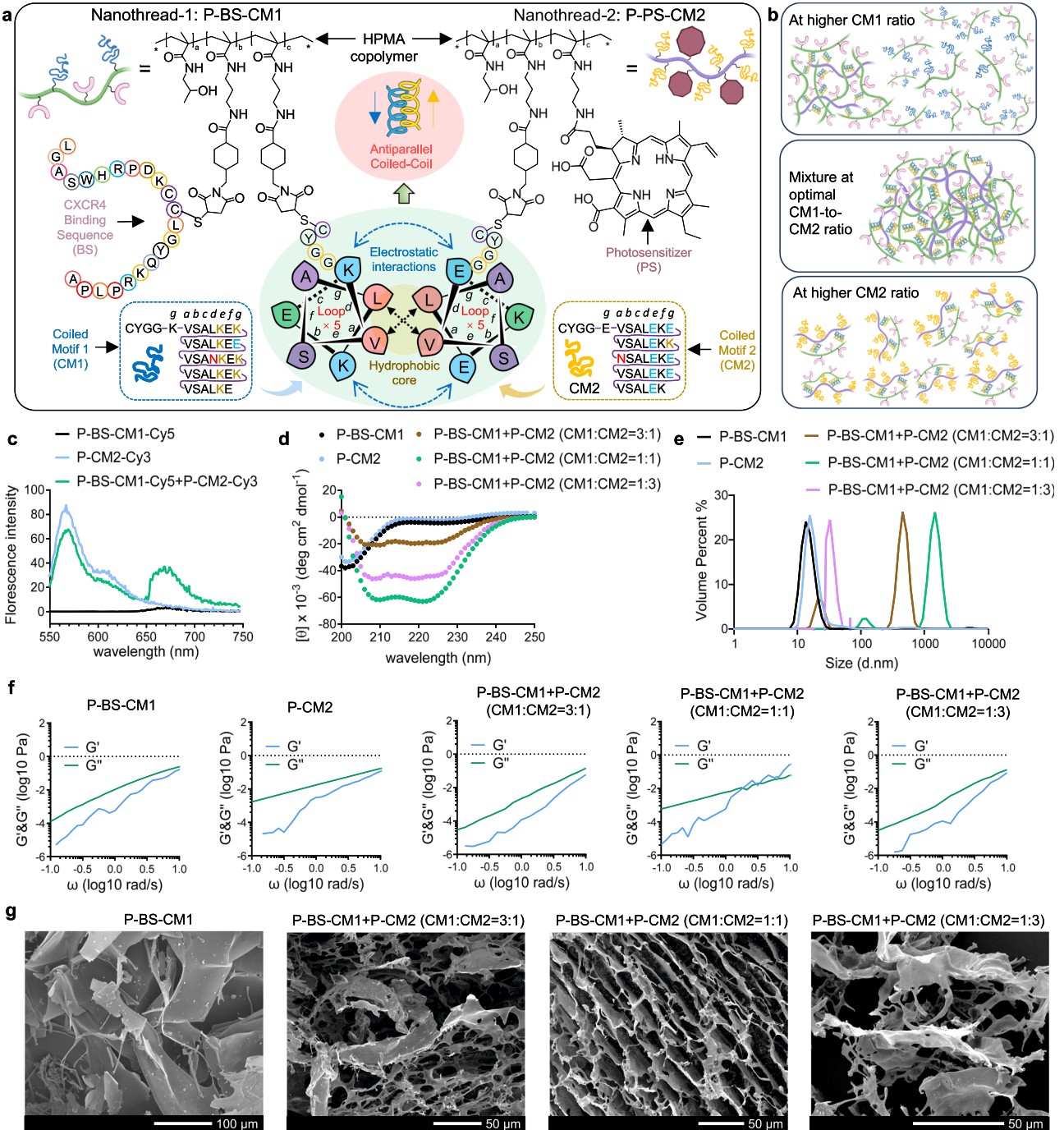

**Fig. 2 | Coiled-coil mediated self-assembly into supramolecular networks.**
**a** Illustration of Nanothread-1 and Nanothread-2 components and helical wheel representation of their antiparallel heterodimeric coiled-coil interaction. CM1 and CM2 were designed with complementary pentaheptad sequences to enable coiled-coil assembly, incorporating stabilizing hydrophobic, electrostatic, and helical propensity effects. The hydrophobic core comprised valine (V) and leucine (L) at the *a* and *d* positions, with one buried asparagine (N) polar substitutions at one position *a* for CM1 and *d* for CM2, imposing antiparallel alignment. Positively charged lysine (K) at *e* and *g* in CM1 enabled electrostatic interactions with negatively charged glutamic acid (E) at *e* and *g* in CM2. Oppositely charged substitutions at one *g* improved orientation specificity. Uncharged serine (S) and alanine (A) at *b* and *c* promoted solubility and helicity. Balanced glutamic acid (E) in CM1 and lysine (K) in CM2 at *f* maintained net charge. N-terminal tetrapeptide spacers enabled

polymer conjugation. **b** Schematic illustration of coiled-coil interaction between P-BS-CM1 and P-CM2 at various CM1-to-CM2 ratios, leading to distinct formations including branched polymer, supramolecular network, and dual-chain structure. **c** Representative fluorescent spectra of P-BS-CM1-Cy5, P-CM2-Cy3, and their CM1-to-CM2 equimolar mixture. The spectra were recorded at the Cy3 excitation wavelength of 550 nm, 25 °C in 10 mM PBS, pH 7.4 after 10 min mixture. **d–g** Representative circular dichroism spectra (**d**), size distribution (**e**), frequency-dependent rheology (**f**), and scanning electron microscopy images (**g**) of individual and mixed P-BS-CM1 and P-CM2 (CM1:CM2 = 3:1; 1;1, 1:3). G′, storage modulus; G″, loss modulus. The experiments in (**c**–**g**) were repeated three times independently with similar results. Chemical structures in (**a**) are created with ChemDraw Professional 16.0 software. Created with BioRender.com (**a**, **b**). Source data are provided as a Source Data file.

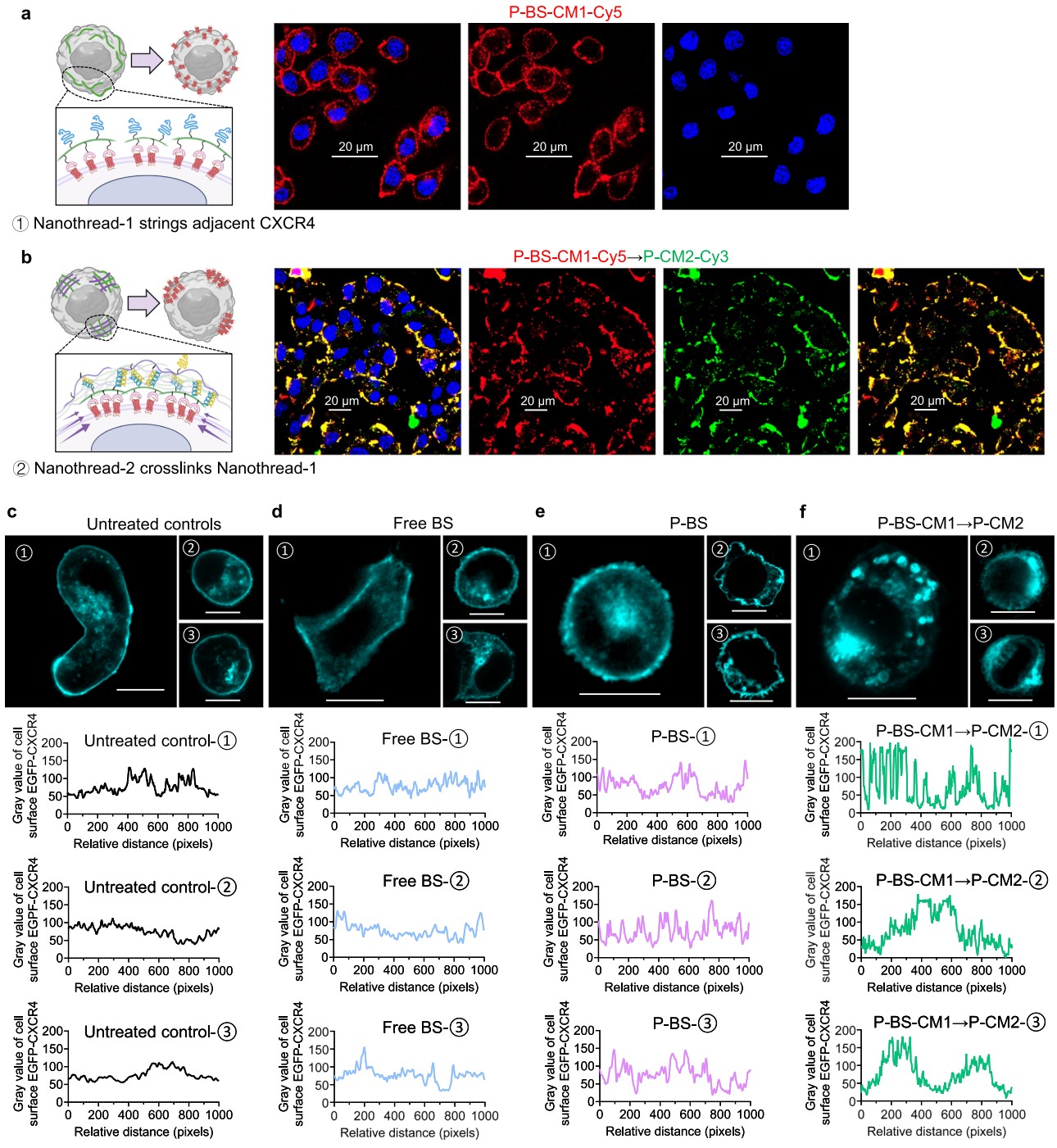

**Fig. 3 | Manipulation of cell surface CXCR4 clustering. a, b** Representative confocal microscopy images of 4T1 cells sequentially treated with (**a**), P-BS-CM1-Cy5 for 1 h followed by cell culture medium for another 1 h; or (**b**), P-BS-CM1-Cy5 for 1 h followed by P-CM2-Cy3 for another 1 h. Blue indicates cell nuclei, red indicates Cy5, green indicates Cy3, yellow shows overlay of Cy5 and Cy3. Scale bars, 20 μm. **c–f** Representative confocal microscopy images and fluctuations curves depicting distance-dependent enhanced green fluorescent protein (EGFP)-tagged CXCR4 fluorescent intensity on the cell membrane after EGFP-CXCR4-transfected cells were either (**c**), left untreated or treated with (**d**), free BS for 1 h followed by cell culture medium for another 1 h, (**e**) P-BS for 1 h followed by cell culture medium for another 1 h, or (**f**) P-BS-CM1 for 1 h followed by P-CM2 for another 1 h (1 mg/mL BS equivalence, CM1:CM2 = 1:1 mol%). After treatments, singly dispersed cells at the same normal state were selected and imaged. Cyan indicates EGFP. Scale bar, 20 μm. The experiments in (**a–f**) were repeated three times independently with similar results. Images in (**c–f**) were analyzed by the Image J software. Source data are provided as a Source Data file.

to CXCR4 receptors on 4T1 cells (Supplementary Fig. 3b). Notably, consecutive P-BS-CM1-Cy5→P-CM2-Cy3 delivery displayed highly overlapped fluorescence on cell surface (Fig. 3b), validating in situ self-assembly. Most significantly, subsequent addition of P-CM2-Cy3 drastically altered the distribution of pre-targeted P-BS-CM1-Cy5, leading to speckled aggregates with enlarged punctate fluorescence

on cell membrane (Fig. 3b). This observation strongly indicates that netlike crosslinking of Nanothread-2 with CXCR4-anchored Nanothread-1 formed cell surface 'patches' that provoked extensive receptor drifting.

To directly confirm whether the cell surface 'patching' through nanothreads induced CXCR4 clustering, we established enhanced

green fluorescent protein (EGFP)-CXCR4-expressing 4T1 cells through plasmid transfection. Confocal microscopic analysis of EGFP-CXCR4-transfected cells revealed the presence of CXCR4 both in the cytoplasm and on the cell surface (Fig. 3c and Supplementary Fig. 4). In individual cells, CXCR4 fluorescence was predominantly distributed uniformly on the cell membrane, interspersed with several small concentrated fluorescent spots (Fig. 3c). This aligns with previous findings that G protein-coupled receptors exist in equilibrium between monomers and oligomers[6,7]. It should be noted that CXCR4 fluorescence could be enriched at the contact interface between two cells (Supplementary Fig. 4). Thus, to accurately reflect changes to surface CXCR4 spatial organization after treatments, we deliberately imaged singly dispersed cells at the same normal state. Similar to untreated controls, cells treated with free BS antagonist displayed a ring-pattern of EGFP-tagged CXCR4 surrounding the cells. They exhibited no observable effects on escalating the degrees of fluctuations in the curves depicting distance-dependent EGFP-tagged CXCR4 fluorescent intensity on the cell membrane (Fig. 3d). Treatment with P-BS resulted in moderate changes in the fluorescent fluctuations around EGFP-CXCR4-transfected cells, whereas the traction from a single multivalent polymer was insufficient to cause noticeable receptor aggregation (Fig. 3e). Interestingly, being different from BS and P-BS, sequential nanothreads P-BS-CM1 → P-CM2 effectively aggregated adjacent CXCR4 into receptor condensates on the cell membrane. This exhibited adequate traction to successfully induce CXCR4 clustering, accompanied by substantial fluctuation in cell surface fluorescent intensity (Fig. 3f). These results directly demonstrated that P-BS-CM1 → P-CM2 could considerably escalate CXCR4 clustering and perturb their spatial organization on cell surface.

## Amplified disruption of downstream pro-metastatic signal

Prior studies have demonstrated that stimulation of G-protein-coupled CXCR4 by CXCL12 activates G-proteins, eliciting extracellular calcium influx[4,6]. For further investigation, 4T1 cells were pretreated with free BS, P-BS, or sequential P-BS-CM1 → P-CM2, followed by CXCL12 stimulation. Thereafter, inhibition of calcium influx was evaluated. As anticipated, untreated 4T1 cells exhibited significantly elevated intracellular calcium upon CXCL12 stimulation (Supplementary Fig. 5). Critically, the mode of CXCR4 antagonism considerably impacted the susceptibility of calcium influx to inhibition. Multivalent CXCR4 binding by P-BS demonstrated greater efficacy in inhibiting calcium influx versus monovalent binding by free BS. However, cell surface 'patching' through netlike crosslinking of P-BS-CM1 and P-CM2 provoked an even more pronounced effect (Fig. 4a), suggesting that P-BS-CM1 → P-CM2 might exert greater manipulation on downstream signaling.

Proteomics analyses revealed pronounced alterations in the differentially expressed protein (DEP) profile (FC > 2; $P < 0.05$) of 4T1 cells following P-BS-CM1 → P-CM2 treatment, whereas free BS and P-BS elicited negligible effects (Fig. 4b–d). Gene Ontology enrichment of the top 20 DEPs uncovered significant enrichment for biological processes related to CXCR4 spatial reorganization, including ion transport, extracellular structure/matrix organization, molecular function regulation, and secretion (Fig. 4e). This signifies escalated CXCR4 clustering by P-BS-CM1 → P-CM2 reshapes the cell phenotype. Additional assessments of CXCR4 downstream signaling cascades demonstrated P-BS-CM1 → P-CM2, compared to untreated controls or monovalent/multivalent competitors, markedly upregulated cell death pathway proteins while downregulating cell survival and antioxidative damage proteins (Fig. 4f), implying reduced malignancy and increased susceptibility to tumoricidal therapies. Moreover, P-BS-CM1 → P-CM2 substantially downregulated transduction of CXCR4-CXCL12 axis intracellular signals, Ras-associated metastatic pathways, proteins involved in cell migration/adhesion, tumor invasion/

dissemination, alongside expression/secretion of metastasis-promoting factors (Fig. 4f). In vitro wound healing assay, transwell migration and invasion assay verified P-BS-CM1 → P-CM2 exhibited the greatest capacity to inhibit the lateral mobility, longitudinal mobility, and invasiveness of 4T1 cells (Fig. 4g and Supplementary Fig. 6). Collectively, these findings demonstrate escalated CXCR4 clustering by P-BS-CM1 → P-CM2 can profoundly amplify disruption of downstream pro-survival and pro-metastatic signaling, potentially sealing the metastatic potential of tumor cells.

## In vivo biorecognition

We next investigated the biorecognition of sequentially delivered Nanothread-1 and Nanothread-2 in mice bearing orthotopic 4T1 breast tumors overexpressing CXCR4. Figure 5a compared tumor targeting of P-CM1-Cy5 and P-BS-CM1-Cy5 at the first step. After intravenous injection, P-BS-CM1-Cy5 with the CXCR4-specific BS exhibited substantially higher tumor accumulation than P-CM1-Cy5 that passively reached the tumor. Figure 5b studied the tumor arrival of P-CM2-Cy5 at the second step. Mice were consecutively treated with P-BS-CM1 and P-CM2-Cy5 with a lag of 24 h. The time interval was selected because substantial P-BS-CM1-Cy5 had accumulated in tumors while its circulating levels declined sharply at 24 h post-injection. Compared to P-CM2-Cy5 alone, P-CM2-Cy5 accumulation in tumors significantly improved with P-BS-CM1 pretreatment (Fig. 5b), due to the interaction between tumor pre-targeted P-BS-CM1 and P-CM2-Cy5 via CM1 decoys and complementary CM2. Following a consecutive delivery of P-BS-CM1-Cy5→P-CM2-Cy3, their in vivo biorecognition was confirmed by the substantial overlap of dual fluorescence and conspicuous generation of FRET signal in tumor tissues (Fig. 5c). Compelling in vitro evidence showed that multivalent P-BS-CM1 binding to CXCR4 receptors (Fig. 3a) followed by P-CM2 crosslinking (Fig. 3b) created cell-surface CXCR4 superclusters (Fig. 3f) that enhanced downstream interference of calcium influx associated with CXCL12 signaling (Fig. 4a). We further confirmed that this process could be recapitulated in vivo (Fig. 5d). Following an identical in vitro trend, tumor cells isolated from mice treated with P-BS-CM1-Cy5→P-CM2-Cy3 exhibited significantly higher Cy5 fluorescence compared to P-CM1-Cy5→P-CM2-Cy3 lacking receptor binding capability. Additionally, P-BS-CM1-Cy5→P-CM2-Cy3 generated a considerably stronger FRET signal than P-BS-Cy5→P-CM2-Cy3 lacking coiled-coil interaction. Moreover, P-BS-CM1-Cy5→P-CM2-Cy3 demonstrated a greater efficacy in inhibiting CXCL12-stimulated calcium influx compared to the other two controls. While direct in vivo visualization of receptor clustering was technically constrained in this study, the observed effects on upstream binding and crosslinking events as well as downstream signaling suggest that P-BS-Cy5→P-CM2-Cy3 treatment might promote higher-order CXCR4 assembly beyond multivalent binding.

We next investigated the biodistribution of both nanothreads. After 72 h post injection of Cy5-labeled nanothread (P-BS-CM1-Cy5→P-CM2 or P-BS-CM1 → P-CM2-Cy5), considerable accumulation of either Nanothread-1 or Nanothread-2 was found in tumors, which was significantly higher than in other major organs (Supplementary Fig. 7). This result correlated with previous studies that HPMA polymer-based nanovehicles have excellent tumor targeting capability[21]. We also confirmed the biosafety of four-cycled weekly treatment with P-BS-CM1 → P-CM2 on healthy mice as serum chemistry analysis and hematological cell counts showed no significant differences compared to the control group treated with saline (Supplementary Fig. 8). Moreover, histology analysis of major organs revealed no pathological abnormalities or tissue damage (Supplementary Fig. 9), indicating no obvious off-target toxicity.

Sufficient circulation and efficient tumor localization are essential prerequisites for achieving receptor clustering through nanothread assembly at the tumor cell surface. As shown in Supplementary Fig. 10, we compared the effects of consecutive delivery, simultaneous

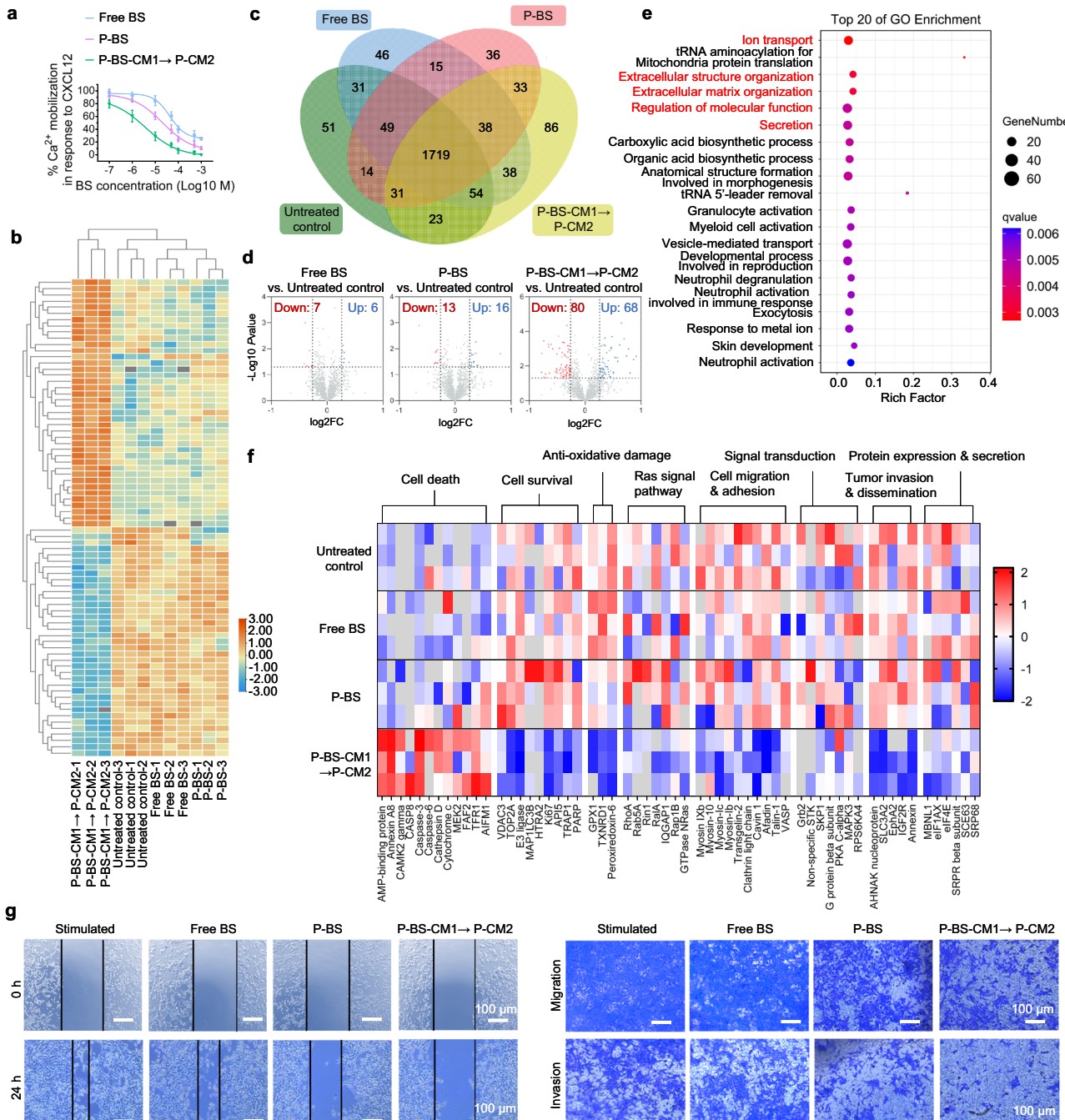

**Fig. 4 | Manipulation of downstream signaling.** Untreated 4T1 cells were stimulated with CXCL12 for 1 h. For free BS and P-BS groups, 4T1 cells received the treatments for 1 h, followed by culture in fresh medium for 25 h and CXCL12 stimulation, prior to analysis. For the P-BS-CM1 → P-CM2 group, 4T1 cells underwent consecutive treatment with P-BS-CM1 for 1 h and P-CM2 for 1 h, then culture in fresh medium for 24 h and CXCL12 stimulation before analysis. **a** Intracellular calcium levels in CXCL12 stimulated 4T1 cells in response to escalating concentrations of free BS, P-BS, and sequential P-BS-CM1 → P-CM2. $n = 3$ biologically independent samples per group. Data were presented as mean ± SD. **b**–**f** Proteomics analyses of 4T1 cells left untreated or treated with free BS, P-BS, or sequential P-BS-CM1 → P-CM2, $n = 3$ biologically independent samples per group, including: (**b**) clustering heat map reflecting the overview of significantly regulated differentially expressed proteins (DEPs) with FC > 2 and $P < 0.05$ determined via one-way comparison; (**c**) four-list Veen diagram of DEPs depicting the number of shared and unique proteins in each group; (**d**) volcano plot exhibiting significantly up/down regulated proteins identified through multiple comparison; (**e**) enrichment analysis of Gene Ontology (GO) terms of DEPs performed via one-way comparison; (**f**) heat map display of selected proteins involved in cell survival and tumor metastasis. **g** Representative images of evaluating lateral mobility, longitudinal mobility, and invasiveness of 4T1 cells by wound healing, migration, and invasion assays. The experiments in (**a**, **g**) were repeated twice independently with similar results. Source data are provided as a Source Data file.

delivery, and post-assembly delivery of both Nanothread-1 and Nanothread-2. With a 24-hour lag in consecutive delivery, the time-staggered approach enabled substantial tumor arrival of the two interactable nanothreads after sequential injection into the circulation. In stark contrast, post-assembly delivery of pre-mixed Nanothread-1

and Nanothread-2 led to a significant decrease in tumor accumulation of both nanothreads, due to the rapid clearance of large particles by reticuloendothelial system in blood. Compared with consecutive delivery, considerable decrease in circulation half-life and tumor accumulation was also observed in simultaneous delivery. This was

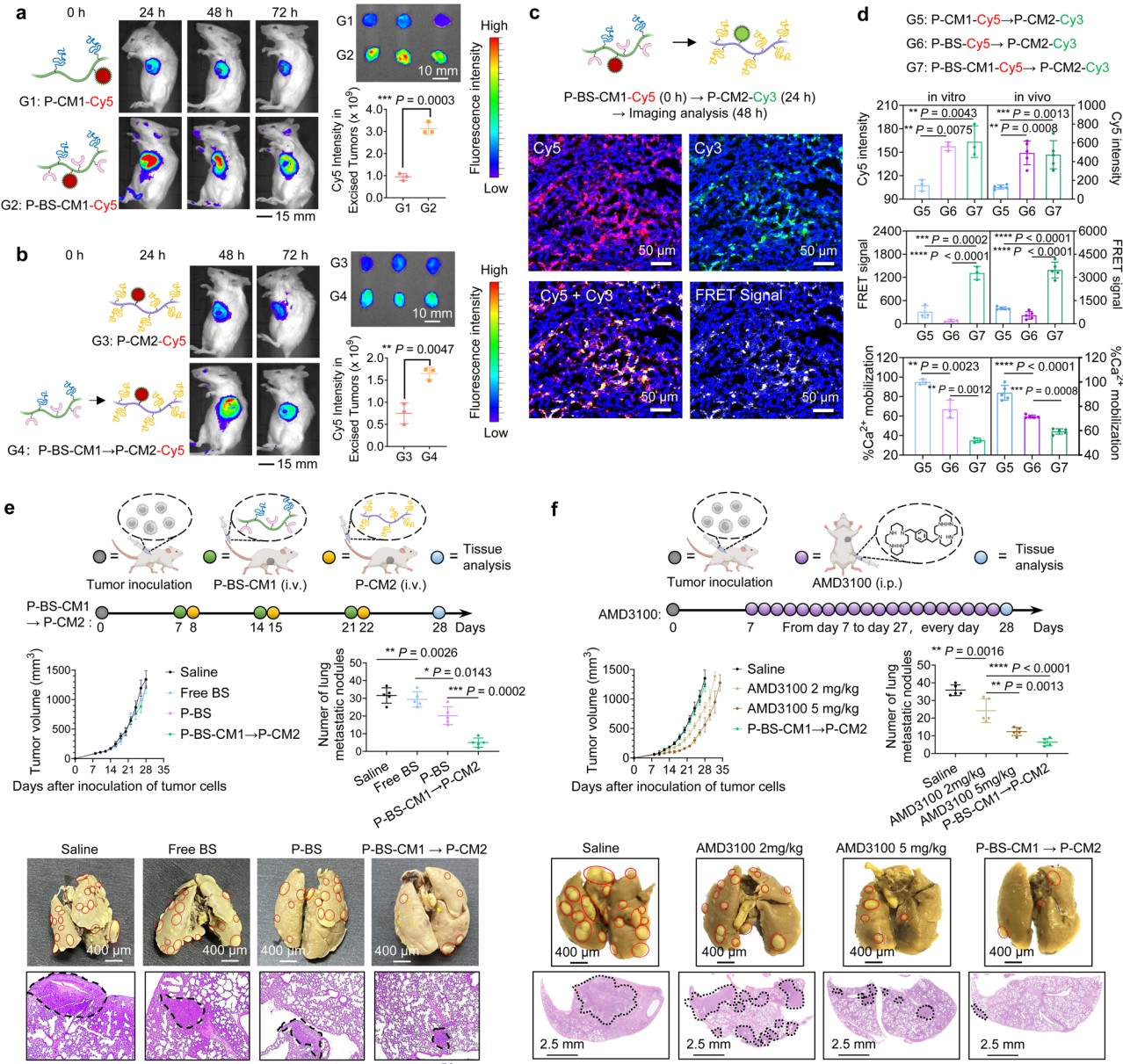

**Fig. 5 | In vivo metastasis inhibition. a** Representative images exhibiting tumor accumulation of Nanothread-1. BABL/c female mice orthotopically bearing 4T1 breast tumors were intravenously injected with either P-CM1-Cy5 or P-BS-CM1-Cy5. $n = 3$ animals per group. **b** Representative images showing tumor accumulation of Nanothread-2. After intravenous injection with saline or P-BS-CM1, 4T1 tumor-bearing mice were treated with P-CM2-Cy5. $n = 3$ animals per group. **c** Representative cryosections of tumor tissues from orthotopic breast tumor mouse models consecutively treated with P-BS-CM-Cy5 at 0 h and P-CM2-Cy3 at 24 h were analyzed by confocal microscopy at 48 h. Blue indicates cell nuclei, red indicates Cy5, green indicates Cy3, yellow shows overlay of Cy5 and Cy3, and white depicts förster resonance energy transfer (FRET) signal between Cy3 and Cy5. Scale bar, 50 μm. **d** Evaluation of receptor clustering in vivo by comparing the cellular (left) and tumoral levels (right) of Nanothread-1 binding, Nanothread-2 cross-linking, and downstream calcium influx interference. $n = 3$ biologically independent samples per group for in vitro studies, $n = 5$ animals per group for in vivo studies. **e, f** Tumor volume changes, analysis of lung metastatic nodules, and representative histology images of lung lobe sections at the endpoint after treatments. Spontaneous lung metastasis mouse models of female BALB/c mice orthotopically bearing murine 4T1 breast tumors received (**e**) three cycles of intravenously administered free BS, P-BS, or sequential P-BS-CM1 → P-CM2 (24 h time lag) treatments on days 7, 14, and 21; or (**f**), intraperitoneal injection of AMD3100 (2 mg/kg or 5 mg/kg) daily from day 7 to day 27. $n = 5$ animals per group (**e, f**). Metastatic nodules are indicated with red circles in bright field images and black circles in histology images. The experiments in (**a–f**) were repeated twice independently with similar results. Data are presented as mean ± SD, with statistics calculated by Student's two-sided $t$-test (**a, b**) and one-way ANOVA with Tukey's multiple comparisons without adjustments (**d–f**). *$P < 0.05$, **$P < 0.01$, ***$P < 0.001$, ****$P < 0.0001$. Source data are provided as a Source Data file.

most likely attributed to the mutual interaction and crosslinking between two nanothreads in the blood, which accelerated their clearance from circulation.

These results highlighted the necessity of delivering Nanothread-1 and Nanothread-2 consecutively. Since the nanothreads were modified with multiple copies of exogenous peptide segments which often exhibit higher immunogenicity, repeated administration may lead to the production of antibodies against the peptides, potentially inducing immune clearance of nanothreads. We further provided data showing that repetitive injection did not result in accelerated clearance or affect the pharmacokinetics of both nanothreads (Supplementary Fig. 11).

## Inhibition of spontaneous metastasis in vivo

We next sought to validate the anti-metastatic effects in the more complex in vivo setting using an immunocompetent mouse model orthotopically bearing murine 4T1 breast tumors that could spontaneously metastasize to distant lung. During three weekly treatment cycles of free BS, P-BS, or P-BS-CM1 → P-CM2, primary tumor growth was monitored, with lung metastasis analyzed at the study endpoint (Fig. 5e). Examination of lung lobe sections at endpoint revealed high metastatic nodule burdens in the saline and free BS groups. P-BS modestly inhibited metastasis (~30% reduction). In comparison, P-BS-CM1 → P-CM2 significantly reduced pulmonary metastases (~80% inhibition), exhibiting promising anti-metastatic activity. However, P-BS-CM1 → P-CM2 failed to retard primary tumor growth. In another individual experiment (Fig. 5f), a consistent result was obtained for P-BS-CM1 → P-CM2 that exhibited no inhibitory effect on primary tumor, whereas AMD3100, a licensed CXCR4 antagonist, showed a dose-dependent anti-tumor activity. Nevertheless, AMD3100 even daily given at a relatively high dose only delayed the tumor growth initially, with tumor eventually growing large in the end. In terms of metastasis evaluation, weekly intravenous administration of P-BS-CM1 → P-CM2 exerted significantly superior anti-metastasis effect compared with AMD3100 intraperitoneally given daily at 2 mg/kg dose. Furthermore, P-BS-CM1 → P-CM2 also reduced the number of pulmonary metastatic nodules to a greater extent than AMD3100 at the higher dose of 5 mg/kg, although the difference was not statistically different. These results demonstrate P-BS-CM1 → P-CM2 can suppress spontaneous metastasis, but lacks efficacy against primary tumors.

## Interception of spontaneous metastasis cascade

CXCR4 can confer metastatic phenotypes to cancer cells by triggering endothelial-to-mesenchymal transition (EMT), enabling tumor cell "seeds" to depart the primary site and invade distant tissues[5]. However, secondary site "soils" are not passive receivers of circulating tumor cells, but are selectively modified by the primary tumor prior to metastasis[1]. Tumor cell colonization at distant sites relies on primary tumor-derived secreted factors (TDSFs) that establish PMN[2]. CXCR4 reportedly impacts multiple steps of this "seed-soil" crosstalk cascade (Fig. 6a): (i) CXCR4-associated signaling may consolidate tumor desmoplasia and exacerbate hypoxia[16]; (ii) This aggravates TDSF release (e.g., transforming growth factor-β (TGF-β), lysyl oxidase (LOX)) into circulation where they accumulate in target organs, recruiting CD11b+ bone marrow-derived cells (BMDCs)[22,23]; (iii) Recruited BMDCs then produce cytokines (e.g., S100A8) to further attract BMDCs, jointly forming the PMN[24]; (iv) The PMN releases chemokines like CXCL12 to draw CXCR4+ metastatic "seeds" and furnish a hospitable "soil" for colony expansion[22].

Spontaneous lung metastasis mouse models bearing orthotopic 4T1 tumors were administered three cycles of free BS, P-BS, or sequential P-BS-CM1 → P-CM2 (24 h lag) on days 7, 14, and 21. Analyses on day 28 revealed amplified interference of CXCR4 downstream signaling by P-BS-CM1 → P-CM2, with pleiotropic effects on blocking pro-metastatic processes: (i) Investigation of tumor desmoplasia revealed that, compared to free BS and P-BS, P-BS-CM1 → P-CM2 resulted in reduced collagen deposition in the tumor matrix as evidenced by the lowest levels of intratumoral α-smooth muscle actin (α-SMA, a cancer-associated fibroblast (CAF) marker), fibronectin (an extracellular matrix component produced by CAFs), collagen hydroxyproline, and fibrosis (Fig. 6b and Supplementary Fig. 12). Intratumoral desmoplasia can constrict vasculature, decreasing oxygenation and causing hypoxia[16]. Owing to the inhibition of fibrosis, P-BS-CM1 → P-CM2 substantially decreased intratumoral hypoxia inducible factor-1α (HIF-1α) expression versus other groups (Fig. 6c). (ii) With apparent E-cadherin upregulation and vimentin downregulation, P-BS-CM1 → P-CM2 increased epithelial properties and decreased mesenchymal features of tumor cells (Fig. 6d), indicating EMT inhibition. CXCR4-associated EMT-promoting

factors like matrix metalloproteinase-9 (MMP-9) and TGF-β were also maximally reduced by P-BS-CM1 → P-CM2 (Supplementary Fig. 13). (iii) Quantification of intratumoral, systemic, and pulmonary levels of LOX and TGF-β showed P-BS-CM1 → P-CM2 mediated significant reductions in these TDSFs in the metastasis cascade, disrupting "seed-soil" crosstalk between primary tumors and pre-metastatic lungs (Fig. 6e). (iv) Consequently, P-BS-CM1 → P-CM2 substantially regressed pulmonary PMN hallmarks like LOX accumulation (Supplementary Fig. 14), BMDC recruitment, S100A8 secretion (Fig. 6f), and E-cadherin expression (Supplementary Fig. 15) to levels close to healthy tumor-free mice, whereas free BS and P-BS had limited effects. (v) By normalizing the pulmonary microenvironment, P-BS-CM1 → P-CM2 drastically reduced CXCL12 in lungs, which correlated with inhibited BMDC recruitment (Fig. 6g). Collectively, P-BS-CM1 → P-CM2 significantly disrupted the CXCR4 downstream cascade (reshaping metastatic "seeds", intercepting "seed-soil" crosstalk, regressing PMN "soil"), creating a prerequisite for metastasis eradication.

Considering the key role of CXCR4 in tumor immunosuppression[16,17], we investigated the immunomodulatory effects of P-BS-CM1 → P-CM2 on primary tumors (Supplementary Fig. 16a). Flow cytometry of immune cells in primary tumor tissues showed that immunosuppressive regulatory T cells (Tregs) and myeloid-derived suppressor cells (MDSCs) were not affected by free BS, and moderately decreased by multivalent CXCR4-binding P-BS. However, Tregs and MDSCs were more effectively depleted by P-BS-CM1 → P-CM2, which had the repertoire to further escalate CXCR4 clustering. As Rakesh K. Jaina et al. reported, blocking CXCR4 by AMD3100 decreases immunosuppression in metastatic breast tumor largely through alleviating tumor desmoplasia and hypoxia[16]. In consistence, decreased collagen deposition (Fig. 6b) and intratumoral hypoxia (Fig. 6c) could be achieved by CXCR4 antagonism by P-BS-CM1 → P-CM2, whereas antagonizing CXCR4 by free BS or multivalent P-BS appeared to be less effective. Thus, the superior depletion of Tregs and MDSCs by P-BS-CM1 → P-CM2 can be attributed to the substantial disruption of reciprocal tumor desmoplasia-hypoxia pathways. Additional mechanisms not studied in this study are likely contributory as well. Moreover, P-BS-CM1 → P-CM2 also demonstrated better efficiency in decreasing the frequencies of intratumoral MDSCs and Tregs compared to AMD3100 (Supplementary Fig. 16b). However, tumor-infiltrating CD8 + T cells remained low for both AMD3100 and P-BS-CM1 → P-CM2, aligning with previous studies showing CXCR4 blockade alone is insufficient to activate anti-tumor immunity[12,22].

The above results demonstrated that beyond interception of the CXCR4-mediated "seed-soil" metastatic cascade to reduce spontaneous tumor metastasis, escalated CXCR4 clustering by P-BS-CM1 → P-CM2 removed physical barriers by alleviating desmoplasia and reduced immunological barriers by reversing immunosuppression. This paved the way for T cell penetration and tumor-reactive activity. However, P-BS-CM1 → P-CM2 alone could not actively recruit CD8+ T cells into the tumor microenvironment (Supplementary Fig. 16) or exert adequate therapeutic effect against primary tumor (Fig. 5e, f), highlighting the need for complementary T cell recruitment strategies.

## Potentiation of PDT

One solution may be inclusion of PDT that not only kill cancer cells directly but also induces ICD. Cancer cells succumbing to ICD release danger associated molecular patterns (DAMPs) that promote antigen-presenting cell engulfment, antigen presentation to T cells, and ultimate T cell recruitment[25]. To confer tumor-killing capability, we designed sequential delivery of P-BS-CM1 → P-PS-CM2 to integrate PDT into the netlike crosslinking system. Unmodified polymer-PS conjugate (P-PS) and BS-multivalently-modified polymer-PS conjugate (P-BS-PS) were also synthesized as controls (Supplementary Fig. 1 and Supplementary Table 1). As shown in Fig. 7a, CXCR4-targeted P-BS-PS displayed significantly higher affinity for 4T1 cells compared to

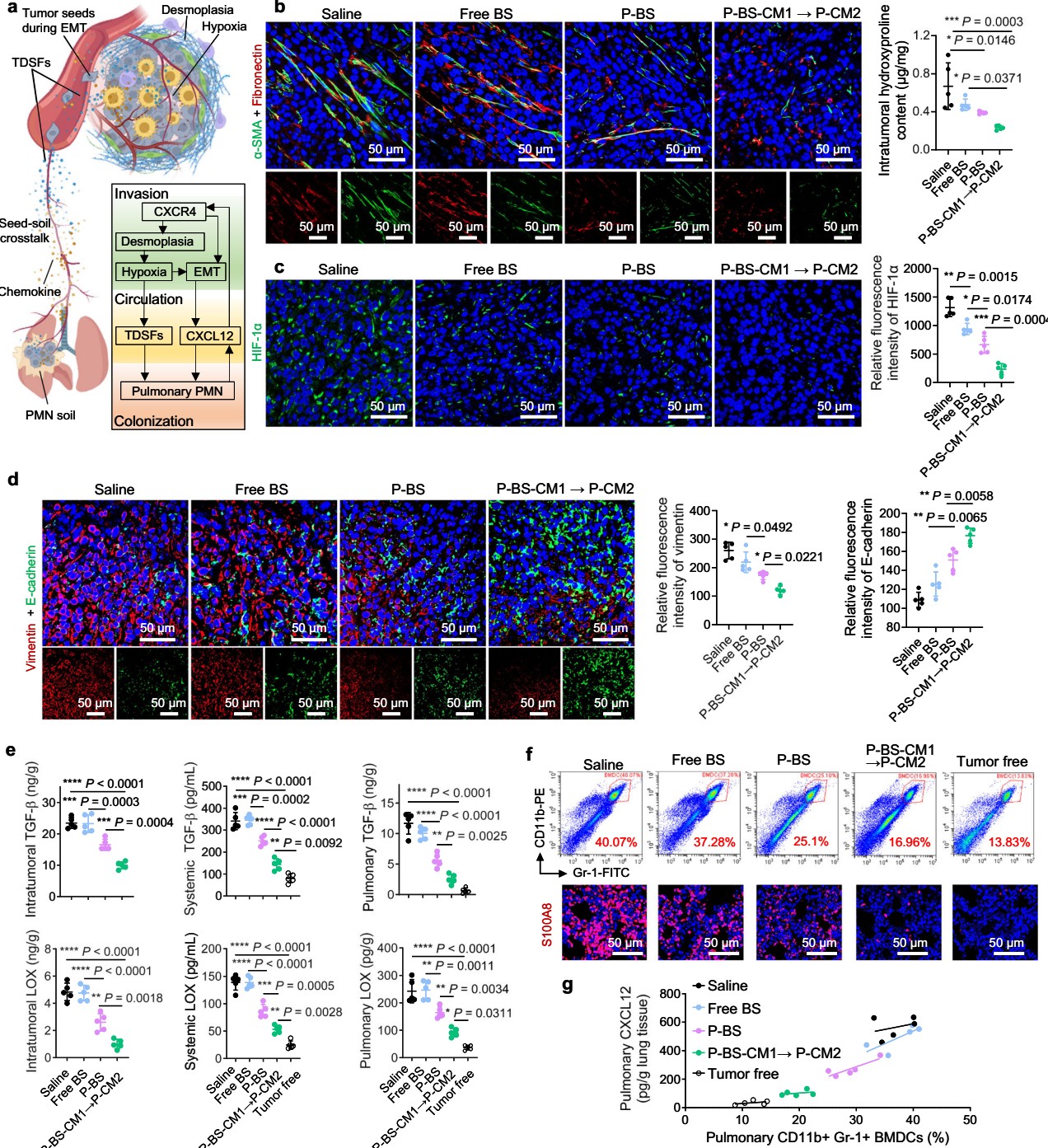

**Fig. 6 | Interference with spontaneous metastasis cascade to distant lung.**
Spontaneous lung metastasis mouse models of female BALB/c mice bearing orthotopic 4T1 tumors received three cycles of free BS, P-BS, or sequential P-BS-CM1→P-CM2 (24 h lag) on days 7, 14, and 21. Analyses occurred on day 28.
**a** Schematic illustration showing CXCR4-mediated metastasis cascade. EMT, epithelial−mesenchymal transition; TDSF, tumor-derived secreted factors; PMN, pre-metastatic niche. Created with BioRender.com. **b** Representative immunofluorescent images of α smooth muscle actin (α-SMA, green) and fibronectin (red), and quantification analysis of hydroxyproline (unique collagen amino acid) in tumor tissues. **c** Representative immunofluorescent images and quantitative flow cytometry analysis of intratumoral hypoxia-inducible factor 1α (HIF-1α) expression

(green). **d** Representative immunofluorescent images and quantitative flow cytometry analysis of EMT markers in primary tumors. Blue, nuclei; green, E-cadherin; red, vimentin. **e** Intra-tumoral, systemic, and pulmonary levels of TDSFs including lysyloxidase (LOX) and transforming growth factor-β (TGF-β). **f** Representative flow cytometry plots of pulmonary bone marrow derived cells (BMDCs, CD11b + Gr1 + ) and immunofluorescence of pulmonary S100A8 (red). Blue, nuclei. **g** Correlation between pulmonary BMDCs and Chemokine (C-X-C Motif) ligand 12 (CXCL12) after treatments. Scale bar, 50 µm. *n* = 5 mice per group (**b**–**g**). Data are presented as mean ± SD, with statistics determined by one-way ANOVA with Tukey's multiple comparisons without adjustments. *$P$ < 0.05, **$P$ < 0.01, ***$P$ < 0.001, ****$P$ < 0.0001. Source data are provided as a Source Data file.

non-targeted P-PS. Stepwise targeting by P-BS-CM1 → P-PS-CM2 also considerably increased cell binding of PS, albeit slightly lower than direct P-BS-PS anchoring to cell surface receptors. Consistently, P-PS barely interacted with 4T1 cells, while P-BS-PS distributed evenly around cells in an apparent ring pattern. Additionally, PS fluorescence from P-BS-CM1 → P-PS-CM2 drifted into punctate, aggregated distributions on the cell surface, attributable to the cell-surface 'patching' process via netlike crosslinking of CXCR4-anchored nanothreads that dragged receptor clustering (Fig. 7b). Under laser irradiation, P-BS-PS induced higher cytotoxicity and apoptosis than P-PS. Meanwhile, despite reduced PS binding, P-BS-CM1 → P-PS-CM2 exerted superior cytotoxic effects over P-BS-PS (Fig. 7c, d). This is because P-BS-CM1 → P-PS-CM2 escalated CXCR4 clustering and amplified interference with

downstream survival pathways (Fig. 4f), including phosphoinositide 3 kinase (PI3K) signaling (Fig. 7e and Supplementary Fig. 17), thereby increasing the cells' susceptibility to PDT. Consequently, with irradiation, PDT in the form of P-BS-CM1 → P-PS-CM2 triggered more potent ICD than P-PS or P-BS-PS, evidenced by greater calreticulin (CRT) exposure and adenosine triphosphate (ATP) release (Fig. 7f).

## Generation of local immune response against primary tumor

Having incorporated PDT into the stepwise CXCR4 clustering strategy, we assessed the in vivo therapeutic efficacy of sequential P-BS-CM1 → P-PS-CM2 under laser-off (−) and laser-on (+) conditions. Female mouse models bearing orthotopic 4T1 tumors with spontaneous lung metastases received four cycles of P-BS-CM1 → P-PS-CM2 with a 24 h lag

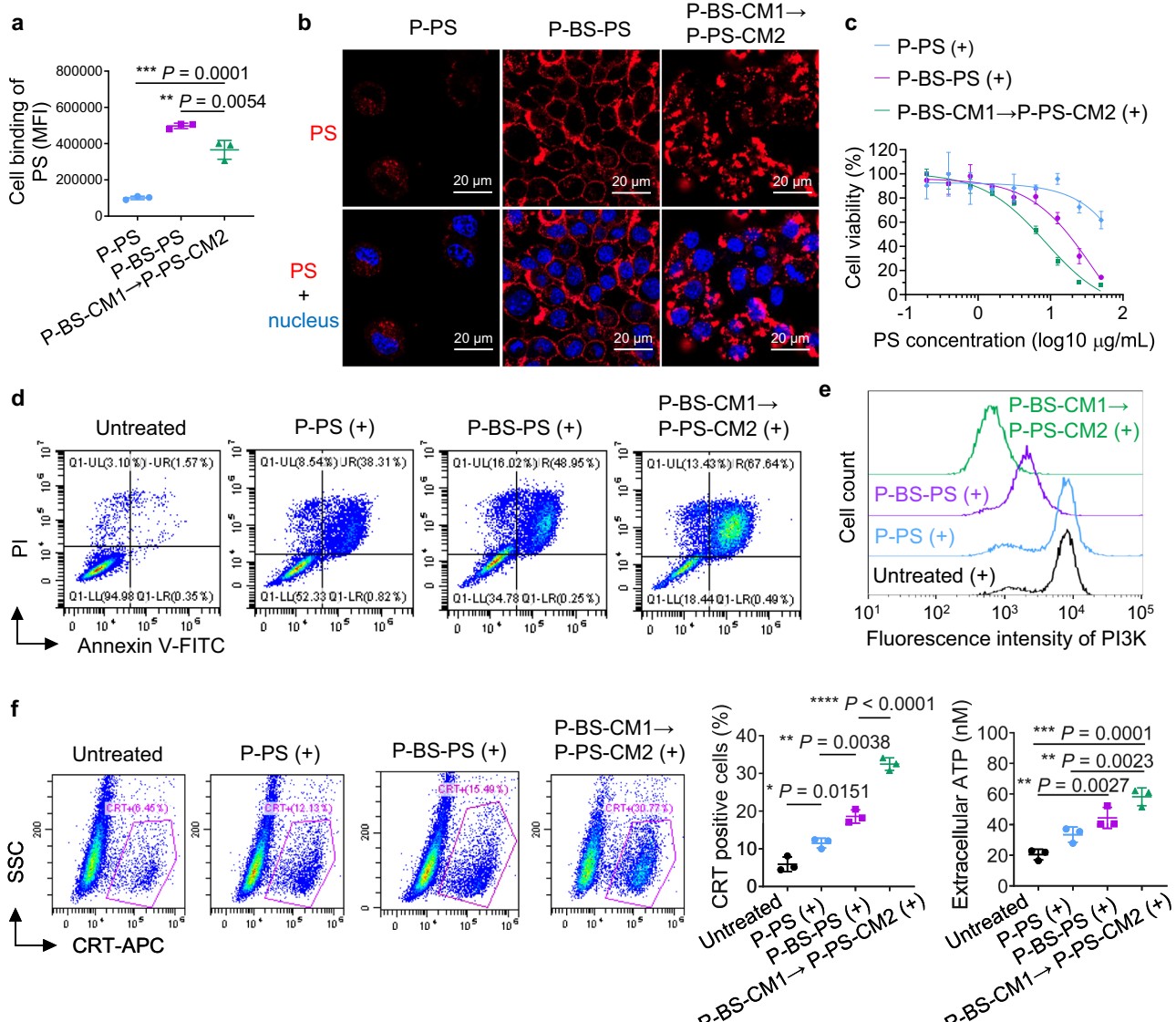

**Fig. 7 | Potentiation of PDT. a** Evaluation of photosensitizer (PS) binding in 4T1 cells after 1 h treatment with P-PS or P-BS-PS, or sequential treatment with P-BS-CM1 for 1 h followed by P-PS-CM2 for 1 h by flow cytometry analysis. **b** Representative confocal microscopy images of 4T1 cells treated with P-PS or P-BS-PS for 1 h, or consecutively treated with P-BS-CM1 for 1 h followed by P-PS-CM2 for 1 h. Blue indicates cell nuclei, red indicates PS fluorophores. Scale bars, 20 μm. **c–f** Evaluations in 4T1 cells after various treatments with laser irradiation (+), including: (**c**) Cell viability; (**d**), Apoptosis induction; (**e**), Intracellular expression of phosphatidylinositol 3-kinase (PI3K); (**f**), ICD hallmarks of calreticulin (CRT) exposure and extracellular adenosine triphosphate (ATP) release. For P-PS and P-BS-PS

groups, treatments were administered for 1 h followed by 5 min of laser irradiation and 24 h of further culture prior to analysis. For the P-BS-CM1 → P-PS-CM2 group, consecutive 1 h treatments with P-BS-CM1 and P-PS-CM2 were administered, followed by 5 min of laser irradiation and 24 h of culture prior to analysis. $n = 3$ biologically independent samples per group (**c, f**). All the experiments were repeated twice independently with similar results. Data are presented as mean ± SD. Statistics are calculated by one-way ANOVA with Tukey's multiple comparisons without adjustments. *$P < 0.05$, **$P < 0.01$, ***$P < 0.001$, ****$P < 0.0001$. Source data are provided as a Source Data file.

(Fig. 8a), followed by 24 h post-irradiation (Fig. 8b). Without irradiation, P-BS-CM1 → P-PS-CM2 (−) profoundly inhibited lung metastasis (~85% reduction) but did not inhibit tumor growth or improve survival (Fig. 8a, Supplementary Fig. 18). In contrast, P-BS-CM1 → P-PS-CM2 (+) showed superiority with irradiation. Compared to P-PS (+) and P-BS-PS (+), P-BS-CM1 → P-PS-CM2 (+) significantly suppressed tumor growth long-term, controlling tumor size to remain small, and extended survival (Fig. 8b, Supplementary Fig. 19). Notably, coordinated PDT and CXCR4 clustering by P-BS-CM1 → P-PS-CM2 (+) led to a significant reduction in BMDC recruitment and CXCL12 levels in the lungs

(Supplementary Fig. 20). This effect could be largely attributed to the ability of P-BS-CM1 → P-CM2 to intercept the crosstalk between the primary tumor and pulmonary PMN, resulting in normalization of lung microenvironment, as observed in Fig. 6. As a result of substantial pulmonary PMN regression, P-BS-CM1 → P-PS-CM2 (+) completely halted metastasis to the lungs, whereas other groups showed various degrees of metastatic nodules in pulmonary tissues (Fig. 8b). Analysis of post-treatment tumors revealed negligible ICD induction by P-BS-CM1 → P-PS-CM2 (−) (Fig. 8c). Conversely, P-BS-CM1 → P-PS-CM2 (+) generated substantial DAMPs like surface-exposed calreticulin,

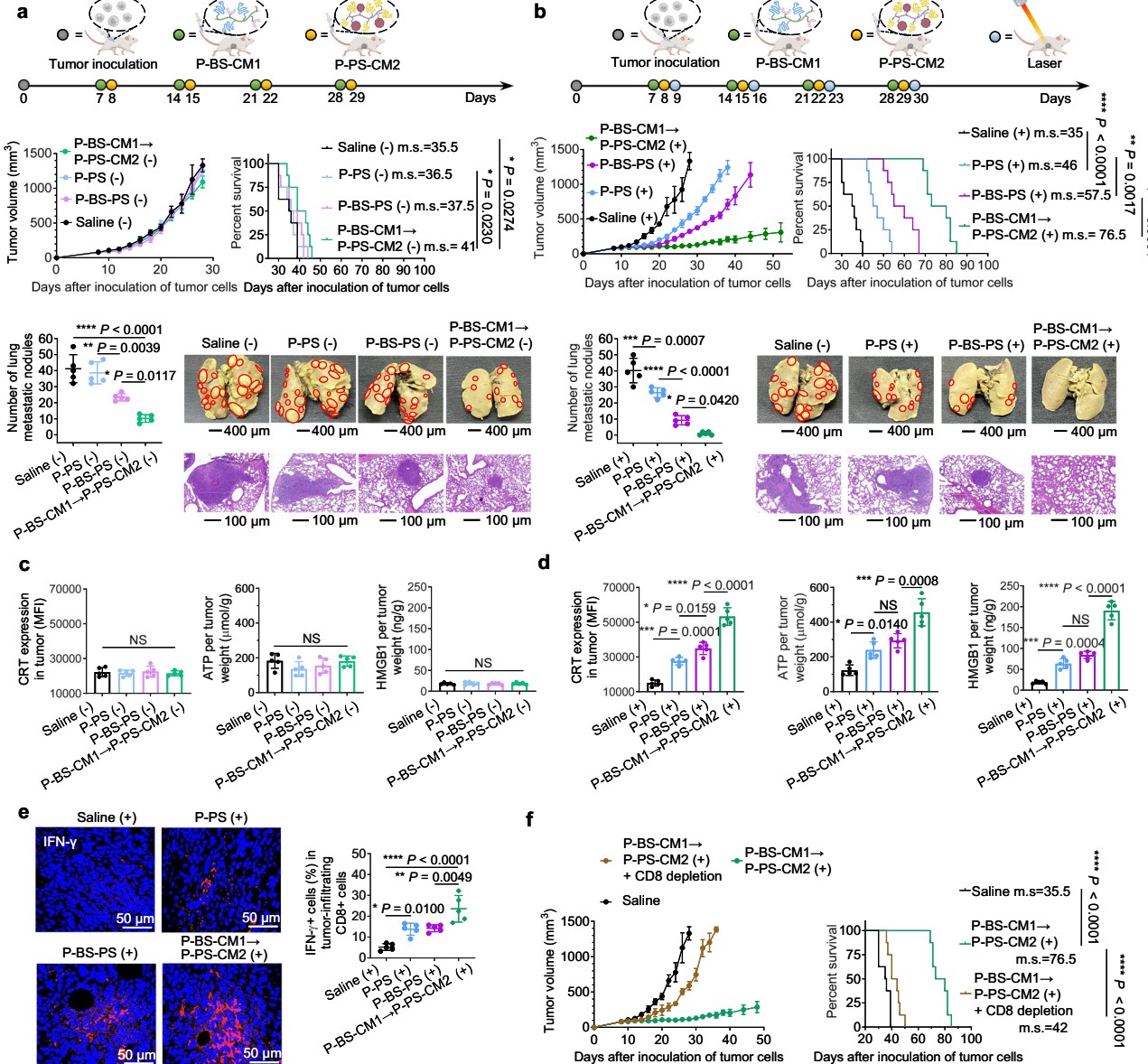

**Fig. 8 | Generation of local immune response against primary tumor.**
**a**, **b** Primary tumor growth, survival, and lung metastasis in spontaneous lung metastasis mouse models of female BALB/c mice orthotopically bearing murine 4T1 breast tumors. Timelines indicate schedules for four cycle treatments with P-PS, P-BS-PS or P-BS-CM1 → P-PS-CM2 without (**a**) or with (**b**) laser irradiation. Metastatic nodules were indicated with red circles. Scale bar was 400 μm for photographs of whole lungs (top) and was 100 μm for hematoxylin-eosin staining images (bottom). **c**, **d** Intratumoral level of calreticulin (CRT) exposure, adenosine triphosphate (ATP) secretion, and high mobility group protein B1 (HMGB1) release in primary tumor tissues on Day 28 after three cycle treatments without (**c**) and with (**d**) laser irradiation. **e** Representative immunofluorescent images of interferon-γ (IFN-γ) and

quantification of IFN-γ⁺CD8⁺ T lymphocytes in primary tumors on Day 28 after three cycle treatments with laser irradiation. Blue indicates cell nuclei. Red indicates IFN-γ. Scale bars, 50 μm. **f** Tumor growth and survival rate over time after cotreatment with P-BS-CCM1 → P-PS-CCM2 (+) and CD8-depleting antibodies. *n* = 8 animals per group for primary tumor growth and survival studies in (**a**, **b**, **f**). *n* = 5 animals per group for other studies. Data are presented as mean ± SD. Statistics for survival curves in (**a**, **b**, **f**) are calculated by log-rank (Mantel–Cox) test. Statistics of others are calculated by one-way ANOVA with Tukey's multiple comparisons without adjustments. NS not significant, *$P < 0.05$, **$P < 0.01$, *** $P < 0.001$, ****$P < 0.0001$. Source data are provided as a Source Data file.

extracellular ATP, and high mobility group box 1 (HMGB1). Despite potentially greater PS delivery, P-BS-PS (+) showed weaker ICD induction than P-BS-CM1→P-PS-CM2 (+) (Fig. 8d). This suggests that the orchestration of CXCR4 pre-targeting and clustering by P-BS-CM1→P-PS-CM2, which potentiated PDT by alleviating tumor hypoxia (Fig. 6c) and increasing susceptibility (Fig. 7), exceeded the tumor accumulation benefits conferred by multivalent CXCR4 targeting in the P-BS-PS group.

Consequently, P-BS-CM1→P-PS-CM2 (+) primed anti-cancer immunity in the local tumor microenvironment, partially through amplified ICD-inducing PDT recruiting CD8$^+$ T cells into the tumor bed, and partially through escalated CXCR4 clustering reducing immuno-suppressive Tregs and MDSCs (Supplementary Fig. 21). This further unleash the tumoricidal activity of tumor-infiltrating CD8$^+$ T cells, evidenced by the highest frequencies of tumor-reactive (IFN-γ$^+$) CD8$^+$ T cells in P-BS-CM1→P-PS-CM2 (+) treated tumors compared to the P-PS (+) and P-BS-PS (+) groups (Fig. 8e). To determine whether the enhanced immune response contributed to the anti-tumor effects, 4T1-tumor bearing mice were subjected to CD8$^+$ T cell ablation using CD8-depleting antibodies during P-BS-CM1→P-PS-CM2 (+) treatment. Concurrent CD8$^+$ T cell ablation markedly diminished the tumor regression and survival benefits of P-BS-CM1→P-PS-CM2 (+), validating generation of a localized immune response against the primary tumor (Fig. 8f and Supplementary Fig. 22).

### Generation of abscopal immunological memory against disseminated metastasis

Considering P-BS-CM1→P-PS-CM2 (+) effectively inhibited local tumor growth and spontaneous lung metastasis, we investigated whether it could also exert abscopal effects against disseminated metastatic tumor cells that had already extravasated into circulation. As scheduled (Fig. 9a), tumor-bearing mice receiving treatments were subsequently re-challenged intravenously with luciferase-expressing 4T1 cells. Immediately after the re-challenge, we first analyzed the pulmonary PMN where circulating tumor cells preferentially colonize (Fig. 9b). Compared to tumor-free mice (Control-1), orthotopic tumor-bearing mice (Control-2) showed lungs highly enriched in CD11b$^+$Gr1$^+$ BMDCs, evidencing primary tumor-fostered PMN formation ahead of metastasis. In comparison, P-BS-CM1→P-PS-CM2 (−) or (+) treated mice displayed reduced pulmonary BMDCs, because escalated CXCR4 clustering attenuated primary tumor influence on PMN and intercepted "seed-soil" crosstalk. One week post-rechallenge, bioluminescence imaging of luciferase-expressing 4T1 cells growing in mice were captured (Fig. 9c). Spontaneous lung metastasis occurred in Control-1. Due to PMN stimulation, 3/5 Control-2 mice died before imaging, and the remaining displayed extensive metastases. Mice treated with P-BS-CM1→P-PS-CM2 (−) exhibited reduced lung metastases compared to Control-2, implying disrupted PMN recruitment but incomplete metastasis prevention. Notably, P-BS-CM1→P-PS-CM2 (+) protected the re-challenged mice with almost no visible metastatic lesions. Since P-BS-CM1→P-PS-CM2 (+) could augment the local ICD of primary tumor and generate vaccine-like functions, this result potentially reflected the establishment of PDT-induced tumor-specific immunologic memory.

To assess anti-tumor memory, we measured splenic CD8$^+$CD44$^+$CD62L$^-$ effector memory T cells (CD8$^+$ Tems) that could elicit immediate protections by producing cytokines (Fig. 9d). Compare to Control-1, P-BS-CM1→P-PS-CM2 (−) did not increase CD8$^+$ Tems. However, P-BS-CM1→P-PS-CM2 (+) significantly expanded these populations. Meanwhile, coculture of peripheral blood mononuclear cells (PBMCs) isolated from P-BS-CM1→P-PS-CM2 (+) mice with live 4T1 cells also elicited higher frequency of tumor cell-reactive T cells (IFN-γ$^+$CD8$^+$), indicating durable immunity against the same tumor type is established (Fig. 9e). Importantly, these results clearly demonstrate the bidirectional functions of P-BS-CM1→P-PS-CM2 (+) in

rejecting disseminated tumor cells through: (1) escalation of CXCR4 clustering to minimize pro-metastatic influence of PMN, and (2) potentiation of ICD-inducing PDT to generate abscopal immunological memory against circulating tumor cells.

## Discussion

Numerous studies demonstrate CXCR4 antagonists can hinder breast cancer metastasis[8–12]. However, some antagonists ineffectively disable downstream signaling despite having high affinity for CXCR4[8]. In the present study, we have fixed this 'bug' by 'stitching patches' to connect an expanded cell-surface area of CXCR4 receptors, integrating them into a supercluster, and synchronizing transmembrane signaling that intercepts metastasis cascade. Specifically, our approach includes the following sequential steps (Fig. 1). (1) Nanothread-1 actively recognizes and multivalently binds CXCR4, stringing adjacent receptors while presenting decoys on cell surface. (2) Nanothread-2 engages these decoys and intertwines with Nanothread-1 into a coiled-coil network, clustering an expansive vicinity of CXCR4. (3) Photosensitizers 'hitchhike' on Nanothread-2 to the tumor site for targeted PDT, inducing ICD to generate an in-situ vaccine. Antagonism through nanothread 'patching'-induced CXCR4 superclusters profoundly amplifies the disruption of down-stream signal cascade, seals the impetus of breast cancer cells for spontaneous metastasis, and constrains cancer cells within the primary tumor. Concurrently, nanothread 'patching'-enabled CXCR4 antagonism consolidates survival pathway interference, hypoxia alleviation and immunosuppression reversal, which potentiates the tumor-localized PDT, thus initiating anti-tumor immune response against the primary tumor while also establishing an abscopal memory effect against disseminated metastasis.

Previous work by others and ourselves have reported the successful use of multivalent ligands, in situ polymerizable strategies, and retractable nano-springs based on polymers, albumins, and nanoparticles cluster various receptors (e.g., CD20[13,26–32], death receptor 5[33], programmed death-ligand 1[21,34]) for amplified efficacy. A typical example is that rituximab binding CD20 alone cannot directly induce apoptosis unless the CD20-bound antibodies are crosslinked by polymeric scaffold to form receptor condensate[27,28], with the cell killing potence positively correlating with the number of engaged receptors[15]. In this study, multivalent polymer-antagonists nanocontructs (P-BS) only moderately enhanced CXCR4 signaling disruption compared to free BS. Considering the nanoscale size, CXCR4 receptors in a wider vicinity are beyond the reach of P-BS, thus leaving small clusters individually scattered on cell surface, and leading to asynchronous generation of suboptimal mechanotransduction. We also show, the 'patching' strategy, through receptor stringing and nanothread assembling, overrides size constraints of conventional receptor-crosslinking strategies, and triggers enlarged CXCR4 clusters for augmented therapeutic benefits.

Collectively, during stepwise delivery of two interactable nanothreads (P-BS-CM1→P-CM2) that self-assemble into supramolecular network via coiled-coil interaction (Fig. 2), we show P-BS-CM1 forms a surface-bound ring pattern surrounding cancer cells through active recognition and multivalent stringing of adjacent CXCR4 (Fig. 3a), and, upon sequential netlike crosslinking, P-CM2 spatially reorganizes CXCR4-anchored Nanothread-1 from uniform distribution to condensed speckles patching on extensive cell-surface area (Fig. 3b). Compared to monovalent binding by free BS and multivalent binding by P-BS, cell surface 'patching' by P-BS-CM1→P-CM2 significantly perturbs the spatial organization of cell-surface CXCR4 and generates supercluster (Fig. 3c–f). Consequently, P-BS-CM1→P-CM2 profoundly interferes with downstream intracellular signal transduction related to metastasis, whereas the effects of free BS and P-BS are moderate (Fig. 4). With CXCR4 antagonism upon receptor clustering escalation, superior effect of P-BS-CM1→P-CM2 is also

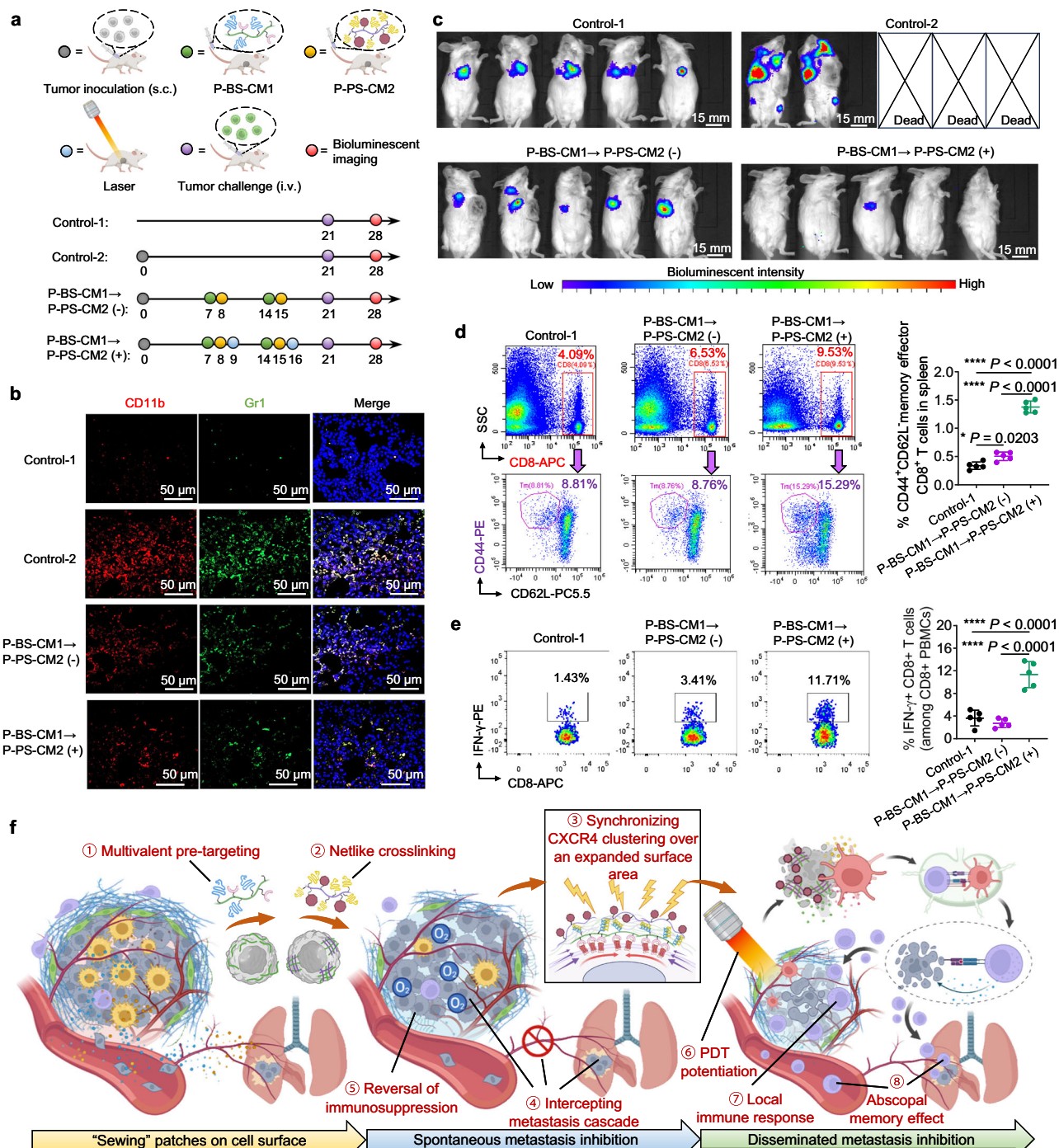

**Fig. 9 | Generation of abscopal immunological memory against disseminated metastasis. a** Schedules of disseminated 4T1 cell rechallenge experiment for Control-1, Control-2, P-BS-CM1 → P-PS-CM2 (−), and P-BS-CM1 → P-PS-CM2 (+) groups. **b** Representative immunofluorescence images of pulmonary CD11b⁺Gr1⁺ BMDCs immediately after rechallenge on day 21. Blue indicates cell nuclei, red indicates CD11b, green indicates Gr1. Scale bars, 50 μm. **c** Bioluminescent images of mice one week after rechallenge with luciferase-expressing 4T1 cells on day 28. **d** Flow cytometry gating strategies and quantification of CD8⁺ Tems (CD8⁺CD44⁺CD62L⁻) in the spleen on day 28. **e** Flow cytometry quantification and representative plots of the percentage of tumor-reactive T cells (IFN-γ⁺CD8⁺)

among PBMCs from mice on day 28 against live 4T1 cells in vitro. *n* = 5 animals per group (**b**−**e**). Data are presented as mean ± SD (**d**, **e**). Statistics are calculated by one-way ANOVA with Tukey's multiple comparisons without adjustments (**d**, **e**). *$P$ < 0.05, ****$P$ < 0.0001. **f** Illustration showing cell-surface 'patching', through netlike crosslinking of CXCR4-anchored nanothreads, synchronizes CXCR4 clustering over an expanded surface area, and augments mechanotransduction to intercept metatatsis cascade, reverse immunosuppression, and potentiate immune-activating PDT, together leading to simultaneous inhibition of primary tumor, spontaneous metastasis, and disseminated metastasis. Created with BioRender.com (**a**, **f**). Source data are provided as a Source Data file.

observed in its ability to disrupt the metastasis cascade by shaping the phenotype of metastatic tumor "seeds", interrupting the "seed-soil" crosstalk, and attenuating the influence of pro-metastatic PMN "soils" (Fig. 6). As a result, we show, both in vitro and in vivo, P-BS-CM1 → P-

CM2 exhibits significantly higher capability to inhibit breast cancer cell metastasis compared to free BS and P-BS (Figs. 4, 5). These results confirmed that CXCR4 antagonism mode impacts downstream signaling, with nanothread 'patching' that assemblies CXCR4 into

supercluster outperforming monovalent or multivalent binding in metastasis inhibition.

Our data also signify that with the coordination by P-BS-CM1 → P-PS-CM2 (+), efficacious CXCR4 antagonism well suits photo-immunotherapy in simultaneous inhibition of spontaneous and disseminated metastasis. We discover CXCR4 antagonism via the 'patching' strategy has the ability to promote PDT in two ways: enhancing the repertoire of PDT and increasing tumor's susceptibility to PDT. First, P-BS-CM1 → P-CM2 decreases intratumoral hypoxia (Fig. 6c), thus favoring oxygen-dependent PDT. Second, P-BS-CM1 → P-CM2 downregulates multiple survival pathways, sensitizing cancer cells to PDT (Fig. 7). To further eliminate disseminated tumor cells that have already extravasated into the circulation, we integrate PDT into the cell-surface "patching" strategy by designing P-BS-CM1 → P-PS-CM2 （+）. CXCR4 pretargeting by Nanothread-1 enables payloads like PS on Nanothread-2 to hitchhike to tumors (Fig. 5a–c). Meanwhile, CXCR4 antagonism via the "patching" removes physical barriers by alleviating desmoplasia (Fig. 6b) and reduces immunological barriers by reversing immunosuppression (Supplementary Fig. 16), which paves the way for T cell penetration and tumor-reactive activity. Consequently, under irradiation, tumor-localized PDT induces robust ICD in primary tumor, generates vaccine-like effects, recruits CD8$^+$ T cells to tumor bed, and activates local immune response against tumor growth (Fig. 8). Concomitantly, disseminated metastasis is also rejected by P-BS-CM1 → P-PS-CM2 (+) throught a two-pronged mechanism: regression of recruiting PMN, and establishment of systemic memory response against circulating tumor cells (Fig. 9a–e). Our data signify that with the coordination by P-BS-CM1 → P-PS-CM2 (+), efficacious CXCR4 antagonism suits photoimmunotherapy well in simultaneous inhibition of spontaneous and disseminated metastasis (Fig. 9f).

Despite these encouraging results, additional efforts remain imperative to address the limitations of its broad application in receptor antagonism. To enable efficient anchoring and stringing of targeted receptors, the polymer length and BS valency of Nanothread-1 must be optimized to mirror the spatial presentation of the cell surface receptor. To maximize receptor clustering, the polymer length, crosslinker valence, time-staggered regimen between Nanothread-1 and 2 require screening and adjustment. Although several hypotheses have been proposed[35–37], in many cases the molecular mechanisms of mechanotransduction from extracellular receptor clustering to intracellular signalling cascades remain elusive. Furthermore, the efficacy and potential side effects of cell surface 'patching' to spatially rearrange and antagonize CXCR4 warrant further investigation using transgenic or humanized animal models that closely recapitulate the development of breast cancer metastasis. Hopefully, our proof-of-concept work—cell surface 'patching' through stepwise actuation of receptor stringing and nanothread crosslinking—provides a flexible solution to reinvigorate some antagonists that currently suffer from poor translation of receptor binding into signal manipulation.

## Methods

### Animals and ethics statement
Female BALB/c mice (6–8 weeks, 20 ± 2 g) were provided by SPF Biotechnology Co., Ltd (Beijing, China). Mice were housed in a specific pathogen-free environment at 21 ± 1 °C and 60 ± 5% humidity, with a 12 h light-dark cycle. Mice were access to food and water free. All animal experiments were approved by the Institutional Animal Care and Ethics Committee of Sichuan University. All animal experiments were conducted in the Animal Laboratory of West China School of Pharmacy in Sichuan University (accreditation number: SYXK (Chuan) 2018-113). Female mice were chosen because the majority of metastatic breast cancers is seen in female patients. According to the guidelines of ethics committee, the maximal tumor size permitted was 1500 mm³. Mice were euthanized when the tumor burden exceeded this threshold.

## Materials
CXCR4 binding sequence (LGASWHRPDKCCLGYQKRPLPA) was synthesized by Apeptide Co. Ltd. (Shanghai, China). Coiled motif 1 (CM1, CYGGKVSALKEKVSALKEEVSANKEKVSALKEKVSALKE), and coiled motif 2 (CM2, CYGGEVSALEKEVSALEKKNSALEKEVSALEKEVSALEK) were synthesized by Chinapeptides Co. Ltd. (Shanghai, China). Chlorin e6 (Ce6) was purchased from Meilunbio Co., Ltd. (Dalian, China). N-(2-hydroxypropyl) methacrylamide (HPMA) and N-(3-aminopropyl) methacrylamide (APMA) was purchased from Bide Pharmatech Ltd. (Shanghai, China). SulfoCy3 SE and SulfoCy5 SE were provided by Beijing Fluorescence Biotechnology Co. Ltd. ADM3100 (Plerixafor, Cat No.: M1898) and CCK-8 (Cell Counting Kit, Cat No.: M4839) were purchased from AbMole (USA). D-fluorescein potassium salt (Cat No.: E011306) was provided by Energy Chemical. Annexin V-FITC/PI Apoptosis detection Kit (Cat No.: 40302ES60) and Fluo-4 AM (Cat No.: 40704ES50) were provided by Yeasen (Shanghai, China). Enhanced BCA Protein Assay Kit (Cat No.: P0009) and Enhanced ATP Assay Kit (Cat No.: S0027) were provided by Beyotime Biotechnology (Shanghai, China). 4',6-Diamidino-2-phenylindole dihydrochloride (DAPI, Cat No.: D8200), Red Blood Cell Lysis Buffer (Cat No.: R1010), Modified Hematoxylin-Eosin Stain Kit (Cat No.: G1121), Modified Masson's Trichrome Stain Kit (Cat No.: G1346), and Modified Sirius Red Stain Kit (No Picric Acid) (Cat No.: GG1472) were provided by Beijing Solarbio Science & Technology Co., Ltd. Recombinant Murine SDF-1α (CXCL12) (Cat No.: 250-20 A) was purchased from PEPROTECH inc. Goat anti-mouse IgG H&L Alexa Fluor® 488 (Cat No.: ab150113) and goat anti-rabbit IgG H&L Alexa Fluor® 647 (Cat No.: ab150079) were provided by Abcam. Anti-CD8α-APC (Cat No.: 100711; clone: 53-6.7), anti-IFN-γ-PE (Cat No.: 505808; clone: XMG1.2), anti-CD45-PerCP/Cy5.5 (Cat No.: 103132; clone: 30-F11), anti-CD44-PE (Cat No.: 163610; clone: QA19A43), anti-CD62L-Percp/Cy5.5 (Cat No.: 161210; clone: 30-F11), anti-CXCR2-PE (Cat No.: 149304; clone: SA004G4), anti-CXCR5-PE (Cat No.: 145504; clone: L138D7), anti-CXCR7-PE (Cat No.: 331104; clone: 8F11-M16) were purchased from Biolegend. Anti-CD16/32 (Cat No.: 553142; clone: 2.4G2), anti-Foxp3-PE (Cat No.: 560408; clone: MF23), transcription factor buffer set (Cat No.: 562574) were provided by BD Biosciences. Anti-Vimentin Recombinant Rabbit Monoclonal Antibody (Cat No.: ET1610-39; clone: SC60-05), Anti-Vimentin Mouse Monoclonal Antibody (Cat No.: EM0401; clone: D4-B11), Anti-PI 3 Kinase p85 alpha Recombinant Rabbit Monoclonal Antibody (Cat No.: ET1608-70; clone: SU04-07), Anti-Fibronectin Mouse Monoclonal Antibody (Cat No.: RT1224; clone: 3G4), Anti-alpha smooth muscle Actin Recombinant Rabbit Monoclonal Antibody (Cat No.: ET1607-53; clone: SY04-07), Anti-Calreticulin Recombinant Rabbit Monoclonal Antibody (Cat No.: ET1608-60; clone: SU37-03), Anti-LOX Recombinant Rabbit Monoclonal Antibody (Cat No.: ET1706-31; clone: JU30-23) Anti-TGF beta 1 Recombinant Rabbit Monoclonal Antibody (Cat No.: HA721143; clone: PD00-17), Anti-HIF1 alpha Rabbit Polyclonal Antibody (Cat No.: ER1802-41), and Anti-HIF-1 alpha Recombinant Rabbit Monoclonal Antibody (Cat No.: HA721997 clone: JE75-33) were provided by Hangzhou HuaAn Biotechnology Co., Ltd. (HUABIO). S100-A8 / MRP8 Rabbit pAb (Cat No.: bs-2696R) and MMP9 Rabbit pAb (Cat No.: bs-0397R) were provided by Bioss Co.,Ltd. FITC Anti-Mouse CD3 Antibody (Cat No.: E-AB-F1013C clone: 17A2), PerCP/Cyanine5.5 Anti-Mouse CD4 Antibody (Cat No.: E-AB-F1097J; clone: GK1.5), PE Anti-Mouse/Human CD11b Antibody (Cat No.: E-AB-F1081D; clone: M1/70), FITC Anti-Mouse Ly-6G/Ly-6C (Gr-1) Antibody (Cat No.: E-AB-F1120C; clone: RB6-8C5) PE Anti-Human CD184/CXCR4 Antibody (Cat No.: E-AB-F1157D; clone: 12G5), and PE Anti-Human CD194/CCR4 Antibody (E-AB-F1366D; clone: L291H4) were provided by Elabscience® Biotechnology Co., Ltd. CoraLite® Plus 647 Anti-Mouse CD324 (E-cadherin; Cat No.: CL647-65241; clone: DECMA-1) was provided by Proteintech Co., Ltd.

InvivoMab anti-mouse CD8α (Cat No.: BE0061; clone: 2.43) was provided by BioXCell. QuantiCyto® Human/Mouse/Rat HMGB1 ELISA Kit (Cat No.: EHRC01.96) was provided by Neobioscience Technology Co,Ltd. Mouse Lysyl oxidase homolog 2 (LOXL2) ELISA kit (Cat No.: CSB-EL013041MO) was provided by CUSABIO. Mouse CXCL12 ELISA Kit (Cat No.: RX201119M) was provided by Quanzhou Ruixin Biological Technology Co., LTD. Mouse TGF-β ELISA Kit (Cat No.: HB1328-Mu) was provided by Shanghai hengyuan biological technology co., LTD. Hydroxyproline Assay Kit (Cat No.: A030-2-1), Alanine aminotransferase Assay Kit (Cat No.: C009-2-1), and Aspartate aminotransferase Assay Kit (Cat No.: C010-2-1) were provided by Nanjing Jiancheng Bioengineer Institute (NJJCBIO). All other reagents were of analytical grade.

## Cell lines
4T1 (Cat No.: CL-0007) murine breast cancer cells were provided by Pricella Life Science&Technology Co., Ltd. Luciferase-expressing 4T1 (4T1-luc, Cat No.: YC-B004-Luc-P) were provided by Guangzhou Ubigene Biosciences Co., Ltd. 4T1 cells and 4T1-luc cells were cultured in RPMI-1640 medium supplemented with 10% v/v fetal bovine serum, 1% antibiotics (penicillin and streptomycin), and incubated at 37 °C humidified environment with 5% $CO_2$ supply.

## Synthesis and characterizations of Nanothread-1 and Nanothread-2
HPMA copolymer containing pendant amino groups (P-$NH_2$) was synthesized by reversible addition-fragmentation chain transfer (RAFT) copolymerization as previously reported[21–24]. Briefly, HPMA (1.40 mmol) and N-(3-aminopropyl) methacrylamide (APMA, 0.16 mmol) were dissolved with 1.1 mL deionized water. 4-cyanopentanoic acid dithiobenzoate as a chain transfer agent (0.57 mg in 10 μL methanol) and 2,2′-azobis[2-(2-dimidazolin-2-yl)propane] dihydrochloride as an initiator (0.19 mg in 10 μL methanol) were added. The solution was bubbled with argon in ice bath for 20 min, sealed and then reacted at 50 °C for 24 h. The copolymer was precipitated three times in acetone and diethyl ether to ensure removal of excess monomers. The dithiobenzoate end group was then removed using excess of 2,2′-Azobis(2,4 dimethyl) valeronitrile at 50 °C for 3 h. After dialysis and lyophilization, P-$NH_2$ was obtained. Afterwards, the pendent amino groups of P-$NH_2$ were converted to maleimides (P-Mal) by reacting with a heterobifunctional amino-thiol coupling agent, succinimidyl-4-(N-maleimidomethyl) cyclohexane-1- carboxylate (SMCC). P-$NH_2$ and SMCC were dissolved in N, N-dimethylformamide (DMF) adding moderate amount of N, N-diisopropylethylamine (DIPEA) (P-$NH_2$: SMCC: DIPEA = 1:1:3 mol%). After reaction at room temperature for 6 h, the solution was added into the mixture of acetone and diethyl ether. The HPMA copolymer containing pendant maleimide groups (P-Mal) was obtained, filtered, and dried under vacuum. The remaining amino group were subsequently reacted with SulfoCy5 SE or SulfoCy3 SE to obtain fluorescent-labeled copolymers.

Next, P-Mal and CXCR4 binding sequence (BS) (P-Mal: BS = 1: 4 mol%) were reacted in phosphate buffer saline solution (PBS, pH 7.0) under an argon atmosphere for 24 h. After dialysis and lyophilization, HPMA copolymer-BS conjugates (P-BS) were obtained. Then, P-BS and coil motif 1 (CM1) were dissolved in PBS (pH 7.0) under an argon atmosphere. After 24 h, the product (P-BS-CM1) was obtained after dialysis and lyophilization. Similarly, HPMA copolymer-CM2 conjugates (P-CM2) were synthesized using the same method as P-BS, except that the BS was replaced with CM2.

To synthesize HPMA copolymer conjugates containing the photosensitizer Ce6, Ce6, DCC, and NHS (Ce6: DCC: NHS = 1:1.2:1.2 mol%) were stirred in DMSO for 3 h. P-Mal was added to the solution (P-Mal: Ce6 = 1:4 mol%) and reacted at room temperature. After 24 h, the reaction solution was filtered to remove insoluble byproducts, and the filtrate was dialyzed against deionized water for 2 days.

To synthesize the HPMA copolymer conjugates containing photosensitizer Ce6, Ce6, DCC, and NHS (Ce6: DCC: NHS = 1: 1.2: 1.2 mol%) were stirred in DMSO for 3 h. P-Mal was added into the solution (P-Mal: Ce6 =1:4 mol%) and reacted at room temperature. After 24 h, the reaction solution was filtered to remove insoluble byproducts, and the filtrate was dialyzed against deionized water for 2 days. Subsequently, P-PS and CM2 or P-PS and BS were separately dissolved in PBS (pH 7.4) under an argon atmosphere. After 24 h, the products (P-PS-CM2 and P-BS-PS) were purified by dialysis and obtained after lyophilization.

The average molecular weight (Mw) and polydispersity (PDI) of the HPMA copolymer conjugates were determined by gel permeation chromatography (GPC) on an AKTA purifier equipment with a Superose 6 10/300 GL analytical column (GE Healthcare, USA) and a differential refraction detector (KNAUER, 2300, Germany). Hydrodynamic sizes and zeta potentials were determined using a Malvern Zetasizer Nano ZS90 equipment (Malvern Instruments Ltd, Malvern, UK). The contents of BS, CM1, and CM2 were determined using a bicinchoninic acid (BCA) protein assay. The PS concentration was determined by ultraviolet-visible (UV-Vis) spectroscopy at its characteristic absorbance peak (660 nm).

## Investigation of coiled-coil assembly between Nanothread-1 and Nanothread-2
To investigate the coiled-coil assembly, varying CM1-to-CM2 ratios (3:1, 1:1, and 1:3) of P-BS-CM1 and P-CM2 were mixed in 10 mM PBS (pH 7.4) at 25 °C for 10 min. The total polymer concentration was kept constant at 10 mg/mL. Circular dichroism spectra, size distribution, and rheological properties of P-BS-CM1, P-CM2, and their mixture were measured. The surface morphology of P-BS-CM1 and the mixture (P-BS-CM1 + P-CM2) was observed using scanning electron microscopy. Additionally, for the FRET assay, CM1-to-CM2 equimolar of P-BS-CM1-Cy5 and P-CM2-Cy3 were mixed in 10 mM PBS (pH 7.4) at 25 °C for 10 min. Fluorescent spectra of P-BS-CM1-Cy5, P-CM2-Cy3, and the mixture (P-BS-CM1-Cy5+P-CM2-Cy3) were recorded at the Cy3 excitation wavelength of 550 nm.

## Investigation of the specificity of BS to CXCR4
To validate the specificity of the BS (CXCR4-binding sequence) to CXCR4, a dose-dependent chemokine receptors occupation assay and a chemokine receptors competitive binding assay were conducted. In the dose-dependent chemokine receptors occupation assay, 4T1 cells were exposed to AMD3100 or free BS, with concentrations ranging from 0.1 nM to 1 mM equivalent, for 1 h. Unoccupied chemokine receptors (CXCR4, CXCR7, CXCR2, CXCR5, and CCR4) on cell surface were stained with PE anti-human CD184/CXCR4 Antibody (1:300 dilution), anti-CXCR7-PE (1:300 dilution), anti-CXCR2-PE (1:300 dilution), anti-CXCR5-PE (1:300 dilution), and PE anti-human CD194/CCR4 Antibody (1:300 dilution) at 4 °C for 1 h, prior to flow cytometry analysis. In the chemokine receptors competitive binding assay, 4T1 cells (5 × $10^5$ cells) were pre-blocked with PE anti-human CD184/CXCR4 Antibody (1:300 dilution), anti-CXCR7-PE (1:300 dilution), anti-CXCR2-PE (1:300 dilution), anti-CXCR5-PE (1:300 dilution), and PE anti-human CD194/CCR4 Antibody (1:300 dilution) at 4 °C for 1 h. Subsequently, these cells were treated with P-Cy5-BS-CM1-Cy5 (5 nM Cy5 equivalence) for 1 h. Following the treatments, cells were washed three times with PBS and subjected to flow cytometry analysis.

## Investigation of Nanothreads crosslinking on 4T1 tumor cell surface
For in vitro studies, 4T1 cells (2 × $10^5$ cells) were seeded on coverslips (NEST Biotechnology) for attachment. Subsequently, the cells were treated with i) Cy5-labeled P-CM1 (P-CM1-Cy5) for 1 h, followed by incubation in cell culture medium for an additional 1 h, ii) Cy5-labeled P-BS-CM1 (P-BS-CM1-Cy5) for 1 h, followed by incubation in cell culture medium for an additional 1 h, or iii) P-BS-CM1-Cy5 for 1 h, followed by

Cy3-labeled P-CM2 (P-CM2-Cy3) for an additional 1 h (5 nM Cy5 and 5 nM Cy3 equivalence), respectively. Following the treatments, cells were washed three times with PBS, stained with DAPI, and observed using a confocal microscope. For in vivo studies, 4T1 cells ($1 \times 10^6$) were injected into the third mammary fat pad of female BALB/c mice (6–8 weeks, $20 \pm 2$ g) ($n = 3$) on day 0 to establish orthotopic breast tumor models. For investigation of CXCR4 binding, mice on day 14 (tumor volume, ~200 mm³) were intravenously injected with P-CM1-Cy5 or P-BS-CM1-Cy5 (equivalent Cy5 dose, 5 nmol/mouse, $n = 3$). Whole-body living imaging at 24, 48, and 72 h post-administration as well as excised tumors at the end point, were captured using the IVIS optical imaging system (IVIS Lumina Series III, PerkinElmer, USA). For the analysis of CM1/CM2 biorecognition in the second step, 4T1 tumor-bearing mice (tumor volume, ~200 mm³) were intravenously injected with Cy5 labeled P-CM2 (P-CM2-Cy5) alone on day 15, or pre-injected P-BS-CM1 on day 14 followed by intravenous injection of P-CM2-Cy5 on day 15 (equivalent Cy5 dose, 5 nmol/mouse, $n = 3$). Whole-body living imaging at 24 and 48 h post-P-CM2-Cy5 administration, as well as excised tumors at the end point, were captured using the IVIS optical imaging system. For further investigation, mice (tumor volume, ~200 mm³) were intravenously injected with P-BS-CM1-Cy5 on day 14 and then P-CM2-Cy3 on day 15 (5 nmol of Cy5 and 5 nmol of Cy3 per mouse). One day later, tumor tissues were collected for frozen sectioning and further observation by confocal microscope.

### Cell transfection and investigation of CXCR4 clustering

Lipofectamine™ 3000 (Cat No.: L3000008, Thermo Fisher Scientific) was employed to transfect EGFP-tagged *Mus musculus* C-X-C motif chemokine receptor 4 (EGFP-CXCR4) plasmids (GENEWIZ) into 4T1 cells. Prior to transfection, 4T1 cells ($2 \times 10^5$ cells) were seeded on coverslips (NEST Biotechnology) and cultured until reaching 50 – 70% confluence. For the transfection procedure, 5 µg of EGFP-CXCR4 plasmid was mixed with 7.5 µL of Lipofectamine 3000 reagent and 10 µL of P3000 reagent in 250 µL Opti-MEM™ (Cat No.: 31985062, Thermo Fisher Scientific), followed by a 15-minute incubation. Subsequently, the DNA−lipid complex was added to the cells cultured in Opti-MEM™ and incubated for 1 day. The transfected cells were then either left untreated or treated with: i) free BS for 1 h followed by culture in fresh medium for 13 h, ii) P-BS for 1 h followed by culture in fresh medium for 13 h, or iii) P-BS-CM1 for 1 h followed by P-CM2 for an additional 1 h, then cultured in fresh medium for 12 h (1 mg/mL BS equivalence, CM1:CM2, 1:1 mol%). Following the respective treatments, cells were observed using a confocal microscope. To investigate the inhibition of calcium influx following CXCR4 clustering, 4T1 cells ($5 \times 10^5$ cells) were seeded in 12-well plates (NEST Biotechnology). For free BS and P-BS groups, 4T1 cells were treated with concentrations ranging from 0 to 1 mM BS equivalence for 1 h, followed by cultivation in fresh medium for 25 h. In the P-BS-CM1 → P-CM2 group, 4T1 cells underwent consecutive treatment with P-BS-CM1 (concentrations ranging from 0 to 1 mg/mL BS equivalence) for 1 h and P-CM2 (CM1:CM2, 1:1 mol%) for 1 h, followed by culture in fresh medium for 24 h. Subsequent to the treatments, intracellular calcium levels were assayed.

### Investigation of the influence of delivery sequences on tumor accumulation and pharmacokinetics

Three distinct delivery approaches were implemented in 4T1 tumor-bearing female BALB/c mice (6–8-week, tumor volume~200 mm³, $n = 5$ per group). i) Consecutive delivery: Nanothread-1 and nanothread-2 were sequentially administered via intravenously injection with a 24 h time interval. ii) Simultaneous delivery: Nanothread-1 and Nanothread-2 were concurrently administered through two tail veins of one mouse. iii) Post-assembly delivery: Nanothread-1 and Nanothread-2 were premixed before intravenous injection (5 nmol of Cy5 and 5 nmol of Cy3 per mouse). For the tumor accumulation study, whole-body living

imaging was conducted at 24, 48, and 72 h post-administration of Cy5-labeled Nanothread. Additionally, excised tumors were imaged at the end point using the IVIS optical imaging system. For the pharmacokinetics study, blood samples were extracted at predetermined time points post-administration of fluorescence-labeled Nanothread, $n = 5$ per group. The fluorescent intensity in blood samples was measured with a microplate reader at the wavelengths of Cy5 (Ex: 630 nm, Em: 670 nm) and Cy3 (Ex: 530 nm, Em: 570 nm). Pharmacokinetic parameters, including half-life ($T_{1/2z}$), area under curve (AUC), and mean residence time (MRT), were calculated using DAS 2.0 software.

### Investigation of Nanothread-1 binding, Nanothread-2 cross-linking, and downstream calcium influx interference

For in vitro studies, 4T1 cells ($2 \times 10^5$ cells/well) were seeded into 24-well plates (NEST Biotechnology) and subjected to the following treatments: i) P-CM1-Cy5 for 1 h followed by P-CM2-Cy3 for another 1 h, ii) P-BS-Cy5 for 1 h followed by P-CM2-Cy3 for another 1 h, or iii) P-BS-CM1-Cy5 for 1 h followed by P-CM2-Cy3 for another 1 h, respectively (5 nM Cy5 and 5 nM Cy3 equivalence). For Nanothread-1 binding investigation, cells were washed three times with PBS, and the fluorescent intensity was measured in the APC channel of flow cytometry (Ex: 538 nm, Em: 660 ± 20 nm), representing Cy5 labeled Nanothread-1. For Nanothread-2 crosslinking investigation, cells were washed, and the fluorescent intensity was measured on the PC5.5 channel (Ex: 488 nm, Em: 690 ± 50 nm), representing the FRET signal. For calcium influx interference investigation, cells were cultured in fresh medium for an additional 24 h and then subjected to intracellular calcium assay. For in vivo studies, 4T1 tumor-bearing female BALB/c mice (6–8-week, tumor volume~200 mm³, $n = 3$ per group) were intravenously injected with: i) P-CM1-Cy5→P-CM2-Cy3, ii) P-BS-Cy5→P-CM2-Cy3, iii) P-BS-CM1-Cy5→P-CM2-Cy3 with a time lag of 24 h (5 nmol of Cy5 and 5 nmol of Cy3 per mouse, $n = 5$). 24 h post-P-CM2-Cy3 administration, tumor cells were isolated from excised tumors for flow cytometry analysis, as describe above in in vitro studies.

### Investigation of Nanothreads biodistribution and biosafety

For biodistribution study, 4T1 tumor-bearing female BALB/c mice (6–8-week, tumor volume~200 mm³, $n = 3$ per group) were intravenously injected with either i) P-BS-CM1-Cy5 → P-CM2, or ii) P-BS-CM1 → P-CM2-Cy5 with a time interval of 24 h (equivalent Cy5 dose, 5 nmol/mouse). After 72 h post-injection of Cy5-labeled Nanothreads, tumors, hearts, livers, spleens, lungs, and kidneys were harvested and imaged using the IVIS optical imaging system. For biosafety study, female BALB/c mice (6–8-week, tumor volume ~200 mm³, $n = 4$ per group) were intravenously injected with P-BS-CM1 on day 7, 14, 21, and 28 post-tumor inoculation, followed by P-PS-CM2 on day 8, 15, 22, and 29 post-tumor inoculation (5 mg/kg BS equivalence, CM1:CM2 1:1 mol%, 2.5 mg/kg PS equivalence). On day 30, blood samples, hearts, livers, spleens, lungs, and kidneys were harvested. Serum biochemistry and hematological cell status in bloods were analyzed. Organ histological morphologies were assessed by hematoxylin-eosin staining.

### Investigation of metastasis inhibition

Wound healing assay and transwell migration and invasion assay were conducted to investigate in vitro metastasis inhibition. For the wound healing assay, 4T1 cells ($5 \times 10^5$ cells) were cultured in 12-well plates (NEST Biotechnology) as monolayer with more than 90% coverage. Sterile 200 µL pipette tips were used to scrape off the cells, generating a linear cell-free area in the middle of the well. Subsequently, cells were washed by PBS to remove debris. In the stimulated group, 4T1 cells were stimulated with CXCL12 (100 ng/mL) for 1 h. For the free BS and P-BS groups, 4T1 cells received the treatments for 1 h prior to CXCL12 stimulation. In the P-BS-CM1 → P-CM2 group, 4T1 cells underwent consecutive treatment with P-BS-CM1 for 1 h and P-CM2 for 1 h

before CXCL12 stimulation (1 mg/mL BS equivalence, CM1:CM2, 1:1 mol %). Images of the wound were recorded by a microscope at 0 h and 24 h after CXCL12 stimulation, and the wound healing rate was calculated using Image J software. For the transwell migration and invasion assay, 4T1 cells ($5 \times 10^5$ cells) were seeded in the non-coated upper chamber (Millipore, 8 μm) or GelNest™ matrigel (NEST Biotechnology) pre-coated upper chamber. The cells underwent the same treatment conditions as wound healing assay. After 24 h of treatment, migrated or invaded cells across the membrane were stained with 0.1% crystal violet, imaged by microscope, and quantified by absorbance at 570 nm after dissolution.

For proteomics analysis, 4T1 cells ($5 \times 10^5$ cells) were seeded in 12-well plates (NEST Biotechnology). Subsequently, cells were either left untreated or treated as follows: i) with free BS for 1 h followed by culture in fresh medium for 25 h, ii) with P-BS for 1 h followed by culture in fresh medium for 25 h, or iii) with P-BS-CM1 for 1 h followed by P-CM2 for another 1 h, and then cultured in fresh medium for 24 h (1 mg/mL BS equivalence, CM1:CM2, 1:1 mol%). Cell samples were submitted for label-free proteomics analysis (Shanghai Omics-space Biotech Co., Ltd., Shanghai, China). The analysis included protein extraction, protein digestion, LC − MS/MS detection (Thermo Scientific Orbitrap-Fusion Lumos), protein quantitation and identification (MaxQuant 1.5.5.1), and bioinformatics analysis.

For the in vivo investigation of pulmonary metastasis, 4T1 cells ($1 \times 10^6$ cells) were injected into the third mammary fat pad of female BALB/c mice (6–8 weeks). When tumor volumes reached nearly 100 cm³ on day 7 after tumor inoculation, mice were randomly divided into 4 groups ($n = 5$ per group) and intravenously injected with the following samples: (1) saline on day 7, 14, and 21, (2) free BS on day 7, 14, and 21, (3) P-BS on day 7, 14, and 21, or (4) P-BS-CM1 on day 7, 14, and 21, followed by P-CM2 on day 8, 15, and 22 (5 mg/kg BS equivalence, CM1:CM2 1:1 mol%), respectively. Tumor volumes were recorded every other day. On day 28, lungs were harvested and fixed with Bouin's solutions. The pulmonary metastatic nodules were counted and analyzed by hematoxylin-eosin staining. In another experiment, the anti-metastasis effect of P-BS-CM1 → P-CM2 therapy was compared with daily treatment of AMD3100 at 2 mg/kg and 5 mg/kg via *i.p.* injection from day 7 to day 28 ($n = 5$).

### Investigation of CXCR4 associated metastasis cascade

4T1 cells ($1 \times 10^6$ cells) were injected into the third mammary fat pad of female BALB/c mice (6–8-week). Upon reaching tumor volumes of nearly 100 cm³ on day 7 post-tumor inoculation, mice were randomly divided into saline, free BS, P-BS, and P-BS-CM1 → P-CM2 groups ($n = 5$ per group), with the treatment regimens as described above. On day 28, blood samples, lungs and tumors were harvested.

To assess fibrosis in tumor tissues, Masson staining, Sirius staining, immunofluorescent staining of α-SMA and fibronectin, and a quantitative hydroxyproline assay were employed. For immunofluorescent staining of α-SMA and fibronectin, tumor slices were stained with anti-alpha smooth muscle actin recombinant rabbit monoclonal antibody (1:500 dilution) and anti-fibronectin mouse monoclonal antibody (1:500 dilution) at 4 °C overnight, followed by staining with goat anti-mouse IgG H&L Alexa Fluor® 488 (1:1000 dilution) and goat anti-rabbit IgG H&L Alexa Fluor® 647 (1:1000 dilution).

For evaluating the hypoxic conditions in tumor tissues, immunofluorescent staining and flow cytometry analysis of HIF-1α were conducted. For immunofluorescent staining of HIF-1α, tumor slices were sequentially stained with anti-HIF-1 alpha recombinant rabbit monoclonal antibody (1:500 dilution) and goat anti-rabbit IgG H&L Alexa Fluor® 647 (1:1000 dilution). For flow cytometry analysis of HIF-1α, single-cell suspensions of tumor tissues were sequentially stained with anti-HIF1 alpha rabbit polyclonal antibody (1:500 dilution) and goat anti-rabbit IgG H&L Alexa Fluor® 647 (1:1000 dilution) before analysis.

To investigate the epithelial-mesenchymal transition process in tumor tissues, immunofluorescent staining of E-cadherin, vimentin, MMP-9, and TGF-β, along with flow cytometry analysis of E-cadherin and vimentin, were applied. For immunofluorescent staining of E-cadherin and vimentin, tumor slices were stained with CoraLite® Plus 647 anti-mouse CD324 (E-cadherin, 1:300 dilution) and anti-vimentin mouse monoclonal antibody (1:500 dilution) at 4 °C overnight, followed by staining with goat anti-mouse IgG H&L Alexa Fluor® 488 (1:1000 dilution). For flow cytometry analysis of E-cadherin and vimentin, single-cell suspensions of tumor tissues were stained with CoraLite® Plus 647 anti-mouse CD324 (E-cadherin, 1:300 dilution) or sequentially stained with anti-vimentin recombinant rabbit monoclonal antibody and goat anti-rabbit IgG H&L Alexa Fluor® 647 (1:1000 dilution), respectively. For immunofluorescent staining of MMP-9 and TGF-β, tumor slices were separately stained with MMP9 Rabbit pAb (1:500 dilution) or anti-TGF beta 1 recombinant rabbit monoclonal antibody (1:500 dilution), followed by staining with goat anti-rabbit IgG H&L Alexa Fluor® 647 (1:1000 dilution), respectively.

To explore the "seed-soil" crosstalk, the serum, intra-tumoral, and pulmonary concentration of LOX was determined using enzyme-linked immunosorbent assay kits (CSB-EL013041MO, CUSABIO, https://www.cusabio.com/) according to the manufacturer's instructions. The serum, intra-tumoral, and pulmonary concentration of TGF-β was determined using mouse TGF-β ELISA Kit (Cat No.: HB1328-Mu). The formation of PMN was investigated through lung slices subjected to immunofluorescent staining of LOX, E-cadherin, and S100A8. For immunofluorescent staining of E-cadherin, Lung slices were stained with CoraLite® Plus 647 anti-mouse CD324 (E-cadherin, 1:300 dilution). For immunofluorescent staining of LOX and S100A8, Lung slices were separately stained with anti-LOX recombinant rabbit monoclonal antibody (1:500 dilution) or S100-A8 / MRP8 rabbit pAb (1:500 dilution), followed by staining with goat anti-rabbit IgG H&L Alexa Fluor® 647 (1:1000 dilution), respectively. Single-cell suspensions of lungs were analyzed using flow cytometry to assess BMDC (CD11b⁺Gr1⁺) with PE anti-mouse/human CD11b antibody (1:300 dilution) and FITC anti-mouse Ly-6G/Ly-6C (Gr-1) antibody (1:300 dilution). Chemokine concentrations in lungs were determined using a mouse CXCL12 ELISA kit.

### Investigation of PDT potentiation

For cell binding investigation, 4T1 cells ($4 \times 10^5$ cells/well) were seeded onto 12-well plates (NEST Biotechnology), and treated with i) P-PS for 1 h, ii) P-BS-PS for 1 h, or iii) P-BS-CM1 for 1 h followed by P-PS-CM2 for another 1 h (50 μg/mL BS equivalence, CM1:CM2 1:1 mol%, 25 μg/mL PS equivalence), respectively. Subsequently, cells were washed three times with PBS and observed using flow cytometry and confocal microscopy.

Cytotoxicity of PDT was investigated by CCK-8 cell viability assay and Annexin V-FITC/PI double staining cell apoptosis assay. For the CCK-8 assay, 4T1 cells ($5 \times 10^3$ cells/well) were seeded in the 96-well plate (NEST Biotechnology), and treated with: i) P-PS for 1 h, ii) P-BS-PS for 1 h, or iii) P-BS-CM1 for 1 h followed by P-PS-CM2 for another 1 h (concentrations ranging from 0−50 μg/mL PS equivalence, CM1:CM2 1:1 mol%), respectively. After treatment, cells were irradiated with a 660 nm laser for 5 min (280 mW/cm²) and then incubated for an additional 24 h. Subsequently, CCK-8 solution (1x) was added, and cell viability was calculated based on the absorbance at 450 nm.

For the Annexin V-FITC/PI double staining cell apoptosis assay, 4T1 cells ($2 \times 10^5$ cells/well) were seeded into 24-well plates (NEST Biotechnology) and treated with: i) P-PS for 1 h, ii) P-BS-PS for 1 h, or iii) P-BS-CM1 for 1 h followed by P-PS-CM2 for another 1 h (25 μg/mL BS equivalence, CM1:CM2 1:1 mol%, 12.5 μg/mL PS equivalence), respectively. After treatment, cells were irradiated with a 660 nm laser for 5 min (280 mW/cm²) and then incubated for an additional 24 h. Cells were stained with FITC Annexin-V and PI, and analyzed using flow cytometry.

For the investigation of CXCR4 downstream PI3K signals, cells were fixed, permeabilized, and stained with anti-PI 3 kinase p85 alpha recombinant rabbit monoclonal antibody (1:500 dilution, 4 °C, 1 h) and goat anti-rabbit IgG H&L Alexa Fluor® 647 (1:1000 dilution, 4 °C, 30 min), followed by flow cytometry analysis. For the investigation of ICD hallmarks, cells were stained with anti-calreticulin recombinant rabbit monoclonal antibody (1:500 dilution, 4 °C, 1 h) and goat anti-rabbit IgG H&L Alexa Fluor® 647 (1:1000 dilution, 4 °C, 40 min), followed by flow cytometry analysis. ATP levels in the cell culture supernatant were measured following the protocol of luciferase-based ATP assay kit.

### Investigation of local immune response against primary tumor growth and metastasis

4T1 cells ($1 \times 10^6$ cells) were injected into the third mammary fat pad of female BALB/c mice (6–8-week). When tumor volumes reached nearly 100 cm$^3$ on day 7 after tumor inoculation, mice were randomly divided into 8 groups ($n = 8$ per group). For four groups without laser irradiation (−), mice were intravenously injected with the following samples: (1) saline on day 7, 14, 21, and 28, (2) P-PS on day 7, 14, 21, and 28, (3) P-BS-PS on day 7, 14, 21, and 28, or (4) P-BS-CM1 on day 7, 14, 21, and 28, followed by P-PS-CM2 on day 8, 15, 22, and 29 (5 mg/kg BS equivalence, CM1:CM2 1:1 mol%, 2.5 mg/kg PS equivalence), respectively. For four groups with laser irradiation (+), mice were treated with the same regimens and irradiated at 650 nm laser (580 mW/cm$^2$, 5 min) on day 9, 16, 23 and 30. Tumor volumes and mice survivals were recorded every other day. For the investigation of immune response and lung metastasis, 4T1 tumor-bearing female BALB/c mice (6–8-week, tumor volume-100 mm$^3$) were randomly divided into 8 groups ($n = 5$ per group) and given three rounds of above treatments. On day 28, lungs and tumors were harvested. lungs were fixed with Bouin's solution. The pulmonary metastatic nodules were counted and analyzed by hematoxylin-eosin staining. Single-cell suspensions of tumor tissues were stained with anti-CD45-PerCP/Cy5.5 (1:300 dilution) and anti-calreticulin recombinant rabbit monoclonal antibody (1:500 dilution) at 4 °C for 1 h, and then stained with goat anti-rabbit IgG H&L Alexa Fluor® 647 (1:1000 dilution, 4 °C, 45 min), followed by flow cytometry analysis. ATP and HMGB1 concentrations in the supernatant of cell suspension were measured using a luciferase-based ATP assay kit and a mouse HMGB1 ELISA kit, respectively. Single-cell suspensions of tumors were stained with FITC anti-mouse CD3 antibody (1:300 dilution), PerCP/Cyanine5.5 anti-mouse CD4 antibody (1:300 dilution), anti-CD8α-APC (1:300 dilution), and anti-Foxp3-PE (1:300 dilution) for flow cytometry analysis of T cell subtypes (CD8$^+$ T cells, CD4$^+$ effector T cells, and Foxp3$^+$ regulatory T cells), and stained with PE anti-mouse/human CD11b antibody (1:300 dilution) and FITC anti-mouse Ly-6G/Ly-6C (Gr-1) antibody (1:300 dilution) for flow cytometry analysis of MDSCs (CD11b$^+$Gr1$^+$). Cytotoxic T cells in tumor tissues were investigated with immunofluorescent staining of IFN-γ and flow cytometry analysis using anti-IFN-γ-PE (1:300 dilution). For CD8-depletion assay, 4T1 tumor-bearing female BALB/c mice (6–8-week, tumor volume-100 mm$^3$) were randomly divided into 3 groups ($n = 8$ per group). Mice were intravenously given four doses of P-BS-CM1 on day 7, 14, 21 and 28, combined with P-PS-CM2 on day 8, 15, 22 and 29 (5 mg/kg BS equivalence, CM1:CM2 1:1 mol%, 2.5 mg/kg PS equivalence) in the presence or absence of InvivoMab anti-mouse CD8α (100 μg/mice). Mice were irradiated at 650 nm laser (580 mW/cm$^2$, 5 min) on day 9, 16, 23 and 30. Tumor volumes and mice survivals were recorded every other day.

### Investigation of abscopal immune memory against disseminated tumor metastasis

4T1 tumor-bearing female BALB/c mice (6–8-week, tumor volume-100 mm$^3$) were randomly divided into 3 groups ($n = 5$ per group) and treated with: (1) saline on day 7 and 14, (2) P-BS-CM1 + P-PS-CM2 without laser irradiation (−) on day 7-8 and 14-15, or (3) P-BS-CM1 + P-PS-CM2 with laser irradiation (+) on day 7–9 and 14–16, as indicated above. On day 21, mice were intravenously injected with 4T1–luc cells ($4 \times 10^5$ cells). Tumor-free mice were also intravenously injected with 4T1-luc cells ($4 \times 10^5$ cells) to function as controls. Disseminated tumor metastatic niches were captured on day 28 using the IVIS optical imaging system. Lungs, spleens, and bloods of these mice were collected on day 28. Lung slices underwent immunofluorescent staining of CD11b and Gr1. Single-cell suspensions of spleen tissues were subjected to flow cytometry analysis of CD8$^+$ Tems (CD8$^+$CD44$^+$CD62L$^-$) using anti-CD8α-APC (1:300 dilution), anti-CD44-PE (1:300 dilution), and anti-CD62L-Percp/Cy5.5 (1:300 dilution). Peripheral blood mononuclear cells (PBMCs) were isolated from bloods of these mice and cultured in T cell medium. A total of $5 \times 10^5$ PBMCs were incubated with $1 \times 10^5$ living 4T1 cells for 16 h in the presence of brefeldin A. Then, IFN-γ expression in PBMCs was analyzed with flow cytometry using anti-CD8α-APC (1:300 dilution) and anti-IFN-γ-PE (1:300 dilution).

### Statistical analysis

Statistical data was analyzed using Graphpad Prism 8 software and presented as mean ± SD. For two-group comparison, statistical significance was determined using Student's two-sided $t$-test. For multiple comparison, statistical significance was determined using one-way ANOVA with Tukey's multiple comparisons without adjustments. A significant difference was considered when the $P$ value was less than 0.05.

### Reporting summary

Further information on research design is available in the Nature Portfolio Reporting Summary linked to this article.

## Data availability

The mass spectrometry proteomics data used in this study are available in the ProteomeXchange partner repository with the dataset identifier PXD050511. The remaining data are available within the Article, Supplementary Information of Source Data file. Source data are provided with this paper.

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

## Acknowledgements

This work was supported by the National Natural Science Foundation of China (82104103 to Li, L.), and Science & Technology Department of Sichuan Province (2022NSFSC1291 to Li, L.).

## Author contributions

Zhou, M.L. and Li, L. designed the research. Zhou, M.L., Liu, C.D., Li, B., Li, J.L., and Zhang, P. performed the experiments and collected the data. Liu, C.D., Huang, Y., and Li, L. contributed to writing the manuscript, discussing the results and implications, and editing the manuscript at all stages.

## Competing interests

The authors declare no competing interests.
