## [Peer Review File · Nature Communications]

Cell Surface Patching via CXCR4-targeted Nanothreads for Cancer Metastasis InhibitionREVIEWER COMMENTS

Reviewer #1 (Remarks to the Author):

In this work, the authors developed a strategy that the nanowire supramolecular network patch rearranged cell-surface receptors to improve therapeutic outcomes. In antagonizing chemokine receptors on cancer cells, the patch blocked the metastasis cascade, reversed immunosuppression, and enhanced photodynamic immunotherapy to eliminate spontaneous and diffuse metastasis. However, there were some key experiments that should be added. Thus, I recommend a major revision.

1. The traction of the aggregation receptors produced by nanothread-1 and nanothread-2 should be characterized.
2. In order to prove that the equimolar mixture CM1-to-CM2 was optimal, the signal of coiled-coil formation, supramolecular size, rheological properties and the morphology of the material coil should be tested.
3. The effects of simultaneous delivery and post-assembly delivery of both nanothread-1 and nanothread-2 should be investigated.
4. In the immunofluorescent images of Fig. 4a and 4b, the expression localization of α -SMA, hypoxia and HIF-1 α in cells was not evident, and the experimental data should be checked and re-provided. Quantitative data for Fig. 4b and 4c should be provided.
5. The mechanism of P-BS-CM1 \rightarrow P-CM2 more efficiently depleting Tregs and MDSCs should be further investigated.
6. In animal experiments with P-BS-CM1 \rightarrow P-PS-CM2 on the enhancement of immunity by PDT, the amount of CXCL12 in the lungs should be provided.
7. In animal experiments, the effect of P-BS-CM1 \rightarrow P-CM2 and P-BS-CM1 \rightarrow P-PS-CM2 on blocking the metastasis cascade after lung stimulation of CXCL12 should be investigated.
8. The scale bar should be added in Fig. 3a, 3b, 3i, 4a, 5a, 7a and 7b. Please carefully check the scale bar in the figures.
9. The text description should be added to the diagram of Fig. 2e.
10. In Fig. 4a, the locally enlarged area should be in the same size.

Reviewer #2 (Remarks to the Author):

In this manuscript, the authors introduced a strategy for hierarchical CXCR4 clustering using two interactable polymer nanothreads. The process involves sequential delivery of Nanothread-1 and Nanothread-2 to assemble a netlike patch on the cell surface, integrating and clustering receptors over an expanded area. By targeting CXCR4, a receptor known for its role in tumor metastasis, and the authors propose that this approach holds great potential in inhibiting tumor metastasis. Additionally, the incorporation of photosensitizers on the nanothread-2 could induced anti-tumor immune responses. Overall, although this study presents an approach with the potential to significantly inhibit the growth and metastasis of primary tumor, the novelty of this work is lacking because they have previously introduced similar concepts. In addition, the following issues should be addressed.

1. The introductory section lacks logical coherence. Additionally, is CXCR4 clustering necessary for better inhibition of tumor cell metastasis? Has the underlying mechanism been confirmed? This forms the theoretical basis for the author's invention of this strategy.
2. Multiple peptide segments have been modified on the Nanothread to target CXCR4 and induce receptor clustering. Multivalent exogenous peptides often exhibit higher immunogenicity, and repeated administration may lead to the production of antibodies against the peptides, potentially inducing immune clearance of the Nanothread. The author should investigate this issue.
3. CXCR4 is a G-protein coupled receptor with seven transmembrane domains, highly expressed in various cells, including lymphocytes, endothelial cells, epithelial cells, hematopoietic stem cells, stromal fibroblasts, and cancer cells. The author should provide a more comprehensive presentation of the biological distribution of the Nanothread to assess its safety, rather than just showing its enrichment in tumors as in Fig 3a.
4. Figure 3C illustrates that Nanothread-1 and Nanothread-2 bind to the surface of tumor cells, but it cannot be concluded that they form the expected structure leading to receptor clustering. Tissue-level FRET efficiency needs to be compared with cellular-level FRET efficiency to determine whether Nanothread-2 effectively induces the aggregation of Nanothread-1 in vivo.
5. The excessive use of obscure words should be Avoid. Furthermore, this manuscript requires overall refinement.

Reviewer #3 (Remarks to the Author):

The binding of therapeutic antagonists to their receptors often fails to translate into adequate manipulation of downstream pathways. To address this issue, the authors devised a strategy that stitches cell surface 'patches' to promote receptor clustering. The patches contain two interactable polymer nanothreads. Nanothread-1 strings together adjacent receptors and presents decoy receptors. Nanothread-2 targets these decoys multivalently, intertwining with Nanothread-1 into a coiled-coil network. These create a force that clusters receptors to disrupt their signal transduction. Using this strategy, the authors cross-linked P-BS (CXCR4-binding sequence)-CM1 and P-CM2 and antagonized CXCR4 on breast cancer cells by the product (P-BS-C1/P-CM2), inhibiting their metastases and infiltration of regulatory T cells (Tregs) and myeloid-derived suppressor cells (MDSCs) and increasing infiltration of CD8+ T cells. These results are interesting and important; however, the major following concerns must be addressed by the authors.

1. The authors should include the data showing that BS (CXCR4-binding sequence) and P-BS-C1/P-CM2 act on CXCR4 but not other chemokine receptors, including CXCR2 and CXCR5.
2. The authors should compare the effects of P-BS-C1/P-CM2 treatment with those of treatment of known CXCR4 antagonist AMD3100 on metastasis of breast cancer cells and infiltration of immune cells in the tumors.

Minor points

1. The authors should mention that they antagonized CXCR4 on breast tumors in the abstract.
2. Fig. 3e; The difference between upper and lower panels seems small and not impressive.

Editorial Note: Schematics in the figures in the following pages were created with BioRender.com.

Point-by-Point Response

We are grateful for the reviewers' support and assistance in strengthening our manuscript. As suggested, we have performed additional experiments to confirm and validate our findings. We have also extended relevant discussions to provide a more complete interpretation of the data. Our point-by-point response are list below. The changes/additions to the manuscripts are highlighted in red.

Reviewer #1: In this work, the authors developed a strategy that the nanowire supramolecular network patch rearranged cell-surface receptors to improve therapeutic outcomes. In antagonizing chemokine receptors on cancer cells, the patch blocked the metastasis cascade, reversed immunosuppression, and enhanced photodynamic immunotherapy to eliminate spontaneous and diffuse metastasis. However, there were some key experiments that should be added. Thus, I recommend a major revision.

1. The traction of the aggregation receptors produced by nanothread-1 and nanothread-2 should be characterized.

Thank you for the suggestion. To directly visualize and semi-quantitatively characterize the traction and formation of CXCR4 receptor aggregation induced by sequential delivery of Nanothread-1 and Nanothread-2, we have established EGFP-CXCR4-expressing 4T1 cells through plasmid transfection. We then treated these cells with free antagonist (BS), multivalent polymer-antagonist conjugate (P-BS), or consecutive nanothreads (P-BS-CM1→P-CM2), imaged the cells under confocal microscopy, and analyzed the cell surface fluorescence of EGFP-tagged CXCR4 receptors (Fig. 3c). We showed that neither BS itself nor did P-BS obviously perturb the spatial organization of CXCR4 receptors on cell surface. Interestingly, being different from BS and P-BS, P-BS-CM1→P-CM2 effectively aggregated adjacent CXCR4 on the cell membrane, exhibiting adequate traction force to successfully induce CXCR4 clustering.

Accordingly, we have re-organized the results in Fig. 3, and made the following changes as shown below:

Manipulation of receptor clustering. Having demonstrated the interchain assembly, we next examined the cell-surface distribution of sequentially delivered Nanothread-1 and Nanothread-2. Initially, the hydrophilicity and macromolecular size of P-CM1-Cy5 resulted in minimal cellular uptake (Fig. S2). After multivalent modification with BS, P-BS-CM1-Cy5 showed a uniform ring-like binding pattern around 4T1 cell surfaces (Fig. 3a). We further demonstrated high specificity of BS for CXCR4 rather than other chemokine receptors frequently observed in breast cancer (Fig. S3a). Moreover, P-BS-CM1-Cy5 exclusively bound to CXCR4 receptors on 4T1 cells (Fig. S3b). Notably, consecutive P-BS-CM1-Cy5→P-CM2-Cy3 delivery displayed highly overlapped fluorescence on cell surface (Fig. 3b), validating in situ self-assembly. Most significantly, subsequent addition of P-CM2-Cy3 drastically altered the pre-targeted distribution of P-BS-CM1-Cy5, leading to speckled aggregates with enlarged punctate fluorescence on cell membrane (Fig. 3b). This observation strongly indicates that netlike crosslinking of Nanothread-2 with CXCR4-anchored Nanothread-1 formed cell surface 'patches' that provoked extensive receptor drifting.

To directly confirm whether the cell surface 'patching' through nanothreads induced CXCR4 clustering, we established enhanced green fluorescent protein (EGFP)-CXCR4-expressing 4T1 cells through plasmid transfection. Confocal microscopic analysis of EGFP-CXCR4-transfected cells revealed the presence of CXCR4 receptors both in the cytoplasm and on the cell surface (Fig. 3c and Fig. S4). In individual cells, CXCR4 fluorescence was predominantly distributed uniformly on the cell membrane, interspersed with several small concentrated fluorescent spots (Fig. 3c). This aligns with previous findings that G protein-coupled receptors exist in equilibrium between monomers and oligomers^{6,7}. It should be noted that CXCR4 fluorescence could be enriched at the contact interface between two cells (Fig. S4). Thus, to accurately reflect

changes to surface CXCR4 spatial organization after treatments, we deliberately imaged singly dispersed cells. Similar to untreated controls, cells treated with free BS antagonist displayed a ring-pattern of EGFP-tagged CXCR4 surrounding the cells. They exhibited no observable effects on escalating the degrees of fluctuations in the curves depicting distance-dependent EGFP-tagged CXCR4 fluorescent intensity on the cell membrane (Fig. 3d). Treatment with P-BS resulted in moderate changes in the fluorescent fluctuations around EGFP-CXCR4-transfected cells, whereas the traction from a single multivalent polymer was insufficient to cause noticeable receptor aggregation (Fig. 3e). Interestingly, being different from BS and P-BS, sequential nanothreads P-BS-CM1→P-CM2 effectively aggregated adjacent CXCR4 into receptor condensates on the cell membrane. This exhibited adequate traction to successfully induce CXCR4 clustering, accompanied by substantial fluctuation in cell surface fluorescent intensity (Fig. 3f). These results directly demonstrated that P-BS-CM1→P-CM2 could considerably perturb the spatial organization of CXCR4 receptors on cell surface.

Fig. 3 | Manipulation of cell surface CXCR4 clustering. **a, b**, Confocal microscopy images of 4T1 cells sequentially treated with **a**, P-BS-CM1-Cy5 for 1 h followed by cell culture medium for another 1 h; or **b**, P-BS-CM1-Cy5 for 1 h followed by P-CM2-Cy3 for another 1 h. Blue indicates cell nuclei, red indicates Cy5, green indicates Cy3, yellow shows overlay of Cy5 and Cy3. Scale bars, 20 μ m. **c-f**, Confocal microscopy images and

fluctuations curves depicting distance-dependent EGFP-tagged CXCR4 fluorescent intensity on the cell membrane after EGFP-CXCR4-transfected cells were either **c**, left untreated or treated with **d**, free BS for 1 h followed by cell culture medium for another 1 h, **e**, P-BS for 1 h followed by cell culture medium for another 1 h, or **f**, P-BS-CM1 for 1 h followed by P-CM2 for another 1 h (1 mg/mL BS equivalence, CM1:CM2=1:1 mol%) Scale bar, 20 μm . **g**, Intracellular calcium levels in CXCL12 stimulated 4T1 cells in response to escalating concentrations of free BS, P-BS, and sequential P-BS-CM1→P-CM2 ($n = 3$ biologically independent samples). Untreated 4T1 cells were stimulated with CXCL12 for 1 h. For free BS and P-BS groups, 4T1 cells received the treatments for 1 h, followed by culture in fresh medium for 25 h, prior to CXCL12 stimulation and analysis. For the P-BS-CM1→P-CM2 group, 4T1 cells underwent consecutive treatment with P-BS-CM1 for 1 h and P-CM2 for 1 h, then culture in fresh medium for 24 h, before CXCL12 stimulation and analysis.

2. In order to prove that the equimolar mixture CM1-to-CM2 was optimal, the signal of coiled-coil formation, supramolecular size, rheological properties and the morphology of the material coil should be tested.

Thank you for the constructive suggestion. We have performed additional characterizations, including coiled-coil formation, supramolecular size, rheological property, and morphology studies, to demonstrate that equimolar CM1-to-CM2 mixture of P-BS-CM1 and P-CM2 was advantageous in facilitating coiled-coil assembly between the two nanothreads compared to higher (3:1) or lower (1:3) CM1-to-CM2 mixture ratios.

The updated results are presented as Fig. 2d-h, shown as below:

To confirm the interchain biorecognition, Förster resonance energy transfer (FRET) pair, Cy5 and Cy3, were conjugated to P-BS-CM1 and P-CM2, respectively. Fluorescent spectra at Fig. 2c showed that irradiation of P-BS-CM1-Cy5 + P-CM2-Cy3 with equimolar CM1-to-CM2 mixture at the Cy3 excitation wavelength of 550 nm gave rise to a peak at the Cy5 emission wavelength of 670 nm, clearly indicating an energy transfer due to the close interactions of the two polymers. Since P-BS-CM1 and P-CM2 have similar polymer lengths (molecular weights) but differ in coiled motif valency, mixing them at various CM1-to-CM2 ratios may lead to distinct structure formations (Fig. 2d). (1) When the CM1 ratio is excessively high, P-CM2 exhibits a preference for acting as a backbone scaffold for branched polymers, allowing the grafting of multiple P-BS-CM1 while also leaving a partial proportion of P-BS-CM1 unbound. (2) At an optimal ratio, multiple chains of P-BS-CM1 and P-CM2 can crosslink and self-assemble into a supramolecular network. (3) At a lower CM1 ratio, steric hindrance and motif accessibility increase the likelihood of P-BS-CM1 and P-CM2 forming dual-chain structures. To characterize polymer assembly with varying CM1-to-CM2 ratios while maintaining a constant total polymer concentration, three ratios (3:1, 1:1, and 1:3) were selected for analysis. The complementary CM1 and CM2 pentaheptad peptides were designed to enable coiled-coil assembly through optimized hydrophobic cores, electrostatic interfaces, and helical propensities¹⁸⁻²⁰, as shown by the wheel diagram (Fig. 2a). As expected, circular dichroism spectra showed that P-BS-CM1 and P-CM2 individually were unfolded, whereas significant signal of coiled-coil formation (minima at 208 and 222 nm) was observed upon their CM1-to-CM2 equimolar mixture (Fig. 2e). Comparatively, P-BS-CM1 and P-CM2 mixtures at the imbalanced ratios (CM1:CM2=3:1 and 1:3) generated impaired coiled-coil signals. Relative to the individual nanothreads (~20 nm), dynamic light scattering revealed spontaneous expansion of P-BS-CM1+P-CM2 (CM1:CM2=1:1) into supermolecules larger than 1000 nm. Meanwhile, the major peaks of P-BS-CM1 and P-CM2 mixture at the imbalanced ratio (CM1:CM2=3:1 and 1:3) shifted to ~500 nm and ~30 nm, respectively, forming smaller assemblies (Fig. 2f). Additionally, rheological measurements demonstrated sol-to-gel transition for P-BS-CM1+P-CM2 (CM1:CM2=1:1), which was in contrast to polymers alone or their unequal CM1-to-CM2 molar mixtures (Fig. 2g). Scanning electron microscopy revealed a transformation from a flat texture of the polymer alone to an organized, porous, net-like morphology upon gelation of P-BS-CM1+P-CM2 (CM1:CM2=1:1) (Fig. 2h). The morphology of P-BS-CM1+P-CM2 (CM1:CM2=1:3) remained close to that of polymer alone, while P-BS-CM1+P-CM2 (CM1:CM2=3:1) showed an inhomogeneous morphology, resembling an intermediate state between a flaky texture and porous structure. Together, these data demonstrate that Nanothread-1 and

Nan thread-2 can spontaneously assemble into supramolecular networks through coiled-coil interactions when appropriately mixed.

Conjugates	Mw (kDa)	PDI	ζ -potential (mV)	Size (nm)	BS content (wt%)	No. of BS grafts per polymer	CM content (wt%)	No. of CM grafts per polymer	Cy3, Cy5, or PS content (wt%)
P-Cy5	107.6	1.22	-3.72±1.31	8.6±1.5	-	-	-	-	0.92
P-BS	117.9	1.24	-15.53±1.45	15.7±0.8	11.1	~6	-	-	-
P-BS-Cy5	121.1	1.44	-17.56±0.58	14.8±1.6	11.7	~6	-	-	0.87
P-BS-CM1	131.1	1.39	-5.67±0.79	17.5±2.1	10.9	~6	12.2	~4	-
P-BS-CM1-Cy5	133.7	1.46	-6.27±1.21	20.1±0.7	10.8	~6	13.2	~4	0.82
P-CM2	151.2	1.21	-8.73±0.34	19.2±0.7	-	-	32.5	~12	-
P-CM2-Cy3	149.4	1.31	-9.21±0.67	20.4±0.5	-	-	34.8	~12	0.82
P-CM2-Cy5	148.8	1.27	-8.94±0.45	19.8±0.7	-	-	32.7	~12	0.77
P-PS	115.4	1.25	-5.02±2.19	22.2±1.7	-	-	-	-	7.12
P-BS-PS	128.9	1.45	-18.24±2.31	22.4±1.4	11.1	~6	-	-	6.71
P-PS-CM2	155.4	1.22	-11.21±1.24	25.8±1.2	-	-	31.4	~12	6.34

Fig. 2 | Coiled-coil mediated self-assembly into supramolecular networks. **a**, Illustration of Nanothread-1 and Nanothread-2 components and helical wheel representation of their antiparallel heterodimeric coiled-coil interaction. CM1 and CM2 were designed with complementary pentaheptad sequences to enable coiled-coil assembly, incorporating stabilizing hydrophobic, electrostatic, and helical propensity effects. The hydrophobic core comprised valine (V) and leucine (L) at the *a* and *d* positions, with one buried asparagine (N) polar substitutions at one position *a* for CM1 and *d* for CM2, imposing antiparallel alignment. Positively charged lysine (K) at *e* and *g* in CM1 enabled electrostatic interactions with negatively charged glutamic acid (E) at *e* and *g* in CM2. Oppositely charged substitutions at one *g* improved orientation specificity. Uncharged serine (S) and alanine (A) at *b* and *c* promoted solubility and helicity. Balanced glutamic acid (E) in CM1 and lysine (K) in CM2 at *f* maintained net charge. N-terminal tetrapeptide spacers enabled polymer conjugation. **b**, Characterization of copolymers. **c**, Fluorescent spectra of P-BS-CM1-Cy5, P-CM2-Cy3, and their CM1-to-CM2 equimolar mixture. The spectra were recorded at the Cy3 excitation wavelength of 550 nm, 25 °C in 10 mM PBS, pH 7.4 after 10 min mixture. **d**, Schematic illustration of coiled-coil interaction between P-BS-CM1 and P-CM2 at various CM1-to-CM2 ratios, leading to distinct formations including branched polymer, supramolecular network, and dual-chain structure. **e-h**, Circular dichroism spectra (e), size distribution (f), frequency-dependent rheology (g), and scanning electron microscopy images (h) of individual and mixed P-BS-CM1 and P-CM2 (CM1:CM2=3:1, 1:1, 1:3).

3. The effects of simultaneous delivery and post-assembly delivery of both nanothread-1 and nanothread-2 should be investigated.

Thank you for the suggestion. Sufficient circulation and efficient tumor localization are essential prerequisites for tumor cell surface 'patching' using nanothreads. To assess the influence of delivery sequences on the overall pharmacokinetics and tumor accumulation of Nanothread-1 and Nanothread-2, we conducted additional experiments. These investigations included consecutive delivery, simultaneous delivery, and post-assembly delivery approaches. Our results demonstrated that simultaneous delivery and post-assembly delivery led to a compromised effect, resulting in a significant decrease in the circulation half-life and tumor accumulation of both Nanothread-1 and Nanothread-2 compared to consecutive delivery.

Relevant results have been added as Fig. 4e-h in the revised manuscript, shown as below:

Sufficient circulation and efficient tumor localization are essential prerequisites for achieving receptor clustering through nanothread assembly at the tumor cell surface. As shown in Fig.4e-h, we compared the effects of consecutive delivery, simultaneous delivery, and post-assembly delivery of both Nanothread-1 and Nanothread-2. With a 24-hour lag in consecutive delivery, the time-staggered approach enabled substantial tumor arrival of the two interactable nanothreads after sequential injection into the circulation. In stark contrast, post-assembly delivery of pre-mixed Nanothread-1 and Nanothread-2 led to a significant decrease in tumor accumulation of both nanothreads, due to the rapid clearance of large particles by reticuloendothelial system in blood. Compared with consecutive delivery, considerable decrease in circulation half-life and tumor accumulation was also observed in simultaneous delivery. This was most likely attributed to the mutual interaction and crosslinking between two nanothreads in the blood, which accelerated their clearance from circulation. These results highlighted the necessity of delivering Nanothread-1 and Nanothread-2 consecutively.

Fig. 4 | e-h, Representative fluorescence images exhibiting tumor accumulation (e, g) and pharmacokinetics evaluation (f, h) of Nanothread-1 (e, f) and Nanothread-2 (g, h) upon consecutive delivery, simultaneous delivery, and post-assembly delivery in orthotopic breast cancer mouse models. Whole body real-time fluorescence imaging was performed over 72 h post-injection utilizing an IVIS optical imaging system. Fluorescent images of excised tumors were captured at the endpoint and semi-quantified ($n=3$ biologically independent mice). PK parameters of statistical moment analysis were calculated using DAS 2.0 software ($n=5$ biologically independent mice). $T_{1/2Z}$: half life, AUC: area under curve, MRT: mean residence time. Data are mean \pm SD. Statistics by one-way ANOVA with Tukey's multiple comparisons test. * $P < 0.05$, ** $P < 0.01$, *** $P < 0.001$, **** $P < 0.0001$.

Q. 4: In the immunofluorescent images of Fig. 4a and 4b, the expression localization of α -SMA, hypoxia and HIF-1 α in cells was not evident, and the experimental data should be checked and re-provided. Quantitative data for Fig. 4b and 4c should be provided.

As the reviewer suggested, we have re-performed the relevant experiments and re-provided the updated results in Fig. 6b-d and Fig. SX with both representative images and quantification data, as shown below.

(a) To strengthen the investigation of tumor desmoplasia in original Fig. 4a, we have added the results of immunofluorescent images of intratumoral α -smooth muscle actin (α -SMA), a cancer-associated fibroblast (CAF) marker, and fibronectin, an extracellular matrix (ECM) component produced by CAFs after various treatments (Fig. 6b). Collagen levels have also been quantified using hydroxyproline assay (Fig. 6b). We have further provided high-resolution histological images of fiber collagens using Masson staining and Sirius red staining (Fig. SX).

Investigation of tumor desmoplasia revealed that, compared to free BS and P-BS, P-BS-CM1 \rightarrow P-CM2 resulted in reduced collagen deposition in the tumor matrix as evidenced by the lowest levels of intratumoral

α -smooth muscle actin (α -SMA, a cancer-associated fibroblast (CAF) marker), fibronectin (an extracellular matrix component produced by CAFs), collagen hydroxyproline, and fibrosis (Fig. 6b and Fig. S11).

Fig. 6 | Interference with spontaneous metastasis cascade to distant lung. Spontaneous lung metastasis mouse models of BALB/c mice bearing orthotopic 4T1 tumors received three cycles of free BS, P-BS, or sequential P-BS-CM1→P-CM2 (24 h lag) on days 7, 14, and 21. Analyses occurred on day 28: **a**, Schematic illustration showing CXCR4-mediated metastasis cascade. **b**, Representative immunofluorescent images of α -SMA (CAF marker, green) and fibronectin (produced by CAF, red), and quantification analysis of hydroxyproline (unique collagen amino acid) in tumor tissues ($n = 5$ biologically independent mice). Scale bar, 50 μm . Data are mean \pm SD. Statistics by one-way ANOVA with Tukey's multiple comparisons test. * $P < 0.05$, ** $P < 0.01$.

Supplementary Figure 11. Representative histological images of fiber collagens using Masson staining and Sirius staining. Spontaneous lung metastasis mouse models of BALB/c mice bearing orthotopic 4T1 tumors received three cycles of free BS, P-BS, or sequential P-BS-CM1→P-CM2 (24 h lag) on days 7, 14, and 21. Analyses occurred on day 28.

(b) To improve the investigation of tumor hypoxia in original Fig. 4b, we have re-provided the immunofluorescent images and quantitative flow cytometry analysis for intratumoral HIF-1 α expression.

Intratumoral desmoplasia can constrict vasculature, decreasing oxygenation and causing hypoxia. Owing to the inhibition of fibrosis, P-BS-CM1→P-CM2 substantially decreased intratumoral hypoxia inducible factor-1 α (HIF-1 α) expression versus other groups (Fig. 6c).

Fig. 6 | c, Representative immunofluorescent images and quantitative flow cytometry analysis of intratumoral HIF-1 α expression (green) ($n = 5$ biologically independent mice). Scale bar, 50 μm . Data are mean \pm SD. Statistics by one-way ANOVA with Tukey's multiple comparisons test. * $P < 0.05$, ** $P < 0.01$, *** $P < 0.001$.

(c) To strengthen the investigation of EMT inhibition in original Fig. 4c, we have re-performed the experiments and provided both representative images of tumor tissues dually stained with vimentin and E-cadherin, and the quantitative flow cytometry analysis in Fig. 6d of revised manuscript.

With apparent E-cadherin upregulation and vimentin downregulation, P-BS-CM1→P-CM2 increased epithelial properties and decreased mesenchymal features of tumor cells (Fig. 6d), indicating EMT inhibition.

Fig. 6 | d, Representative immunofluorescent images and quantitative flow cytometry analysis of EMT markers in primary tumors ($n = 5$ biologically independent mice). Blue, nuclei; green, E-cadherin; red, vimentin. Scale bar, 50 μm . Data are mean \pm SD. Statistics by one-way ANOVA with Tukey's multiple comparisons test. ** $P < 0.01$.

5. The mechanism of P-BS-CM1→P-CM2 more efficiently depleting Tregs and MDSCs should be further investigated.

Tumor recruit Tregs and MDSCs through various interconnected mechanisms, especially tumor desmoplasia and hypoxia. Desmoplasia, characterized by the deposition of extracellular matrix components (e.g., collagen, fibronectin), provides a scaffold for the migration of MDSCs and Tregs into the tumor. Cancer-associated fibroblasts (CAFs) secrete chemokines like CCL2, CCL5, and CXCL12, which attract MDSCs and Tregs to hypoxic tumor areas. Hypoxia, in turn, activates hypoxia-inducible transcription factors that induce the production of chemokines, cytokines, and growth factors, recruiting MDSCs and Tregs. Tumor glycolysis under hypoxic conditions leads to the production of lactate, which polarizes tumor-infiltrating myeloid cells towards pro-tumorigenic MDSC phenotypes. There is a reciprocal relationship between tumor desmoplasia and hypoxia. Desmoplasia can contribute to hypoxia by impairing blood vessel formation and reducing oxygen diffusion. Hypoxia, on the other hand, can promote desmoplasia by stimulating the activation of CAFs and increasing the production of extracellular matrix components. For example, tumor hypoxia promotes TGF- β 1, which drives matrix production by CAFs, while mechanical compression of tumor blood vessels by CAFs and matrix leads to tissue hypoxia.

The CXCR4 signaling pathway plays a critical role in connecting tumor desmoplasia and hypoxia. As Rakesh K. Jaina et al reported (*Proc. Natl. Acad. Sci. U.S.A.* 2019, **116**, 4558), blocking CXCR4 alleviates desmoplasia and hypoxia in metastatic breast cancer. Our data aligns with this finding, showing decreased collagen deposition (Fig. 6b) and intratumoral hypoxia (Fig. 6c) upon CXCR4 antagonism by P-BS-CM1→P-CM2, whereas antagonizing CXCR4 by free BS or multivalent P-BS appeared to be less effective. Thus, the superior depletion of Tregs and MDSCs by P-BS-CM1→P-CM2 can be attributed to the substantial disruption of reciprocal tumor desmoplasia-hypoxia pathways as demonstrated in Fig. 6b, c. Additional mechanisms are likely contributory and warrant further research in the future.

To more comprehensively interpreting the result of immunosuppression reversal by P-BS-CM1→P-CM2, we have added the above discussion in the revised manuscript.

6. In animal experiments with P-BS-CM1→P-PS-CM2 on the enhancement of immunity by PDT, the amount of CXCL12 in the lungs should be provided.

Thank you for the suggestion. We have supplemented the relevant results as Fig. S19 shown below:

Notably, coordinated PDT and CXCR4 clustering by P-BS-CM1→P-PS-CM2 (+) led to a significant reduction in BMDC recruitment and CXCL12 levels in the lungs (Fig. S19). This effect could be largely attributed to the ability of P-BS-CM1→P-CM2 to intercept the crosstalk between the primary tumor and pulmonary PMN, resulting in normalization of lung microenvironment, as observed in Fig. 6. As a result of substantial pulmonary PMN regression, P-BS-CM1→P-PS-CM2 (+) completely halted metastasis to the lungs, whereas other groups showed various degrees of metastatic nodules in pulmonary tissues (Fig. 8b).

Supplementary Figure 19. Correlation between pulmonary BMDCs and CXCL12 after orthotopic 4T1 tumor bearing mice received three weekly treatment cycles of P-PS, P-BS-PS or P-BS-CM1→P-PS-CM2 with laser irradiation starting on Day 7 (n = 5 biologically independent mice). Analyses occurred on day 28.

7. In animal experiments, the effect of P-BS-CM1→P-CM2 and P-BS-CM1→P-PS-CM2 on blocking the metastasis cascade after lung stimulation of CXCL12 should be investigated.

Thank you for the suggestion. On the context of spontaneous metastasis, breast cancer preferentially metastasizes to lung which secretes high level of CXCL12 that acts as a chemoattractant that drives CXCR4+ tumor cells towards secondary metastatic sites (*Clin Cancer Res* 2010, 16, 2927e31). According to our previous studies (*Acta Pharm. Sin. B* 2022, 12, 3383, *ACS Nano* 2022, 16, 6064), during early progression of orthotopic 4T1 breast tumor in Balb/c mice, although lung metastasis was not detected, pulmonary pre-metastatic niches (PMN) had already been formed in the lungs, accompanied by a substantial increase in CXCL12 secretion in lung ahead of cancer metastasis (Figure below). Therefore, in our current study, by the time P-BS-CM1→P-CM2 and P-BS-CM1→P-PS-CM2 were given to spontaneous lung metastasis mouse models of BALB/c mice bearing orthotopic murine 4T1 breast tumors, there was an enrichment of CXCL12 in the lungs, with no need for additional stimulation of the lungs with extra CXCL12.

Figure. Pulmonary CXCL12 level in tumor-free healthy Balb/c mice and mice bearing orthotopic 4T1 breast tumor at early progression stage (n = 5 biologically independent mice). Data are mean ± SD. Statistics by unpaired two-tailed Student's t-test. ***P < 0.001.

8. The scale bar should be added in Fig. 3a, 3b, 3i, 4a, 5a, 7a and 7b. Please carefully check the scale bar in the figures.

We appreciate the reviewer's suggestion. We have added scale bars to these figures, and carefully checked the scale bars throughout the revised manuscript including the new data.

9. The text description should be added to the diagram of Fig. 2e.

We appreciate the reviewer's suggestion. The original Fig. 2e has been replaced by Fig. 2d with text descriptions in both diagram and figure caption.

10. In Fig. 4a, the locally enlarged area should be in the same size.

We appreciate the reviewer's suggestion. According to your suggestion in comment No. 4, we have re-provided the relevant data as Fig. S11, keeping enlarged area within the same size.

Reviewer #2: In this manuscript, the authors introduced a strategy for hierarchical CXCR4 clustering using two interactable polymer nanothreads. The process involves sequential delivery of Nanothread-1 and Nanothread-2 to assemble a netlike patch on the cell surface, integrating and clustering receptors over an expanded area. By targeting CXCR4, a receptor known for its role in tumor metastasis, and the authors propose that this approach holds great potential in inhibiting tumor metastasis. Additionally, the incorporation of photosensitizers on the nanothread-2 could induced anti-tumor immune responses. Overall, although this study presents an approach with the potential to significantly inhibit the growth and metastasis of primary tumor, the novelty of this work is lacking because they have previously introduced similar concepts. In addition, the following issues should be addressed.

1. The introductory section lacks logical coherence. Additionally, is CXCR4 clustering necessary for better inhibition of tumor cell metastasis? Has the underlying mechanism been confirmed? This forms the theoretical basis for the author's invention of this strategy.

We emphatically agree that the concerns raised by the reviewer is critical to form the theoretical basis for our proposed strategy in this study. Here, we verify the necessity of antagonism through CXCR4 clustering for better inhibition of tumor cell metastasis from the following three aspects:

(a) Background knowledge indicates that manipulating CXCR4 clustering as a promising anti-metastatic strategy is theoretically feasible, but lacks mechanistic elucidation of allosteric signaling regulation.

Cancer cell metastasis is orchestrated by coordinated migration towards chemoattractant gradients, with the chemokine CXCL12 binding to its receptor CXCR4, a G protein-coupled receptor spanning cell membranes to integrate extracellular stimuli and intracellular signaling. CXCR4 exhibits dynamic equilibrium as monomers, dimers, and higher-order assemblies (*Mol. Cell* 2018, 70, 106). This flexibility in aggregation and dissociation allows cells to correctly sense gradients, adapt their migration, and metastasize in a non-random fashion (*Trends Biochem. Sci.* 2023, 48, 156). Theoretically, spatially rearranging cell-surface CXCR4, such as simultaneously crosslinking multiple CXCR4 to form receptor condensate, can perturb the dynamic CXCR4 monomer-dimer-multimer equilibrium, thereby thwarting metastasis. So far, the only two licensed CXCR4-targeted therapies are bicyclam AMD3100 and cyclic-peptide motixafortide, both relying on single molecule-receptor interactions. Strategies beyond standard monovalent antagonism are currently lacking. It is not yet fully understood whether or how CXCR4 antagonism through the mode of receptor clustering impacts downstream signaling allosterically.

(b) Existing evidences support that CXCR4 antagonism engaging higher number of concurrently crosslinked receptors is linked to better metastasis inhibition.

Recent growing evidence has highlighted that CXCR4 antagonists capable of dimerizing or oligomerizing the receptors can better inhibit metastasis. Individual CXCR4-antagonizing peptides or liposomes functionalized with lower peptide densities failed to induce the same anti-metastatic effect as when the peptides were arrayed on a liposome surface at a defined density that mimics the distances between neighboring CXCR4 on the cell membrane (*Nat. Commun.* 2018, 9, 2612). The trivalent ligand, with a rigid linker length enabling it to reach a third CXCR4 receptor at a proximal distance, showed a higher rate of competitive inhibition compared to its bivalent counterpart (*Org. Biomol. Chem.* 2015, 13, 8734-8739). Polymer-CXCR4 antagonists, enabling tunable multivalent display of ligands, significantly suppressed CXCL12-induced cell migration compared to free antagonists (*ACS Macro Lett.* 2014, 3, 1240). Additionally, increasing the valency of these antagonists further enhances their ability to inhibit metastasis (*J. Controlled Release* 2021, 334, 248). These findings suggest that antagonists presented by nanoconstructs with diverse modes of crosslinking CXCR4 may cause specific conformational changes, differentially disrupt signal transmission, and lead to an escalation in metastasis inhibition efficiency that positively correlates with the number of engaged receptors per cluster. Therefore, complete eradication of metastasis may be warranted through CXCR4 antagonism within larger receptor cluster.

(c) Present data confirm, CXCR4 antagonism mode impacts downstream signaling, with nanothread 'patching' that assembles CXCR4 into supercluster outperforming monovalent or multivalent binding in metastasis inhibition.

Previous work by others and ourselves designed multivalent ligands based on polymers, albumins, and nanoparticles to cluster various receptors for amplified efficacy. However, receptors in a wider vicinity are beyond the reach of these nanoscale constructs, thus leaving small clusters individually scattered on cell surface, and leading to asynchronous generation of suboptimal mechanotransduction. In this study, we show the 'patching' strategy, through receptor stringing and nanothread assembling, overrides size constraints of conventional receptor-crosslinking strategies by connecting an expanded cell-surface area of CXCR4 receptors, integrating them into a supercluster, and synchronizing transmembrane signaling that intercepts metastasis cascade.

Briefly, during stepwise delivery of two interactable nanothreads (P-BS-CM1→P-CM2) that self-assemble into supramolecular network via coiled-coil interaction (Fig. 2), we show P-BS-CM1 forms a surface-bound ring pattern surrounding cancer cells through active recognition and multivalent stringing of adjacent CXCR4 (Fig. 3a), and, upon sequential netlike crosslinking, P-CM2 spatially reorganizes CXCR4-anchored Nanothread-1 from uniform distribution to condensed speckles patching on extensive cell-surface area (Fig. 3b). Compared to monovalent binding by free BS and multivalent binding by P-BS, cell surface 'patching' by P-BS-CM1→P-CM2 significantly perturbs the spatial organization of cell-surface CXCR4 and generates supercluster (Fig. 3f). Consequently, P-BS-CM1→P-CM2 profoundly interferes with downstream intracellular signal transduction related to metastasis, whereas the effects of free BS and P-BS are moderate despite having high affinity for CXCR4 (Fig. 3g). With CXCR4 antagonism upon receptor clustering escalation, superior effect of P-BS-CM1→P-CM2 is also observed in its ability to disrupt the metastasis cascade by shaping the phenotype of metastatic tumor "seeds", interrupting the "seed-soil" crosstalk, and attenuating the influence of pro-metastatic PMN "soils" (Fig. 6). As a result, we show, both in vitro and in vivo, P-BS-CM1→P-CM2 exhibits significantly higher capability to inhibit breast cancer cell metastasis compared to free BS and P-BS (Fig. 5). These results confirmed that CXCR4 antagonism mode impacts downstream signaling, with nanothread 'patching' that assembles CXCR4 into supercluster outperforming monovalent or multivalent binding in metastasis inhibition.

We appreciated the reviewer's suggestion. Accordingly, we have made the following changes:

To improve logical coherence, we have restructured the introduction to improve the clarity on establishing CXCR4 clustering-mediated antagonism as the theoretical basis for enhanced inhibition of tumor cell metastasis, as shown below:

Nearly all breast cancer mortality stems from metastasis, whereby tumor cell "seeds" selectively disseminate along chemokine gradients to organs furnished with pre-metastatic niches (PMN "soils") secreting chemokines¹⁻³. CXC chemokine receptor 4 (CXCR4), a G protein-coupled receptor spanning cell membranes to covert extracellular chemokine CXCL12 binding into intracellular signaling, is frequently hijacked by breast cancer, exerting multifaceted effects on metastatic seeds, pro-metastatic PMN soils, and their crosstalk^{4,5}. CXCR4 exists in dynamic equilibrium as monomers, dimers, and higher-order assemblies⁶. This flexibility in aggregation and dissociation allows cells to correctly sense gradients, adapt their migration, and metastasize in a non-random fashion⁷. So far, the only two licensed CXCR4-targeted therapies are bicyclam AMD3100 and cyclic-peptide motixafortide, both relying on single molecule-receptor interactions. Strategies beyond standard monovalent antagonism are currently lacking. Theoretically, spatially rearranging cell-surface CXCR4, such as simultaneously crosslinking multiple CXCR4 to form receptor condensate, can perturb the dynamic CXCR4 monomer-dimer-multimer equilibrium, thereby thwarting metastasis. However, it remains unclear whether clustering CXCR4 through artificial self-assembly or mechanical traction leads to allosteric regulation of downstream signaling network, and how it relates to final therapeutic outcome of metastasis inhibition.

Recent growing evidence has highlighted that CXCR4 antagonists capable of dimerizing or oligomerizing the receptors can better hinder metastasis⁸⁻¹². Individual CXCR4-antagonizing peptides or liposomes functionalized with lower peptide densities failed to induce the same anti-metastatic effect as when the peptides were arrayed on a liposome surface at a defined density that mimics the distances between neighboring CXCR4 on the cell membrane⁹. The trivalent ligand, with a rigid linker length enabling it to reach a third CXCR4 receptor at a proximal distance, showed a higher rate of competitive inhibition compared to its bivalent counterpart⁹. Polymer-CXCR4 antagonists, enabling tunable multivalent display of ligands, significantly suppressed CXCL12-induced cell migration compared to free antagonists^{10,11}. Additionally, increasing the valency of these antagonists further enhances their ability to inhibit metastasis¹². These findings suggest that antagonists presented by nanoconstructs with diverse modes of crosslinking CXCR4 may cause specific conformational changes, differentially disrupt signal transmission, and lead to an escalation in metastasis inhibition efficiency that positively correlates with the number of engaged receptors per cluster. Therefore, complete eradication of metastasis may be warranted through CXCR4 antagonism within larger receptor cluster. Nevertheless, the size of multivalent constructs limits the further increase of concurrently crosslinked receptors, as receptors in a wider vicinity are beyond the reach of the nanoscale scaffold. Moreover, random receptor collisions in individually scattered clusters can generate separate mechanotransduction pathways that function asynchronously, thereby impairing clustering efficiency¹³⁻¹⁵. To amplify the overall outcome, a next-generation strategy requires synchronizing CXCR4 clustering over an expanded cell surface area.

Herein, we propose a stepwise strategy for hierarchically enlarging CXCR4 clustering using two interactable polymer nanothreads (Fig. 1). The process involves sequential delivery of Nanothread-1 and Nanothread-2, akin to sewing patches on the cell surface. Nanothread-1, delivered first, comprises a polymeric string skeleton with multiple copies of two pendant segments — targeting segments that anchor to CXCR4 receptors, and random coil segments that function as decoy receptors. Nanothread-2, delivered subsequently, has an identical string skeleton grafted with multiple complementary coil segments that can form coiled-coil structures with Nanothread-1 decoys. We expect Nanothread-1 to string numerous receptors together while multivalently presenting decoy receptors on the cell surface. Nanothread-2 is then hypothesized to stretch for the decoys, concurrently crosslinking with different chains of neighboring Nanothread-1. This biorecognition is proposed to intertwine multiple Nanothread-1 and Nanothread-2 strings, assembling a netlike patch on the cell surface. This patch would connect an expanded area of CXCR4 receptors and integrate them into a supercluster that produces an amplified anti-metastatic effect.

Additionally, photosensitizers are further conjugated onto Nanothread-2 for a 'hitchhike' to tumor for targeted photodynamic therapy (PDT). The incorporation of PDT in this nanothread 'patching' system is driven by the multiple benefits of CXCR4 antagonism beyond metastasis inhibition. CXCR4 antagonism can alleviate the hypoxic microenvironment within tumors¹⁶, potentially augmenting the effectiveness of oxygen-dependent PDT. Moreover, CXCR4 antagonism can reverse the immunosuppressive microenvironment of tumors¹⁷, potentially enhancing the anti-tumor immune response induced by PDT. Presumably, nanothread 'patching' could seal the potential for spontaneous metastasis and constrain cancer cells within the primary tumor. In parallel, tumor-localized PDT may transform the tumor into an in situ vaccine by inducing immunogenic cell death (ICD), triggering local anti-tumor immune responses while establishing abscopal disseminated metastasis protection (Fig. 1).

To reiterate the novelty of this work comparing with our previous work with similar concept, we have made the following changes in Discussion.

Numerous studies demonstrate CXCR4 antagonists can hinder breast cancer metastasis⁸⁻¹². However, some antagonists ineffectively disable downstream signaling despite having high affinity for CXCR4⁸. In the present study, we have fixed this 'bug' by 'stitching patches' to connect an expanded cell-surface area of CXCR4 receptors, integrating them into a supercluster, and synchronizing transmembrane signaling that intercepts metastasis cascade. Specifically, our approach includes the following sequential step (Fig. 1). (1) Nanothread-

1 actively recognizes and multivalently binds CXCR4, stringing adjacent receptors while presenting decoys on cell surface. (2) Nanothread-2 engages these decoys and intertwines with Nanothread-1 into a coiled-coil network, clustering an expansive vicinity of CXCR4. (3) Photosensitizers 'hitchhike' on Nanothread-2 to the tumor site for targeted PDT, inducing ICD to generate an in situ vaccine. Antagonism through nanothread 'patching'-induced CXCR4 superclusters profoundly amplifies the disruption of downstream signal cascade, seals the impetus of breast cancer cells for spontaneous metastasis, and constrains cancer cells within the primary tumor. Concurrently, nanothread 'patching'-enabled CXCR4 antagonism consolidates survival pathway interference, hypoxia alleviation and immunosuppression reversal, which potentiates the tumor-localized PDT, thus initiating anti-tumor immune response against the primary tumor while also establishing an abscopal memory effect against disseminated metastasis.

Previous work by others and ourselves have reported the successful use of multivalent ligands, in situ polymerizable strategies, and retractable nano-springs based on polymers, albumins, and nanoparticles cluster various receptors (e.g., CD20^{13,26-32}, death receptor 5³³, programmed death-ligand 1^{21,34}) for amplified efficacy. A typical example is that rituximab binding CD20 alone cannot directly induce apoptosis unless the CD20-bound antibodies are crosslinked by polymeric scaffold to form receptor condensate^{27,28}, with the cell killing potency positively correlating with the number of engaged receptors¹⁵. In this study, multivalent polymer-antagonists nanoconstructs (P-BS) only moderately enhanced CXCR4 signaling disruption compared to free BS. Considering the nanoscale size, CXCR4 receptors in a wider vicinity are beyond the reach of P-BS, thus leaving small clusters individually scattered on cell surface, and leading to asynchronous generation of suboptimal mechanotransduction. We also show, the 'patching' strategy, through receptor stringing and nanothread assembling, overrides size constraints of conventional receptor-crosslinking strategies, and triggers enlarged CXCR4 clusters for augmented therapeutic benefits.

Collectively, during stepwise delivery of two interactable nanothreads (P-BS-CM1→P-CM2) that self-assemble into supramolecular network via coiled-coil interaction (Fig. 2), we show P-BS-CM1 forms a surface-bound ring pattern surrounding cancer cells through active recognition and multivalent stringing of adjacent CXCR4 (Fig. 3a), and, upon sequential netlike crosslinking, P-CM2 spatially reorganizes CXCR4-anchored Nanothread-1 from uniform distribution to condensed speckles patching on extensive cell-surface area (Fig. 3b). Compared to monovalent binding by free BS and multivalent binding by P-BS, cell surface 'patching' by P-BS-CM1→P-CM2 significantly perturbs the spatial organization of cell-surface CXCR4 and generates supercluster (Fig. 3c-f). Consequently, P-BS-CM1→P-CM2 profoundly interferes with downstream intracellular signal transduction related to metastasis, whereas the effects of free BS and P-BS are moderate (Fig. 3g). With CXCR4 antagonism upon receptor clustering escalation, superior effect of P-BS-CM1→P-CM2 is also observed in its ability to disrupt the metastasis cascade by shaping the phenotype of metastatic tumor "seeds", interrupting the "seed-soil" crosstalk, and attenuating the influence of pro-metastatic PMN "soils" (Fig. 6). As a result, we show, both in vitro and in vivo, P-BS-CM1→P-CM2 exhibits significantly higher capability to inhibit breast cancer cell metastasis compared to free BS and P-BS (Fig. 5). These results confirmed that CXCR4 antagonism mode impacts downstream signaling, with nanothread 'patching' that assembles CXCR4 into supercluster outperforming monovalent or multivalent binding in metastasis inhibition.

Our data also signify that with the coordination by P-BS-CM1→P-PS-CM2 (+), efficacious CXCR4 antagonism well suits photoimmunotherapy in simultaneous inhibition of spontaneous and disseminated metastasis. We discover CXCR4 antagonism via the 'patching' strategy has the ability to promote PDT in two ways: enhancing the repertoire of PDT and increasing tumor's susceptibility to PDT.....

..... Hopefully, our proof-of-concept work—cell surface 'patching' through stepwise actuation of receptor stringing and nanothread crosslinking—provides a flexible solution to reinvigorate some antagonists that currently suffer from poor translation of receptor binding into signal manipulation.

2. Multiple peptide segments have been modified on the Nanothread to target CXCR4 and induce receptor clustering. Multivalent exogenous peptides often exhibit higher immunogenicity, and repeated administration may lead to the production of antibodies against the peptides, potentially inducing immune clearance of the Nanothread. The author should investigate this issue.

We appreciate the reviewer's insights. Since our treatment regimen involved four cycles of Nanothread-1→Nanothread-2 interventions, we have further looked into this issue by measuring and comparing the pharmacokinetics of nanothreads at each cycle. Fortunately, accelerated clearance of nanothreads after repetitive injections was not observed. This implied that pharmacokinetics of nanothreads were not affected by repeated administration. We have included the following result as Fig. SX in the revised manuscript.

Since the nanothreads were modified with multiple copies of exogenous peptide segments which often exhibit higher immunogenicity, repeated administration may lead to the production of antibodies against the peptides, potentially inducing immune clearance of nanothreads. We further provided data showing that repetitive injection did not result in accelerated clearance or affect the pharmacokinetics of both nanothreads (Fig. S9).

Supplementary Figure 9. Pharmacokinetic analysis of Nanothread-1 and Nanothread-2 at each cycle in a four-round treatment regimen post injection of Cy5 labeled Nanothread-1 or Cy3 labeled Nanothread-2. The arrows indicated the treatment regimen. PK parameters of statistical moment analysis were calculated using DAS 2.0 software ($n=5$ biologically independent mice). $T_{1/2z}$: half life, AUC: area under curve, MRT: mean residence time. Data are mean ± SD.

3. CXCR4 is a G-protein coupled receptor with seven transmembrane domains, highly expressed in various cells, including lymphocytes, endothelial cells, epithelial cells, hematopoietic stem cells, stromal fibroblasts, and cancer cells. The author should provide a more comprehensive presentation of the biological distribution of the Nanothread to assess its safety, rather than just showing its enrichment in tumors as in Fig 3a.

Thank you for the suggestion. In the revised manuscript, we have provided the biodistribution patterns of Nanothread-1 and Nanothread-2 within tumors and major organs, including the hearts, livers, spleens, lungs, and kidneys (Fig. S6). After 72 h post i.v. injection of Cy5-labeled nanothread (P-BS-CM1-Cy5→P-CM2 or P-BS-CM1→P-CM2-Cy5), considerable accumulation of nanothread was found in tumors, which was significantly higher than in other major organs. This result correlated with previous studies that HPMA polymer-based nanovehicles have excellent tumor targeting capability (*Adv. Funct. Mater.* **2020**, 1908961).

Additionally, as the reviewer mentioned that CXCR4 can be expressed in various cells besides cancer cells, we further investigate whether there is potential off-target toxicity after P-BS-CM1→P-CM2 treatment. Four cycled treatments with P-BS-CM1→P-CM2 were given to healthy Balb/c mice, and then biosafety profiles were thoroughly examined. Serum biochemistry analysis, including albumin (ALB), alkaline phosphatase (ALP), alanine transaminase (ALT), aspartate aminotransferase (AST), creatine kinase isoenzymes-MB (CKMB), glucose (GLUC), lactate dehydrogenase (LDH), total protein (TP), and urea, alongside hematological cell status of erythrocytes, leukocytes, and platelets, revealed no significant differences between P-BS-CM1→P-CM2 group and the control group treated with saline (Fig. S7). Meanwhile, histological analysis of major organs in mice receiving P-BS-CM1→P-CM2 showed no pathological abnormalities in the heart, liver, spleen, lung, and kidney (Fig. S8), indicating no obvious systemic toxicity.

Relevant results were added into the revised manuscript, as shown below:

We next investigated the biodistribution of both nanothreads. After 72 h post injection of Cy5-labeled nanothread (P-BS-CM1-Cy5→P-CM2 or P-BS-CM1→P-CM2-Cy5), considerable accumulation of either Nanothread-1 or Nanothread-2 was found in tumors, which was significantly higher than in other major organs (Fig. S6). This result correlated with previous studies that HPMA polymer-based nanovehicles have excellent tumor targeting capability²¹. We also confirmed the biosafety of four-cycled weekly treatment with P-BS-CM1→P-CM2 on healthy mice as serum chemistry analysis and hematological cell counts showed no significant differences compared to the control group treated with saline (Fig. S7). Moreover, histology analysis of major organs revealed no pathological abnormalities or tissue damage (Fig. S8), indicating no obvious off-target toxicity.

Supplementary Figure 6. Biodistribution analysis of **a**, P-BS-CM1-Cy5→P-CM2 (24 h time lag) or **b**, P-BS-CM1→P-CM2-Cy5 (24 h time lag) after 72 h post-injection of Cy5-labeled nanothread-1 or nanothread-2. n = 3 biologically independent mice. Data are mean ± SD.

Supplementary Figure 7. Serum chemistry, and hematological cell studies at endpoint after tumor-free healthy mice received four-cycled weekly treatment with P-BS-CM1→P-CM2. n = 4 biologically independent mice. Data are mean ± SD.

Supplementary Figure 8. Hematoxylin and eosin staining of major organs (heart, liver, spleen, lung, kidney) at endpoint after tumor-free healthy mice received four-cycled weekly treatment with P-BS-CM1→P-CM2.

4. Figure 3C illustrates that Nanothread-1 and Nanothread-2 bind to the surface of tumor cells, but it cannot be concluded that they form the expected structure leading to receptor clustering. Tissue-level FRET efficiency needs to be compared with cellular-level FRET efficiency to determine whether Nanothread-2 effectively induces the aggregation of Nanothread-1 in vivo.

Thank you for the suggestion. In Fig. 3, we present compelling in vitro evidence supporting the crucial role of multivalent P-BS-CM1-mediated receptor binding (Fig. 3a) and sequential P-CM2-mediated nanothread crosslinking (Fig. 3b) in perturbing the spatial organization of cell-surface CXCR4 and generating superclusters (Fig. 3c-f). These events ultimately enhanced interference with downstream calcium influx in response to CXCL12 stimulation (Fig. 3g). Prior studies have demonstrated that stimulation of G-protein-coupled CXCR4 by CXCL12 activates G-proteins, eliciting extracellular calcium influx. Given that CXCR4 clustering directly correlates with interference of G-protein function to regulate calcium homeostasis, the efficiency of inhibiting calcium influx after CXCL12 stimulation logically reflects the degree of CXCR4 clustering.

In this revision, to investigate whether this process can be recapitulated in vivo, we conducted consecutive treatments of P-BS-CM1-Cy5 on Day 0 and P-CM2-Cy3 on Day 1 in 4T1-tumor bearing mice. On Day 2, tumor cells were isolated from excised tumors for flow cytometry analysis, including assessing cell binding through Cy5 fluorescence and nanothread crosslinking through FRET signal between Cy3 and Cy5. Furthermore, the effect on downstream calcium influx was measured after CXCL12 stimulating of isolated cells.

Mirroring the in vitro trend, in vivo results (Fig. 4d) showed isolated cells from P-BS-CM1-Cy5→P-CM2-Cy3-treated tumors exhibited significantly higher Cy5 fluorescence than tumor cells receiving P-CM1-Cy5→P-CM2-Cy3 treatment, which lacked receptor binding capability. Additionally, P-BS-CM1-Cy5→P-CM2-Cy3-treated cells generated a considerably stronger FRET signal than cells treated with P-BS-Cy5→P-CM2-Cy3, which lacked coiled-coil interaction to induce nanothread crosslinking. Moreover, P-BS-CM1-Cy5→P-CM2-Cy3 demonstrated a greater efficacy in inhibiting calcium influx compared to the other two controls. Although we did not directly visualize receptor clustering in vivo due to technique limitations, the observed effects on upstream binding and crosslinking events as well as downstream signaling suggest that P-BS-Cy5→P-CM2-Cy3 treatment might promote higher-order CXCR4 assembly rather than just multivalent binding.

Accordingly, relevant results were added into the revised manuscript, as shown below:

Following a consecutive delivery of P-BS-CM1-Cy5→P-CM2-Cy3, their in vivo biorecognition was confirmed by the substantial overlap of dual fluorescence and conspicuous generation of FRET signal in tumor tissues (Fig. 4c). Compelling in vitro evidence showed that multivalent P-BS-CM1 binding to CXCR4 receptors (Fig. 3a) followed by P-CM2 crosslinking (Fig. 3b) created cell-surface CXCR4 superclusters (Fig. 3f) that enhanced downstream interference of calcium influx associated with CXCL12 signaling (Fig. 3g). We further confirmed that this process could be recapitulated in vivo (Fig. 4d). Following an identical in vitro trend, tumor cells isolated from mice treated with P-BS-CM1-Cy5→P-CM2-Cy3 exhibited significantly higher Cy5 fluorescence compared to P-CM1-Cy5→P-CM2-Cy3 lacking receptor binding capability. Additionally, P-BS-CM1-Cy5→P-CM2-Cy3 generated a considerably stronger FRET signal than P-BS-Cy5→P-CM2-Cy3 lacking coiled-coil interaction. Moreover, P-BS-CM1-Cy5→P-CM2-Cy3 demonstrated a greater efficacy in inhibiting CXCL12-stimulated calcium influx compared to the other two controls. While direct in vivo visualization of receptor clustering was technically constrained in this study, the observed effects on upstream binding and crosslinking events as well as downstream signaling suggest that P-BS-Cy5→P-CM2-Cy3 treatment might promote higher-order CXCR4 assembly beyond multivalent binding.

Fig. 4 | d, Evaluation of receptor clustering in vivo by comparing the cellular (left) and tumoral levels (right) of Nanothread-1 binding, Nanothread-2 crosslinking, and downstream calcium influx interference (n = 3 biologically independent samples for in vitro studies, n = 5 biologically independent mice for in vivo studies). Data are mean ± SD. Statistics by one-way ANOVA with Tukey's multiple comparisons test. **P < 0.01, ***P < 0.001, ****P < 0.0001.

5. The excessive use of obscure words should be Avoid. Furthermore, this manuscript requires overall refinement.

Thank you for the suggestion. In our revision, we have carefully reviewed the manuscript, substituting any ambiguous terminology with clearer and more straightforward language. We have refined sentence structures, added necessary discussions or explanations, and improved the overall readability of the manuscript. These changes were made to enhance the clarity, coherence, and organization of the content.

Reviewer #3: The binding of therapeutic antagonists to their receptors often fails to translate into adequate manipulation of downstream pathways. To address this issue, the authors devised a strategy that stitches cell surface ‘patches’ to promote receptor clustering. The patches contain two interactable polymer nanothreads. Nanothread-1 strings together adjacent receptors and presents decoy receptors. Nanothread-2 targets these decoys multivalently, intertwining with Nanothread-1 into a coiled-coil network. These create a force that clusters receptors to disrupt their signal transduction. Using this strategy, the authors cross-linked P-BS (CXCR4-binding sequence)-CM1 and P-CM2 and antagonized CXCR4 on breast cancer cells by the product (P-BS-C1/P-CM2), inhibiting their metastases and infiltration of regulatory T cells (Tregs) and myeloid-derived suppressor cells (MDSCs) and increasing infiltration of CD8+ T cells. These results are interesting and important; however, the major following concerns must be addressed by the authors.

1. The authors should include the data showing that BS (CXCR4-binding sequence) and P-BS-CM1/P-CM2 act on CXCR4 but not other chemokine receptors, including CXCR2 and CXCR5.

We appreciate the reviewer’s suggestion. To validate the BS specificity to CXCR4, we performed a dose-dependent chemokine receptor occupation assay (Fig. S3a) and a chemokine receptor competitive binding assay (Fig. S3b). We chose CXCR7, CXCR2, CXCR5, and CCR4 from several chemokine receptors due to their crucial regulatory roles in breast tumor growth, metastasis, and immune suppression (*Breast Cancer Res.* 2014, 16, 54; *Trends Mol. Med.* 2010, 16, 3; *Cancer Res.* 2009, 69, 14).

In dose-dependent chemokine receptor occupation assay, 4T1 cells were exposed to escalating concentrations of free BS. Subsequently, unoccupied chemokine receptors were individually stained using fluorescent-labeled antibodies. The well-established CXCR4 antagonist, AMD3100 (plerixafor), was included as a positive control for occupying CXCR4. Results indicated that free BS exhibited comparable CXCR4 occupation to AMD3100 at equimolar concentrations but did not occupy CXCR7, CXCR2, CXCR5, and CCR4 in a dose-dependent manner (Fig S3a). AMD3100 not only occupied CXCR4 but also exhibited low affinity with CXCR7, which was consistent with previous studies (*Communications Biology* 2021, 4: 1113; *Agents Chemother.* 2000, 44: 1667–1673).

In chemokine receptor competitive binding assay, 4T1 cells were pre-blocked with antibodies against CXCR4, CXCR7, CXCR2, CXCR5, and CCR4, respectively. Subsequently, these cells were treated with BS-modified Nanothread-1 (P-BS-CM1-Cy5), and its cell binding was analyzed using flow cytometry. CXCR4 blockade resulted in a 40% reduction in the cell binding of P-BS-CM1-Cy5, while blocking other chemokine receptors did not affect P-Cy5-BS-CM1 binding to cells (Fig. S3b). This outcome suggested that CXCR4, rather than other chemokine receptors, was involved in the cell binding of P-Cy5-BS-CM1.

We have supplemented this result in the revised manuscript, shown as below:

Initially, the hydrophilicity and macromolecular size of P-CM1-Cy5 resulted in minimal cellular uptake (Fig. S2). After multivalent modification with BS, P-BS-CM1-Cy5 showed a uniform ring-like binding pattern around 4T1 cell surfaces (Fig. 3a). We further demonstrated high specificity of BS for CXCR4 rather than other chemokine receptors frequently observed in breast cancer (Fig. S3a). Moreover, P-BS-CM1-Cy5 exclusively bound to CXCR4 receptors on 4T1 cells (Fig. S3b).

Supplementary Figure 3. Specificity of BS to CXCR4. **a**, Dose-dependent chemokine receptor occupation assay: 4T1 cells were exposed to AMD3100 or free BS with concentrations ranging from 0.1 nM to 1 mM equivalent at 25 °C for 1 hour to occupy chemokine receptors. Unoccupied chemokine receptors (CXCR4, CXCR7, CXCR2, CXCR5, and CCR4) on cell surface were stained with corresponding PE-labeled antibodies (1:100 dilute, 4°C, 1 hour), prior to flow cytometry analysis. **b**, Chemokine receptors competitive binding assay: 4T1 cells were pre-blocked with antibodies against CXCR4, CXCR7, CXCR2, CXCR5, and CCR4, respectively (1:100 dilute, 4°C, 1 hour), and then incubated with P-BS-CM1-Cy5 at 25°C for 1 hour. The cell binding of P-BS-CM1-Cy5 was analyzed using flow cytometry. n = 3 biologically independent samples. Data are presented as mean ± SD. Statistical significance was determined by one-way ANOVA with Tukey's multiple comparisons test. ***P < 0.001.

2. The authors should compare the effects of P-BS-CM1/P-CM2 treatment with those of treatment of known CXCR4 antagonist AMD3100 on metastasis of breast cancer cells and infiltration of immune cells in the tumors.

Thank you for the suggestion. We have conducted an additional individual experiment to compare the effects of P-BS-CM1→P-CM2 and licensed CXCR4 antagonist AMD3100 on the metastasis of breast cancer cells to the lung (Fig. 5h) and the infiltration of immune cells in the primary tumor (Fig. S15). In the experiment, spontaneous lung metastasis mouse models of BALB/c mice orthotopically bearing murine 4T1 breast tumors received either weekly intravenous injection of P-BS-CM1→P-CM2 treatments for three cycles starting on days 7 or daily intraperitoneal injection of AMD3100 (2 mg/kg or 5 mg/kg) from day 7 to day 27.

Our results showed that weekly intravenous administration of P-BS-CM1→P-CM2 exerted significantly superior anti-metastasis effect compared with AMD3100 intraperitoneally given daily at the dosage of 2 mg/kg. Furthermore, P-BS-CM1→P-CM2 also reduced the number of pulmonary metastatic nodules to a greater extent than AMD3100 at the higher dosage of 5 mg/kg, although the difference was not statistically different (Fig. 5h). Regarding tumor-infiltrating immune cells, neither P-BS-CM1→P-CM2 nor AMD3100 had any impact

on promoting CD8+ T cells infiltration. However, P-BS-CM1→P-CM2 demonstrated better efficiency in decreasing the frequencies of intratumoral MDSCs and Tregs compared to AMD3100 at both doses (Fig. S15).

We further included these results in the revised manuscript, show as below:

We next sought to validate the anti-metastatic effects in the more complex in vivo setting using an immunocompetent mouse model orthotopically bearing murine 4T1 breast tumors that could spontaneously metastasize to distant lung. During three weekly treatment cycles of free BS, P-BS, or P-BS-CM1→P-CM2, primary tumor growth was monitored, with lung metastasis analyzed at the study endpoint (Fig. 5g). Examination of lung lobe sections at endpoint revealed high metastatic nodule burdens in the saline and free BS groups. P-BS modestly inhibited metastasis (~30% reduction). In comparison, P-BS-CM1→P-CM2 significantly reduced pulmonary metastases (~80% inhibition), exhibiting promising anti-metastatic activity. However, P-BS-CM1→P-CM2 failed to retard primary tumor growth. In another individual experiment (Fig. 5h), a consistent result was obtained for P-BS-CM1→P-CM2 that exhibited no inhibitory effect on primary tumor, whereas AMD3100, a licensed CXCR4 antagonist, showed a dose-dependent anti-tumor activity. Nevertheless, AMD3100 even daily given at a relatively high dose only delayed the tumor growth initially, with tumor eventually growing large in the end. In terms of metastasis evaluation, weekly intravenous administration of P-BS-CM1→P-CM2 exerted significantly superior anti-metastasis effect compared with AMD3100 intraperitoneally given daily at 2 mg/kg dose. Furthermore, P-BS-CM1→P-CM2 also reduced the number of pulmonary metastatic nodules to a greater extent than AMD3100 at the higher dose of 5 mg/kg, although the difference was not statistically different. These results demonstrate P-BS-CM1→P-CM2 can suppress spontaneous metastasis, but lacks efficacy against primary tumors.

Fig. 5 | Spontaneous metastasis inhibition. **g, h,** Spontaneous lung metastasis mouse models of BALB/c mice orthotopically bearing murine 4T1 breast tumors received **g**, three cycles of intravenously administered free BS, P-BS, or sequential P-BS-CM1→P-CM2 (24 h time lag) treatments on days 7, 14, and 21; or **h**, intraperitoneal injection of AMD3100 (2 mg/kg or 5 mg/kg) daily from day 7 to day 27. Primary tumor growth curves were recorded during the treatment period. Quantitative analysis of lung metastatic nodules was performed at the endpoint on day 28. Representative hematoxylin-eosin (H&E) histology images of lung lobe sections from each treatment group are

shown ($n = 5$ biologically independent mice). Data are presented as mean \pm SD. Statistical significance was determined by one-way ANOVA with Tukey's multiple comparisons test. * $P < 0.05$, ** $P < 0.01$, *** $P < 0.001$, **** $P < 0.0001$.

Considering the key role of CXCR4 in tumor immunosuppression, we investigated the immunomodulatory effects of P-BS-CM1 \rightarrow P-CM2 on primary tumors (Fig. S15a). Flow cytometry of immune cells in primary tumor tissues showed that immunosuppressive regulatory T cells (Tregs) and myeloid-derived suppressor cells (MDSCs) were not affected by free BS, and moderately decreased by multivalent CXCR4-binding P-BS. However, Tregs and MDSCs were more effectively depleted by P-BS-CM1 \rightarrow P-CM2, which had the repertoire to further escalate CXCR4 clustering. As Rakesh K. Jaina et al reported, blocking CXCR4 by AMD3100 decreases immunosuppression in metastatic breast tumor largely through alleviating tumor desmoplasia and hypoxia. In consistence, decreased collagen deposition (Fig. 6b) and intratumoral hypoxia (Fig. 6c) could be achieved by CXCR4 antagonism by P-BS-CM1 \rightarrow P-CM2, whereas antagonizing CXCR4 by free BS or multivalent P-BS appeared to be less effective. Thus, the superior depletion of Tregs and MDSCs by P-BS-CM1 \rightarrow P-CM2 can be attributed to the substantial disruption of reciprocal tumor desmoplasia-hypoxia pathways. Additional mechanisms not studied in this study are likely contributory as well. Moreover, P-BS-CM1 \rightarrow P-CM2 also demonstrated better efficiency in decreasing the frequencies of intratumoral MDSCs and Tregs compared to AMD3100 (Fig. S15b). However, tumor-infiltrating CD8+ T cells remained low for both AMD3100 and P-BS-CM1 \rightarrow P-CM2, aligning with previous studies showing CXCR4 blockade alone is insufficient to activate anti-tumor immunity.

The above results demonstrated that beyond interception of the CXCR4-mediated "seed-soil" metastatic cascade to reduce spontaneous tumor metastasis, escalated CXCR4 clustering by P-BS-CM1 \rightarrow P-CM2 removed physical barriers by alleviating desmoplasia and reduced immunological barriers by reversing immunosuppression. This paved the way for T cell penetration and tumor-reactive activity. However, P-BS-CM1 \rightarrow P-CM2 alone could not actively recruit CD8+ T cells into the tumor microenvironment (Fig. S15) or exert adequate therapeutic effect against primary tumor (Fig. 5g and 5h), highlighting the need for complementary T cell recruitment strategies.

Supplementary Figure 15. Spontaneous lung metastasis mouse models of BALB/c mice orthotopically bearing murine 4T1 breast tumors received **a**, three cycles of intravenously administered free BS, P-BS, or sequential P-BS-CM1 \rightarrow P-CM2 (24 h time lag) treatments on days 7, 14, and 21; or **b**, intraperitoneal injection of AMD3100 (2 mg/kg or 5 mg/kg) daily from day 7 to day 27. Quantification by flow cytometry and representative flow cytometry plots of CD3⁺CD8⁺ T lymphocytes, immunosuppressive Tregs (Foxp3⁺, gated on CD4⁺ T cells), and

immunosuppressive MDSCs (CD11b+Gr1+) in tumor tissues at the endpoint on day 28 (n = 5 biologically independent mice). Data are presented as mean \pm SD. Statistical significance was determined by one-way ANOVA with Tukey's multiple comparisons test. * $P < 0.05$, ** $P < 0.01$, *** $P < 0.001$, **** $P < 0.0001$.

3. The authors should mention that they antagonized CXCR4 on breast tumors in the abstract.

We appreciate the reviewer's suggestion. We have mentioned our approach antagonized CXCR4 on breast tumors in the Abstract.

4. Fig. 3e; The difference between upper and lower panels seems small and not impressive.

We appreciate the reviewer's suggestion. We have re-organized the original Fig. 3e, and presented the new version as Fig. 3b in the revised manuscript.

Once again, we sincerely express our deepest gratitude for all reviewers' constructive suggestions that helped to improve the quality of the manuscript significantly. We hope that reviewers will find the revised manuscript satisfactory for publication in Nature Communications.

REVIEWERS' COMMENTS

Reviewer #1 (Remarks to the Author):

The authors have conducted additional experiments and added several statements.

However, further improvements are necessary to solidify their responses, and a few details need to be amended before publication. Detailed comments are provided below.

1. In response to comment #1, in Figures 3c-f, each group presents three cell images labeled as ①-③. It is necessary to explain whether these images depict the same cell state or different states. This clarification should be clearly stated in the text and legend.
2. The data arrangement in Fig. 2 should be adjusted. The data layout from Fig. 2a to Fig. 2d is unordered, and some material characterization data should be shown in the supplementary information.
3. Fig. 3g is difficult to find and Fig. 3 should be rearranged.
4. The data in Fig. 4 and Fig. 8 are over-saturated, and the test data in Fig. 4e-h and 8e should be shown in the supplementary information.

Reviewer #2 (Remarks to the Author):

The revised manuscript is now acceptable for publication.

Reviewer #3 (Remarks to the Author):

The authors have given a satisfactory response to my concerns, improving the manuscript.

Reviewer #1 (Remarks to the Author):

The authors have conducted additional experiments and added several statements. However, further improvements are necessary to solidify their responses, and a few details need to be amended before publication. Detailed comments are provided below.

1. In response to comment #1, in Figures 3c-f, each group presents three cell images labeled as ①-③. It is necessary to explain whether these images depict the same cell state or different states. This clarification should be clearly stated in the text and legend.
2. The data arrangement in Fig. 2 should be adjusted. The data layout from Fig. 2a to Fig. 2d is unordered, and some material characterization data should be shown in the supplementary information.
3. Fig. 3g is difficult to find and Fig. 3 should be rearranged.
4. The data in Fig. 4 and Fig. 8 are over-saturated, and the test data in Fig. 4e-h and 8e should be shown in the supplementary information.

Response: We sincerely appreciate your valuable feedback on improving the organization and formatting of our revised manuscript. We have carefully considered your suggestions and made the following revisions:

- 1) We have added a clarifying statement “After treatments, singly dispersed cells (①-③) at the same normal state were selected and imaged” in the text and legend for Fig. 3c-f.
- 2) Fig. 2 has been reorganized. with the data layout in alphabetical order. The material characterization data previously included in Fig. 2 has been moved to Supplementary Information as Table S1.
- 3) Fig. 3, 4, and 8 have been optimized to reduce saturation, with aforementioned data relocated to Supplementary Information.

Reviewer #2 (Remarks to the Author):

The revised manuscript is now acceptable for publication.

Response: We sincerely appreciate your positive comments and professional review work on our article.

Reviewer #3 (Remarks to the Author):

The authors have given a satisfactory response to my concerns, improving the manuscript.

Response: We sincerely appreciate your positive comments and professional review work on our article.